



# A dynamical and thermodynamic mechanism to explain heavy snowfalls in current and future climate over Italy during cold spells

Miriam D'Errico[1,*], Pascal Yiou[1], Cesare Nardini[2], Frank Lunkeit[3], and Davide Faranda [1,4]

[1]Laboratoire des Sciences du Climat et de l'Environnement, UMR 8212 CEA-CNRS-UVSQ, Université Paris-Saclay, IPSL, 91191 Gif-sur-Yvette cedex, France
[2]Service de Physique de l'État Condensé, CNRS UMR 3680, CEA-Saclay, 91191 Gif-sur-Yvette, France
[3]Meteorological Institute, CEN, University of Hamburg, Bundesstrasse 55, 20146 Hamburg, Germany
[4]London Mathematical Laboratory, 8 Margravine Gardens London, W6 8RH, UK
[*]miriam.derrico@lsce.ipsl.fr

**Correspondence:** Miriam D'Errico (miriam.derrico@lsce.ipsl.fr)

**Abstract.** Cold and snowy spells are compound extreme events that have many societal impacts. Insight on their dynamics in climate change scenarios could help adaptation. We focus on winter cold and snowy spells over Italy, reconstructing 32 major events in the past 60 years from documentary sources. We show that despite warmer winter temperatures, some recent cold spells show abundant, sometimes exceptional snowfall amounts. In order to explain these compound phenomena, we perform

ensembles of climate simulations in fixed emission scenarios changing boundary conditions (such sea-surface temperature, SST) and detect analogs of observed events. Our results show that the response of extreme cold weather events to climate change is not purely thermodynamic nor linked to the global average temperature increase, but crucially depends on the interactions of the atmospheric circulation at mid-latitudes with the thermodynamic feedback from warmer Mediterranean temperatures. This suggests how Mediterranean countries like Italy could observe large snowfall amounts even in warmer

climates.

## 1    Introduction

Cold and snowy spells are driven by the mid-latitude atmospheric circulation through the amplification of planetary waves (Tibaldi and Buzzi, 1983; Barnes et al., 2014; Lehmann and Coumou, 2015) while they are sustained by thermodynamic effects occurring at local scales (e.g. presence of snow on the ground, availability of humidity) (Screen, 2017; WMO, 1966).

Previous studies on current and future trends in frequency and intensity of cold and snowy spells are not conclusive because of the disagreement in the definition of these events (Peings et al., 2013; Vavrus et al., 2006). If we consider separately cold spells and snowfalls, a large consensus is found in the literature: when focusing on cold spells events only, the Intergovernmental Panel on Climate Change (IPCC) fifth assessment report (Pachauri et al., 2014, Working Group 1, Chapter 4) points as "very likely" a decrease of number of ice days and low temperatures. Indeed, there is also a large consensus that average snowfall

and snow-cover are decreasing in the Northern Hemisphere (Liu et al., 2012; Brown and Mote, 2009; Faranda, 2020). These trends have been observed also for Italy in several studies. The decrease in average snowfall on Northern Italy observed in the





last decades has been linked to the increase of temperature due to global climate change ( Fig. 1) (Asnaghi, 2014; Mercalli and Berro, 2003). Similar conclusions also hold for the Alpine region (Serquet et al., 2011; Nicolet et al., 2016, 2018). For Central and Southern Italy, existing studies (Diodato, 1995; Mangianti and Beltrano, 1991) also confirm these trends. On a more general

basis, the study by Diodato et al. (2019) shows that the variability of average snowfall over Italy for the past millennium can be connected to the changes in temperature, with periods of abundant average snowfalls corresponding to generally colder periods (e.g. the little Ice Age) and warmer periods yielding to limited snow accumulations. These negative trends on average snowfall are also expected in future warmer climate emission scenarios (Pachauri et al., 2014, Working Group 1, Chapter 4).

In this study, we focus on the dynamics of the compound dynamics of extreme cold and snowy events, for which the

response to mean global change might be different than that of the individual variables (temperature and snowfall). Indeed, taking this complementary compound extreme-events point view (Zscheischler et al., 2020), some authors have found that complex interactions between thermodynamic and dynamical processes occur when cold and snowy spells occur (Deser et al., 2017; Overland and Wang, 2010; Strong et al., 2009; Wu and Zhang, 2010; Coumou and Rahmstorf, 2012; Easterling et al., 2000; Marty and Blanchet, 2012). In particular, warmer surface and sea-surface temperatures can enhance convective snowfall

precipitations under specific conditions and over regions with large availability of moisture such as the great Lake in US, Japan and Mediterranean countries (Steiger et al., 2009; Murakami et al., 1994). For Japan, Kawase et al. (2016) have shown that the interaction of the Japan Sea polar air mass convergence zone with the topography may enhance extreme snowfalls in future climates via a thermodynamic feedback. Those analyses raise a number of questions: does the anthropogenic forcing affect the frequency and/or intensity of this kind of compound events? Will the large-scale atmospheric dynamics force high impact cold-

spell events despite the thermodynamic warming signals? Will local feedbacks (i.e. warm sea-surface temperatures enhancing convective snow precipitations) play a role in increasing cold spells hazards? We focus specifically over Italy: recent cold and snowy spells in this country have caused several casualties in the population, strongly affected ground and air transportation and caused disruptions in services (meteogiornale.it, last access: 26/07/2020; ansa.it, last access: 26/07/2020). Our strategy to tackle these questions is to analyse ensemble simulations produced in a Global Circulation Model (GCM) under different

emission scenarios. We first validate the cold and snowy spells produced in a simplified GCM of intermediate complexity, i.e. Planet Simulator (PlaSim) (Fraedrich et al., 2005a, b) against those detected in the reanalysis. Then we analyse cold and snowy spell ensembles under different emission scenarios. This work is structured as follows: in Section 2 we present sources and data-sets used for the detection of compound cold and snow events over Italy. Simulation results obtained with Plasim GCM are presented in Section 3. We discuss our findings and give an outlook for future studies in Section 4.

## 2  Methods

### 2.1  Sources and data-set

Our study is based on the simulations of cold and snowy spells produced with PlaSim. In order to assess the capability of this model in reproducing the dynamics of cold spells over Italy, we have first constructed a dataset of large scale high impacts cold and snowy spells occurring over Italy during winter season. For the detection, we combine both professional and official





sources and non-professional weather-climate websites where collections of weather events are available and counter-check
their validity with station data and trusted documentary sources (Bailey, 1994; Payne and Payne, 2004).

Our documentary sources are local informal networks newspapers and periodicals (see Appendix for details); websites
[ansa.it; meteo net; meteolanguedoc.com; 3bmeteo.com; meteociel.fr; meteogiornale.it, etc.]; temperatures and hydrologi-
cal records [evalmet.it, Servizio Idrografico e Mareografico Nazionale]. At national scale, we will use the National Centers
for Environmental Prediction (NCEPv2) Reanalysis datasets to analyze the meteorological fields observed during the de-
tected events and compare them to those obtained in control runs of Plasim (see Section 3). We give a flavour of the type of
events detected by presenting a local analysis for two cities: Bologna, located in the Po valley on the Northern border of the
Apennines and Campobasso, located on the Adriatic Apennines (Fig. 2). Data for Bologna are taken by https://www.arpae.
it/documenti.asp?parolachiave=sim_annali&cerca=si&idlivello=64] and books (Randi and Ghiselli, 2013), while those used
for Campobasso by [Servizio Idrografico e Mareografico di Pescara, https://www.regione.abruzzo.it/content/annali-idrologici;
http://www.protezionecivile.molise.it/centro-funzionale/la-rete-meteo-idro-pluviometrica.html]. Figure 3 shows the amount of
snowfall, the minimum temperature near the surface and the duration of each cold spell that are recorded in Campobasso and in
Bologna between 1954 and 2018 from hydrological archives [www.arpae.it/documenti.asp]. Since these data are issued from
documentary sources, they cannot be used to perform statistical inference of the general behavior of cold and snowy spells
over these regions. Qualitatively, they indicate that extreme snowfalls have occurred in recent years and even with warmer
temperatures (Fig. 3a). For example, 50 cm to 60 cm snow height was measured in the bordering side of the coasts in Puglia
and Marche during the January 2016 event and a similar amount is recorded in the hill of Campobasso. The snowfall amounts
do not seem to yield decreasing trends, although it can be argued that the duration of the events slightly decreases and their
temperatures slightly increase. In another study performed using reanalysis and observational data, Faranda (2020) performed
yearly block maxima analyses of snowfalls over Europe, showing that contrasting trends appear for extreme snowfalls over
Italian regions.

## 2.2 Model

In this paper we use The Planet Simulator model (Fraedrich et al., 2005a, b). PlaSim is a climate model of intermedi-
ate complexity developed by the University of Hamburg and released open source (see https://www.mi.uni-hamburg.de/en/
arbeitsgruppen/theoretische-meteorologie/modelle/plasim.html). PlaSim has been applied to a variety of problems including
climate response theory (Lucarini et al., 2014), storm tracks (Fraedrich et al., 2005b), climatic tipping points (Boschi et al.,
2013; Lucarini et al., 2010a), to analyse the global energy and entropy budget (Fraedrich and Lunkeit, 2008; Lucarini et al.,
2010b), to simulate extreme European heatwaves (Ragone et al., 2018), and to analyse the late Permian climate (Roscher et al.,
2011). The horizontal resolution used in this study is about 300 km (T42, $\sim 2.8\,^\circ \times 2.8\,^\circ$) with ten vertical, non-equidistant lev-
els. The dynamical core for the atmosphere is adopted from Portable University Model of the Atmosphere (PUMA). The model
includes a full set of parameterization of physical processes such as those relevant for describing radiative transfer, clouds for-
mation, and turbulent transport across the boundary layer. The horizontal heat transport in the ocean can be prescribed or
parameterized by horizontal diffusion. The parametrization by the horizontal diffusion has a simplified representation of the



large scale oceanic heat transport and then it ameliorates the realism of the resulting climate. The atmospheric dynamical
processes are modelled using the primitive equations formulated for vorticity, divergence, temperature, and the logarithm of
surface pressure. The governing equations are solved using a spectral transform method. In the vertical, five non-equally spaced
sigma (pressure divided by surface pressure) levels are used. The model is forced by diurnal and annual cycles.

Most of the events detected in the documentary sources include snowfalls that occur near coastal areas or on low lands.
In PlaSim, snowfall accumulates for ground temperatures equal or less than zero, if surface energy balance prevents melting.
If snow melts according to the energy budget, surface temperature is kept to freezing point until snow is melted completely.
Snow cover significantly changes the surface properties like albedo and heat capacity amplifying cooling by a strong positive
feedback. On the other hand, the transition from snow to snow-free strongly enhances warming trends. Thus, zero degree can
be seen as a threshold were warming and cooling trends may be significantly intensified by a snow cover feedback.

The observational data often come as snow height values (in m), whereas models and reanalysis provide more reliable data on
snowfall and then compute snow depth in $kgm^{-2}$. Since the conversion between these two quantities imply the use of a density
which depends on the nature of the snow, there is no direct equivalence between observational data and model simulations.
However, PlaSim assumes only a single value for the density involved in this conversion ($330\,kgm^{-3}$) (Kiehl et al., 1996).

Finally, in order to analyse the atmospheric stability during snowfalls events, we will use the PlaSim convective precipitation
rate $P_c$ [mm/day] of each cloud layer, defined by

$$P_c = \frac{c_p \Delta p}{Lg\rho_{H_2O}} \frac{(\Delta T)^{cl}}{2\Delta t} \tag{1}$$

where $\Delta p$ is the pressure thickness of the layer and $\rho_{H_2O}$ is the density of water. $(\Delta T)^{cl}$ is temperature tendency and $2\Delta t$ is
the leap frog time step of the model. $L$ is either the latent heat of vaporization or the latent heat of sublimation depending on
the temperature, $c_p$ is the specific heat for moist air at constant pressure and $g$ is the acceleration of gravity.

## 2.3 Experimental set up: the simulation ensemble

The three simulations that we show in this study are performed with T42 resolution, 500 years long, with a mixed layer ocean.
To get reasonable response and present-day climate one needs to add an oceanic heat transport in the mixed layer model
(slab ocean, without motion), this can be done in several ways: for our set up we tuned horizontal diffusion ($h_{\text{diff}}$) to have a
reasonable global mean SST and a realistic response of SST and ice to the forcing ($h_{\text{diff}} = 4 \cdot 10^4$).

The first simulation is the control run (CTRL) with radiative forcing levels relative to nowadays values: $CO_2$ concentration
is set to a value of 360 ppmv, which is representative of the $CO_2$ concentration in 2000. The second simulation is based on
one of the four Representative Concentration Pathways (RCPs) developed for the climate modeling community as a basis
for long-term and near-term modeling experiments (Van Vuuren et al., 2011) (Fig. 4), the baseline emission scenarios (RCP-
8.5) (Riahi et al., 2011). While this scenario consist of increasing greenhouse forcing from 2005 to 2100, in this study the
$CO_2$ concentration is instantaneously set to 1370 ppm (corresponding to a radiative forcing of 8.5 W/m$^2$) and kept constant
afterwards. Our choice of using such a forcing is motivated by the fact that the RCP-8.5 is representative of the high range of
non-climate policy scenarios and corresponds to a high greenhouse gas emissions pathway compared to the scenario literature





(Fisher et al., 2007; Pachauri et al., 2014). The third simulation is the same as control run except adding a uniform +4 K to the sea surface temperature (4K-SST) globally (Giorgetta et al., 2012). One of the reason for choosing such extreme boundary conditions is the faster warming of the Mediterranean sea compared with the other oceans (Vautard et al., 2014) expected to
lead to ∼ +3.5K SST at the end of the 21st century (Adloff et al., 2015).

Our idea is to explore two scenarios where the radiative heat is affecting mostly the atmosphere (RCP 8.5) and where the heat is mostly stored in the ocean (4K SST) and investigate which differences in the dynamics of cold and snowy spells appear.

In order to assess the proximity of cold spells obtained in simulations to those detected from documentary sources, we will use a methodology of analogues of atmospheric circulation (Yiou et al., 2013) to find the simulation days whose SLP fields
over the large Eurasia domain minimize the Euclidean distance from the averaged cold spells detected on NCEP reanalysis. To keep the same statistical sample as that of observations, we select 32 best analogues from simulations.

## 3  Results

### 3.1  Atmospheric mechanisms of observed events

In order to track the evolution of a cold spell, we chose a set of variables capable to follow dynamics, thermodynamics and
physics of these extreme events. We use sea-level pressure (SLP) (Faranda et al., 2017; Faranda, 2020) (Fig. 5a) to track cyclonic structures, temperature at 850 hPa (T850) (Fig. 5b) to track cold air advection without surface disturbances (Grazzini, 2013), geopotential height at 500 hPa (Z500) (Jézéquel et al., 2018) (Fig. 5c) to follow the large scale circulation associated to cold spell dynamics, , and snow depth (Fig. 5d) as precipitation variable. Although cold spells cover an area containing Italian borders, the large scale dynamics needs to be tracked on a much larger region including Eurasia [70–22.5N,10W–70E]. First,
we stress that the average of the four variables over the 32 cold spells shown in figure 5 is similar to a single specific episode. The main characteristics of a cold spell associated with the SLP is the placement of a low pressure over the Mediterranean sea and two high pressure patterns over the Iberian Peninsula and Russia, causing strong cooling over Western/Northern Europe (Fig. 5a). Temperature at 850 hPa, shows the incursion of cold air coming from Scandinavia and parts of Siberia that extends into South-West Europe (Fig. 5b). This corresponds to an increase of the Z500 (here expressed in decametres, dam) troughs all
over Southern Europe (Fig. 5c). Over the region where this cold air is advected (Fig. 6a) snowfall is observed (Fig. 5d). The snow depth anomaly in this day increases snow cover all over Southern Europe including Italy at low altitude (Fig. 6b).

### 3.2  Model assessment for cold and snowy events

The first step is to ensure that the PlaSim model is capable of representing realistic cold and snowy spells. In order to compare the simulated cold spells to those identified in the documentary sources, for each simulation, the 32 best analogues of the NCEP
events are selected. We show the average of the sea-level pressure (SLP) (Fig. 7a,b,c), temperature at 850 hPa (T850) (Fig. 7d,e,f), geopotential height at 500 hPa (Z500) (Jézéquel et al., 2018) (Fig. 7g,h,i) and snow depth (Fig. 7j,k,l) fields for the 32 events identified in the three simulations and visually compare them to those extracted from NCEP (Figure 5). For all runs





we find similar pattern of the evolution of a cold spell in the same set of variables (SLP, T850, Z500, snow depth) as NCEP dataset. the SLP pattern for the control run (Fig. 7a) shows, as for the NCEP data, a deep cyclonic structure over the Balkans

and two high pressure systems located over Russia and Western Europe. Anticyclones are weaker for the RCP-8.5 (Fig. 7c) and stronger for the 4K-SST simulation (Fig. 7c). Despite the warmer ocean, cold air advection at 850 hPa (Figure 7d,e,f) in the Mediterranean basin is deeper in the 4K-SST simulation than in the RCP-8.5 one. This corresponds to a larger geopotential height wave structure (Figure 7g,h,i) with a trough on the Mediterranean sea. This also produces more snowfall over the Alps and Eastern Europe for the 4K-SST run than in the RCP 8.5 simulation (Figure 7j,k,l). We also remark that the average

snowfall amounts in the RCP 8.5 scenario are almost zero (Figure 7k) although some events actually show positive values on the Mediterranean basin.

To better quantify the degree of similarity among cold spells in space and time, we computed pairwise correlations between the events of the anomalies of three atmospheric fields (SLP, T850 and Z500) at time lags of a few (60) days before and after

the event. We construct matrices of those pairwise correlations and average them (Fig. 8a). For the scenario runs we applied the same pairwise correlation as for NCEP dataset and control run and we find correlations significantly non zero for $\sim 10$ days (Fig. 8b,c,d). We tested the significance of these correlations by bootstrapping with 1000 random samples of 32 days during winter within 120 time-lag from the sample, finding always correlation values smaller than 0.1.

### 3.3 Roles of anthropogenic forcing and climate change

Once assessed the capability of PlaSim in detecting cold spell events with a large-scale dynamics alike observations, we study the response of cold spells dynamics to $CO_2$ concentration increase and global SST warming by studying the temporal evolution of CTRL, RCP8.5 and 4K-SST simulations against the events detected in NCEP datasets. First, we perform a spatial average of both T850 (Fig. 9a,b) and snow depth (Fig. 9c,d) variables on the region where the documentary sources about the extreme events are available [35–47.5N, 7.5–20E]. Once performed the mean, we construct the anomalies either just by subtracting the

seasonal average (Fig. 9a,c) or by subtracting the seasonal average and dividing by the standard deviation (Fig. 9b,d) over the selected region at different time lags for each simulation and for the NCEP reanalysis.

Figure 9a) displays different minima of 850 hPa temperature anomaly in $CO_2$ and SST-forced runs with respect to control conditions around lag 0. Furthermore, the spread of the negative temperature anomalies among all simulations is small, implying that the dynamical evolution is comparable through different events. The minimum of the temperature anomaly for the

4K-SST ($-6.7$ °C) is slightly lower compared to all other cases ($-6$ °C in NCEP reanalyses, $-5.6$ °C in CTRL, $-4.8$ °C in RCP8.5). This is a counter intuitive result as warmer anomalies are expected under anthropogenic forcing. It can however be explained by looking at the deep geopotential ridge in Figure 7i) associated with the two large anticyclonic structures over Western Europe and Russia (Figure 7c). The anomalies of snow depth show a peak around lag 0 for each simulations and for NCEP reanalyses (1.83 kgm$^{-2}$). Although the RCP8.5 scenario shows a small amount of snow at lag 0 ($0.31 \times 10^{-2}$ kgm$^{-2}$) compared with all other cases, we find a signal during the cold spells compared to the seasonal mean (Fig. 9d).






We also find the highest snow depth in 4K-SST simulation at lag 0–5 (2.65 kgm$^{-2}$). This can be explained for Italy by the amplification of one effect related to cold air passing over warmer waters (the so called "Lake-effect snow" (Eichenlaub, 1970)). Lake-effect snow forms when a very cold winter air mass flows over relatively warmer waters of large area as for example a lake: the lower layer of air picks up water vapor from the lake surface. This warmer and wetter air rises and cools

as it moves away from the lake. These conditions form convective clouds (see Fig. 10) that transform all the moisture into snow. In our case the warm waters are represented by the Mediterranean sea that get even warmer under climate change. The cold air coming from the Scandinavian region across the warm sea becomes moist and rises over the cooler Italian Apennines mountain range. This effect is amplified by a warmer ocean in the case of 4K-SST simulation causing a cooler and a more snowy cold spell event. Given the low resolution of PlaSim simulation, we need to confirm this hypothesis with information

about atmospheric stability and see whether the 4K-SST run favor instability during cold and snowy spell events. We do this by using the convective precipitation (Kuo, 1965, 1974), whose definition (see Eq. (1) in the Experimental set up) includes the lapse rate parameter so that when the convective precipitation is lower the instability of the atmosphere is higher. In figure 11 the difference of convective precipitation between 4K-SST run and CTRL run exhibits atmospheric instability over the Ionian sea agreeing with the low pressure persistence on the corresponding area (Fig. 5a). Due to the higher instability in the 4K-SST

simulation with respect to the control run, snowfall precipitations are more intense. Other studies have pointed out the role of instability in triggering heavy snowfalls in the proximity of large water basins. For the Mediterranean sea, Faranda (2020) has shown that large Convective Available Potential Energy values are associated with more intense snowfall events in the Balkans. For Japan, Kawase et al. (2016) have shown that anthropogenic forcing may enhance extreme snowfalls in future climates via a thermodynamics feedback occurring during the interaction of polar air mass convergence zone with the Japanese topography.

We now turn the analysis of changes in the frequency of occurrence of the cold spells. The recurrence rate of cold spells in simulations is defined by the number of days that yield atmospheric features that are close to those identified in the NCEP reanalysis. We use both compound SPL-T850 anomalies and SLP-Z500 anomalies to obtain a distance to identified cold spells, and compare those distances to the closest analogues within the NCEP reanalysis. In order to check the robustness of our results against the change of minimal threshold, we use different low quantiles of the distances (Tab. 1). In Table 1 the frequencies of

the two scenarios are expressed as the frequency rate of cold spells with respect to the control simulation.

The RCP8.5 run shows a frequency comparable ($\approx 1.01$) to the control run. This means that the chance of cold event happening in this region does not decrease under anthropogenic forcing. In the 4K-SST simulation, the frequency of the cold spells shows two slightly different decreasing trends: one based on the T850 anomalies that has frequency about $\approx 0.89$ compared with the control run and the second one based on Z500 anomalies has a frequency almost similar to the control run

($\approx 0.98$). This proves that under ocean warming conditions the dynamic processes (related to the atmospheric circulation) are more favored than those due to thermodynamic processes (related to land atmosphere interactions) to determine hazardous cold spell conditions.



## 4   Discussion and conclusion

We have characterized high impact cold spells over Italy over the past 60 years by assessing their common dynamical large scale
signature. Despite the difference in duration, snow depth and intensity recorded during each event, they are all associated to the
amplification of planetary waves and cold air advection from the East. These patterns seem to play a prominent role in present
and future climate in generating hazardous cold spell conditions even in a warming climate. Our results bring two possible
outcomes (or a combination of the two) in the future: one is a decrease of heavy snowfall driven by the RCP-8.5 scenario and
the second one features an increase of heavy snowfall following the 4K-SST simulation. The discriminating factor will be the
rate of warming of the Mediterranean sea, which is expected to be faster than the oceans (Volosciuk et al., 2016; Shaltout and
Omstedt, 2015). If the Mediterranean sea will warm faster than the atmosphere, larger atmospheric instability could still trigger
heavy snowfall in the area. On the other hand, if the atmosphere will warm fast enough as in the RCP8.5 scenario conditions,
then snowfalls in the area will be suppressed. In the current climate, recent snowfall events seem to benefit from this enhanced
thermodynamics feedback through increased instability (Faranda, 2020). In our simulations this feedback is only evident in
the 4K-SST simulation. Our results therefore point to a complex response of extreme snowfalls with respect to the average
decline of snowfall and snow cover observed (Diodato, 1995; Mangianti and Beltrano, 1991; Mercalli and Berro, 2003) and
that thermodynamics feedback could still produce extreme snowfalls in future climates (Pachauri et al., 2014, Working Group
1, Chapter 4).

These conclusions are motivated by a combination of dynamical and thermodynamic analyses. Indeed, in our simulations i)
the abundance of patterns corresponding to the amplification of planetary waves does not depend on the absolute global tem-
perature but rather on the tropics-to-poles temperature difference which is then linked to ocean warming and sea-ice melting,
ii) a warmer ocean can trigger snow-lake-like effects over the Mediterranean sea during cold spell events, enhancing convective
precipitations and favoring heavy snowfalls. These combined effects show that when dealing with compound extreme events,
the thermodynamic average climate change signal must be weighed against other dynamical and physical feedbacks. This
mechanism is a robust signal and it can be generalized to other cold spells affecting other countries at mid-latitudes where
great water masses can have an impact on convection such as Japan, Korea, the region of Great Lakes of North America.

This study comes with some caveats and limitations: although we have validated the behavior of PlaSim against NCEP
reanalysis, results on frequency changes for cold spells crucially depends on the position and the destabilization of the jet
stream. It is known that different climate models have a different response of jet stream dynamics to climate change (Arctic
Amplification (Cohen et al., 2014) or Zonalization (Francis and Vavrus, 2012)). Further studies should also take into account
that snowfall amounts are better predicted using humidity and air temperature in large-scale land surface model runs, than just
using the current and past scheme used in PlaSim as well as in other general circulation models (Jennings et al., 2018).

We doubt however that an analysis of forced non-stationary simulations as those produced in scenario runs may provide
a better understanding, because of their limited duration and the inter-decadal variability superimposed to the non-stationary
signals. Furthermore, we have investigated the lake effect from the point of view of large-scale instabilities. Future studies with
regional climate simulations may focus on the robustness of this phenomenon on smaller scales.



# 5 Appendix

**Cold-spells detection**

In this appendix we describe each extreme cold event selected as a cold spell in this study. The mains characteristics of the
events are the occurrence of snowfalls in region where snow-cover has usually been rare or absent since a long time (e.g.
lowlands and coasts), the societal impact was disaster (e.g. in 2017), extreme minimum temperatures, and extreme amount of
snowfalls. The date reported at the beginning of each event is the one selected as the most representative day of each cold-spell
event and it is the one used for the analogs search. The information about the duration of the events are reported in the text for
each description.


**04th January 1954**. A cold spell rapidly built in the Mediterranean in January 1954 (an exceptional month in Spain). Heavy
snowfalls in lowland areas (Po Valley) affected all of northern Italy. In 24 hours, 60 cm of snow fell over Turin, Brescia, Milano,
Piacenza, Cremona, Reggio Emilia, Bologna and Vicenza (-5°C at 1400 m Osservatorio Meteorologico del Collegio Alberoni
of Piacenza) according to information found in the press (Resto del Carlino 05/01/1954). Many traffic blocks occurred mainly
in Piacenza and Cremona (daily local journal of Cremona 06/01/1954).

**04th February 1956**. One of the coldest and snowiest event of the 20th century in Europe. The Po Valley was below a $-15$°C
isotherm at 850 hPa (1-2/02/1956 Wetterzentrale.de (last access: 26/07/2020)), and snow storms affected the entire country.
Rome experienced a historical snowfall. A powerful extratropical cyclone embedded in very cold mid-tropospheric air core
struck the southern regions causing heavy snowfalls in Rome and throughout central and southern Italy, with blizzards and
freezing temperatures, frost and snow. In those days it snowed even on the Sicilian coasts. In Palermo, the minimum tem-
perature dropped to 0° C (daily data of Palermo and Sicily on 1956, ANNALI/A1956) and the city was blanketed by several
centimeters of snow, which also fell on the southern coasts of Sicily and the island of Lampedusa (Corriere del mezzogiorno
of 07/02/2011).

**17th December 1961**. December was a very cold month for most of Italy with a historical snowfall in southern Italy coastal
areas as in Bari (30 cm, Protezione Civile Puglia of 17/12/1961). After 3 days of heavy snowfall, a record snow height of
370 cm was reported in Roccacaramanico (1050 m of altitude) on December 20th (Annali idrologici Sezione Autonoma del
Genio Civile Pescara 12/1961) and all the Adriatic regions were affected by heavy snowfalls (meteogiornale.it (last access:
26/07/2020) of 18-12-2014).

**31st January 1962**. Sicily reported several historical records of minimal temperature as in Lentini città ($-2.5$°C, 43 m of
altitudine), Caltanissetta ($-4.5$°C), Caltagirone ($-3.2$°C), Castronovo di Sicilia/Piano del Leone ($-8.5$°C)(Osservatorio delle
Acque 01-02/1962). Heavy snowfall occurred in Palermo and Capo d'Orlando (8 m of altitude) (meteolive.it (last access:
26/07/2020) of 28/02/2002, meteosicilia.it (last access: 26/07/2020) of 07/12/2007).

**22nd January 1963**. 1963 was one of the coldest winters in the records for Western Europe. That year, frost trapped Norway's
islanders in ice and caused severe hardships to millions of Europeans. The cold primates belong to Sweden and Finland. In
northern Sweden it went below $-41.2$°C (record of Karesuando, SMHI (last access: 26/07/2020) of 19-01-2014). Average



temperatures for the month were in excess of −5°C below normal from southern England across Europe to the Urals. Warsaw reported an average temperature of −12.4°C for January, while Paris averaged −5.5°C below normal. Even Mediterranean regions averaged about −3°C below normal (James, 1963). The upper reaches of the River Thames froze thamesweb.co.uk (last access: 26/07/2020) and the lowest temperature in the Germany was measured on January 20th at Quedlinburg at −30.2°C

(Eichler, 1971). In Italy the temperature dropped due to strong bora winds (110 km/h, Annali Idrologici Ufficio Idrografico del Po 01/1963) and heavy snow settled over Friuli-Venezia Giulia (5 cm to 10 cm) reaching Venezia (The Venetian Lagoon turned into ice-pack deep 10-15 cm). Arctic temperatures (Trieste: −9°C, Udine: −10°C, Pordenone −15°C, Milano: −8°C, Bologna −7°C, Annali Idrologici Ufficio Idrografico del Po, Arpae.it (last access: 26/07/2020) 01/1963) affected all other regions of Italy causing snowstorms over Toscana, Marche, Abruzzo, Molise, Apulia, and several cities were completely isolated (me-

teogiornale.it (last access: 26/07/2020) of 21/01/2011, Randi and Ghiselli (2013), regione.abruzzo.it (last access: 26/07/2020); protezionecivile.puglia.it (last access: 26/07/2020) 01/1963).

**12th January 1968**. In January 1968 was one of the strongest cold spell that ever affected Tuscany. The cold period lasted from 9th to 15th January. Very low minimum temperatures were reached and even some highs were very cold: Città di Castello

(295 m, Ufficio Idrografico di Roma) −23°C, Arezzo (S. Fabiano) (277 m) −14.2°C, Verghereto (812 m) −15.2°C, Cortona (393 m) −8.7°c (Annali Idrologici Genio Civile Pisa 01/1968). Snow fell all over that area: in Eremo di Camaldoli (1111 m) 65 cm, in Verghereto (812 m) 60 cm, Arezzo (S. Fabiano) (277 m) 15 cm, Laterina (191 m) 18 cm also in Florence (Ximeniano) (51 m) 19 cm of snow (Annali Idrologici Genio Civile Pisa 01/1968); (La Nazione of 11/11/1968).

**28th February 1971**. On February 24th, the penetration of Arctic air into the Mediterranean sea from the eastern edge of an anticyclone extended towards the British isles, triggered heavy snowfalls and a cold spell over Italy. Initially, the event affected northern European countries. The most sensitive drops in temperatures affected Italy between the evening of February 28th and the morning of March 1st. On the morning of 1st March, almost all of Italy recorded minimum temperatures below zero even on the lowland areas: −5°C in Florence and Pisa (Annali Idrologici Genio Civile Pisa), −4°C in Rome (Ardea, Ufficio

Idrografico di Roma 02/1971), −1°C in Naples (Annali Idrologici Genio Civile Napoli 02/1971) with a snowfall that also reached the coastal areas of the city (La Stampa of 6-7/03/1971).

**1st December 1973**. Very Low temperatures leading to negative values were recorded in 1973 throughout the North and Central Italy.The isotherms reach −15°C at 850 hPa in the Alps. Cold conditions persisted for long time, yelding to low minimum temperatures during the first two weeks of December (−7C in Novara and Treviso, −6°C in Udine, but also −6.4°C in Potenza

and −5°C in Foggia, in Apulia). Temperatures decreased to −2°C in Trieste, −7°C/−8°C in Novara and Arezzo, −19°C on Monte Cimone with Nord-East wind at 133km/h. Due to these conditions, the highways remained closed in Tuscany for half a day. In Florence, 17 cm of snow fell, as it did not happen for many years. In Valle del Serchio 30 cm of snow fell where for over 40 years the snow was almost absent. Heavy snowfall (15 cm) fell in Perugia, Gubbio, Assisi, Spoleto, Sangemini. In Friuli, temperatures of −17°C were recorded in Fusine and −12°C in Tarvisio (sienanews.it (last access: 26/07/2020) 13/12/2016,

Annali Idrologici Ufficio Idrografico Magistrato delle Acque di Venezia.; Annali Idrologici Genio Civile Pisa; Annali Idro-





logici Genio Civile Catanzaro; Annali Idrologici Genio Civile Bari; Annali Idrologici Ufficio Idrografico del Po, Arpae.it (last access: 26/07/2020), 12/1973, Aeronautica.Militare (last access: 26/07/2020)).

**15th January 1979**. This cold spell affected most of Europe, including Italy. This exceptional irruption of cold air claimed numerous casualties. In Italy the arrival of cold air first manifested itself with strong winds and storms on the Tyrrhenian

coasts; subsequently there was a sharp decrease in temperature. Snowfall occurred in Tuscany (Annali Idrologici Genio Civile Pisa 01/1979), Sardinia and most of central and Southern Italy, with snowstorms in the Marche, in Abruzzo, Molise (regione.abruzzo.it (last access: 26/07/2020)) and Basilicata (evalmet.it (last access: 26/07/2020)). The most abundant snowfalls were observed on January 9th with the arrival of more temperate and humid masses, mainly from South-Western currents. Ice problems on the roads and in the pipes were reported (Resto del Carlino 13/11/1979).

**08th January 1981**. Western and Central Sicily remained blocked on 08th January 1981 for unprecedented amounts of snow and minimum temperature were recorded. That day it even snowed on the small island of Pantelleria (5 m elevation). Many cities, from Palermo to Trapani from Messina to Enna, remained isolated for whole days. The whole south of Italy was swept by an immense storm with 30 cm of snow fell on Sicilian coasts and the temperature reached a historical minimum of $-0.5$°C in Palermo where continuous snow precipitations for more than 24 hours are an exceptional event (Giornale di Sicilia 7/01/1981,

meteolive.it (last access: 26/07/2020) of 28/02/2002, Annali Idrologici Genio Civile Palermo 01/1981).

**07th January 1985**. From 1st to 15th January 1985, Italy and most of Western Europe were affected by a cold spell. The cold air (associated to a Bora wind that blew up 100km/h) favored cyclogenesis over the country and a minimum of low pressure formed between Tuscany and Lazio. This triggered historical snowfalls that abundantly interested Florence with 40 cm of accumulation (up to 80 cm in the Val di Cecina) and Rome with 30 cm. The minimum moved towards south-east between 6th and

9th January and the snow reached also Campania and the rest of the south with accumulations up to 25 cm on the hilly zones of Naples, as it had not happened since 1956 (Annali Idrologici Genio Civile Napoli 07/1985). In the Northern regions $-20$°C were registered on the Po Valley. Between 10th and 11th January further temperature records were broken in the minimum values as in Florence (Peretola, $-23.2$°C) and Piacenza (S. Damiano, $-22.2$°C)(Aeronautica.Militare (last access: 26/07/2020); valdarnopost.it (last access: 26/07/2020) of 14.01.2015, Il Mattino of 27/02/2018, firenzemeteo.net (last access: 26/07/2020)

of 19/01/2017).

**24th December 1986**. The situation for Christmas day 25th December 1986 was characterized by strong winds and 850 hPa isotherms of $-10$°C that covered most of Italian Peninsula (meteociel.fr (last access: 26/07/2020) 25/12/1986). In Pescara on the evening of December 26th, the temperature reached $-9$°C and about 15 cm of snow fell. Snowfall affected the entire Adriatic side (5 cm in Perugia, more than 30 cm in Molise). In Ancona (Falconara) wind gusts exceeded 95 km/h with a minimum

temperature of -6°C (Aeronautica.Militare, last access: 26/07/2020). The snow then reached Sardinia and even Apulia where the temperature in Bari dropped to $-1$°C (Annali Idrologici Genio Civile Bari 12/1986). A minimum (December) temperature record was held in Pantelleria with 2.6°C on December 25th (Annali Idrologici Genio Civile Palermo 12/1986, meteolive.it (last access: 26/07/2020) of 31/01/2008, meteogiornale.it (last access: 26/07/2020) of 31/12/2014).

**03th March 1987**. Cold air and stormy weather reached the extreme south/east of Italy, with the maximum peak on 8th March

1987 when the -12°C 850 hPa isotherm covered the whole of Apulia (Annali Idrologici Genio Civile Bari 03/1987). Snow fell





also in Naples, Crotone and even in Palermo. The most impressive element was the snow accumulations that were recorded in those days: at Gioia del Colle the snow reached 72 cm of accumulation, for a total of 9 days of permanence of snow on the ground (Annali Idrologici Genio Civile Bari 03/1987, 3bmeteo.com (last access: 26/07/2020) of 08/03/2019; La Repubblica of 12/03/1987, meteogiornale.it (last access: 26/07/2020) of 13-03-2005).

**31st January 1991**. The temperature cooled down under the effect of a Bora wind up and dropped to $-4.2°C$ in Trieste (Annali Idrologici Ufficio Idrografico e Mareografico di Venezia 01-02/1991). Snow fell on Bologna, Rimini, Forlì, and finally even on the Marche coastal area, with 5 cm of snow that fell in the harbour city of Ancona. During the day, the cold air moved to the west over the Po Valley, from Veneto to Piemonte, with widespread snowfalls. Minimum temperatures of $-21.2°C$ were recorded at Passo Rolle, $-12°C$ at Novara, $-11.6°C$ at Bologna (Aeronautica.Militare, last access: 26/07/2020). February 7th

is one of the coldest (Annali Idrologici Ufficio Idrografico e Mareografico di Venezia; Annali Idrologici Ufficio Idrografico e Mareografico di Parma 01-02/1991) days in the history of Northern and Central Italian climatology (Randi and Ghiselli (2013), meteoservice.net (last access: 26/07/2020) of 05/02/2016, recordmeteo.altervista.org (last access: 26/07/2020) of 01/03/2012, La Repubblica of 02/02/1991).

**01st January 1993**. The first day of the year a massive continental polar air patch, the Buran, reached Italy. Southern and Cen-
tral Italian regions were mostly affected, especially the Adriatic slopes where the snow fell also in coastal areas. The absolute minimum temperature record was broken in Bari ($-5.9°C$) (Aeronautica.Militare, last access: 26/07/2020). Snow fell in the southern part of the Italian boot (Messina and Reggio Calabria), in the north plains (Parma, Modena, Reggio Emilia) and on the Adriatic coast, from Rimini to Cattolica. Abundant snowfalls affected the Po Valley, especially Emilia, Tuscany, up to Central Italy. It also snowed in Rome, in the northern areas of the city. In the National Park of Casentino, next to Arezzo, amount of 30
cm of snow fell in few hours. At the same time was snowing simultaneously in the two sides of the Tyrrhenian and the Adriatic coasts, which is rare. The cold air moving westward caused extensive new snowfalls in the North and Central Italy. Intense cold conditions persisted for long time in the Po Valley with record-breaking temperatures, such as $-13°C$ in Milan and almost $-20°C$ in Emilia (meteolive.it (last access: 26/07/2020) of 11/11/2009, meteo.ansa.it (last access: 26/07/2020) of 17/12/2015, Corriere della Sera of 03/01/1993.


**27th December 1996**. This cold spell has also covered England and France ($-7°C$ in Paris; Le Parisien 03/01/1997) causing 200 deaths and freezing over the river Thames in London (Jordan-Bychkov and Murphy, 2008). On December 27th heavy snowfalls striked the Adriatic side of Italy from Romagna to the South. On December 29th a very heavy snowfall affected Central Italy and Southern Tuscany in unusual areas (20 cm on the Lazio coast, 35 cm in Porto Santo Stefano, Aeronautica.Militare
(last access: 26/07/2020)). On December 30th snow appeared in Milan, Como, Varese, Pavia and throughout the whole of Piedmont. A snow storm blowed in Genova. In Bolzano snow fell up 30 cm. Extremely low minimum temperatures affected the areas covered by snow (between $-10°C$ and $-15°C$ in Southern Tuscany and Umbria, Annali Ufficio Idrografico e Mareografico di Pisa 12/1996). The official weather station of the city of Arezzo (Molin Bianco, 248 m) on December 30th recorded a minimum of $-15°C$ (Aeronautica.Militare, last access: 26/07/2020), a monthly record for December from the beginning of
records (1957).(La Repubblica 27/12/1996 and 28/12/1996, meteolive.it (last access: 26/07/2020) of 20/10/2017).



**31st January 1999**. This winter was rather cold and snowy with the peak that was reached between the third decade of January and the first part of February. The freezing air reached Italy, particularly affecting the Central and Northern regions on February 5th. The snow affected all the Po Valley, from Venice to Turin with accumulations up to 30 cm or higher on the plain. The snow also fell abundantly in Forlì, Rimini, Ancona, Grosseto, Parma, Florence, Lucca and Genova. A snowstorm struck Viterbo and

snow flakes were also observed in Rome with −6°C (Annali Ufficio Idrografico e Mareografico di Roma 01-02/1999) where the fountains of the city froze, which is a rather unusual and damaging phenomenon for the historical heritage of the city. The temperatures decreased sharply and there were a few days of ice (maximum below zero) in the city. Among very low temperatures, we remark 3.8°C on 31st January at Palermo (Osservatorio Astronomico di Palermo), −12°C in Norcia (Annali Ufficio Idrografico e Mareografico di Roma) and −21°C in Dobbiaco (1213 m of altitude, (Aeronautica.Militare, last access:

26/07/2020)). The strong Bora wind gusted up to 90 km/h in Trieste on 4th February (Arpa Friuli Venezia Giulia last access: 26/07/2020 02/1999). Snow fell on Sicilian coasts and accumulated up to 5-10 cm for few hours on the beaches of the Nebrodi areas (Arpa Regione Emilia Romagna Annale Idrologico 01-02/1999, La Repubblica of 24/02/1999, meteoweb.eu (last access: 26/07/2020) of 05/01/2017).

**08th December 2001**. Before Northern Italy, the cold air reached parts of Central-Eastern Europe. On the evening of Decem-

ber 13th a snowstorm caused transport disruptions and isolated several small towns in Northern Italy. in Trieste the Bora wind blew at 116km/h with −4°C. In Tarvisio on 15th of December the temperature reached −16°C (Arpa Friuli Venezia Giulia last access: 26/07/2020 12/2001). The Po Valley appeared frozen and white. That cold spell is remembered as the famous "Blizzard of Saint Lucia". That cold spell was so intense that is long lived in the memories of the inhabitants of the Northeast, since the blizzard combined heavy snowfalls with intense wind. On that occasion, the origin of cold air masses was Eastern Europe and

Russia. Due to the strong wind, snow fell horizontally and stuck to the walls of the building, rails and guardrails of highways with heavy transport disruptions. The number of accidents caused by ice was high. The snow accumulation fluctuated between 5 and 25 centimeters on the plains, depending on the area, with greater values in Emilia-Romagna: temperatures of −16°C were recorded in Fiorenzuola (422 m of altitude), −7°C in Reggio Emilia and −5°C in Cesena on 17th and 18th of December (Arpae.it (last access: 26/07/2020)): the snow mantle was irregular due to the strong winds.(La Repubblica 17/01/2001, mete-

oweb.eu (last access: 26/07/2020) of 06/12/2011)

**20th January 2004**. During this event, icy currents flew from the Northeast towards Northern Italy, with weak snowfalls on Emilia, up to medium-low altitudes. In the following days, however, it snowed again in the North and in the Central regions at lower altitudes: snow reached Tuscany, Lazio, with snow flakes even on Rome. The temperatures in these days of January were particularly low on the north-eastern Alps (−11.2°C Dobbiaco, Protezione Civile Provincia Autonoma di Bolzano 01/2004)

and on Central and Southern Italy, where widely negative values were reached on the usually mild Tyrrhenian plains (−5.2°C at Fiumicino, −6.3°C in Rome and −4°C at Ciampino, Agenzia Regionale Protezione Civile Lazio Annali Idrologici 01/2004). The whole region of Lazio experienced particularly cold days. The cold affected Irpinia and Basilicata (with temperature below zero on the Ionian sea, (evalmet.it, last access: 26/07/2020) 01/2004), Molise, Abruzzo and APulia. The snow fell in abundance on the Murge, but also locally on the Apulian coast. During this exceptional event low temperatures were measured in these ar-

eas: temperature dropped zero in Foggia, a coastal town in Apulia, (Annali Ufficio Idrografico e Mareografico di Bari 01/2004)





and the coastal cities of Naples, Lamezia Terme and Catania. Low daily maximum temperatures (below the 10°C degrees) were recorded also on the Sicilian Tyrrhenian coast, from Messina to Trapani and even in the Syracuse area (Osservatorio delle Acque).(meteogiornale.it (last access: 26/07/2020) of 27/01/2004).

**22nd January 2005**. The cold spell hit Western and Central Europe yielding temperatures below the average for almost the
entire winter and reached its peak during the month of January. In Northern Italy abundant snow fell: in Lombardy snow-height reached 30 cm, with peaks up to 40-45 cm. Even the coastal city of Genova suffered snowfalls (Arpa Liguria Annali Idrologici 01/2004, Aeronautica.Militare (last access: 26/07/2020)). Snow fell also in Marche, Abruzzo, Campania and Basilicata where in inland areas it snowed for many days and accumulations exceeded one meter in Abruzzo, as well as in some areas of Irpinia. Snow also fell in Salerno. There were road disruptions as car drivers were trapped in the snow on the highway. Stormy weather
also affected other European countries, particularly in France and Spain. In France four avalanches detached from the mountains in Savoy caused many victims in ski resorts.(meteogiornale.it (last access: 26/07/2020) of 23/01/2005; Corriere della Sera of 25/01/2005).

**02nd March 2005**. The cold and snowy air hit Central and Southern Italy. Snow precipitated to the hills of Naples, to a medium-low altitude in Calabria, with abundant accumulations on the coasts of the Middle Adriatic (starting by the south of
the Marche to Molise). After few days, the snowfall reached the North and Tuscany (Servizio Idrologico Regionale Regione Toscana, last access: 26/07/2020). Thirty centimeters of snow fell on Liguria (Arpa Liguria Annali Idrologici 03/2005), it snowed with abundance on Milan and on most of Lombardy, on the plains of Emilia, on Piedmont, on the north of Tuscany; it snowed in Veneto whitening Verona, Venice and Rovigo. On March 1st, 2005, the lowest temperatures were recorded throughout the winter, with an average over the country of −0.5°C, which entered Italy's climatic history. The temperatures in the
Alps reached peaks of −23°C (Marcesina record of −34°C, (Arpav, Arpa Veneto, last access: 26/07/2020)), in the Apennines −20°C at Cimone, −16°C at Terminillo (Agenzia Regionale Protezione Civile Lazio Annali Idrologici 03/2005), −10.8°C at L'Aquila (Annali Idrologici Servizio Idrografico e Mareografico di Pescara 03/2005). In the lowlands or at low altitude the temperature decreased to −12°C in Piacenza, to −11°C in Novara (Arpa, Piemonte), to −10.4°C at Udine (Arpa Friuli Venezia Giulia last access: 26/07/2020) and −9°C Arezzo (Servizio Idrologico Regionale Regione Toscana, last access: 26/07/2020).
(nordestmeteo.it of 02/11/2019; meteogiornale.it (last access: 26/07/2020) of 03/03/2016).

**13th December 2007**. This cold spell was an extraordinary event characterized by abundant snowfalls, blizzards and an extreme drop in temperatures over most of the Sardinian territory, at altitudes on average above 400 m. Also noteworthy are the 2 meters accumulated over an altitude of 1000 m on the slopes of Mount Limbara. Towns were largely unprepared to manage the event. An electricity blackout affected for several hours Cagliari and the schools remained closed for two days. Difficulties
were reported in road connections: the main road of the Sardinian network of state highways suffered numerous blocks due to some trucks blocking the roads. In Nuoro, the snowfall exceeded 50 cm, breaking the record (Annali Idrologici della Sardegna 12/2007). A strong wind and rough seas were observed in Olbia (30 knots). The drop of temperatures affected also the Central and Southern Italy: −10°C degrees and icy roads were reported in Calabria, snowfalls in Molise as well as in the interland of Bari, Foggia and Taranto (Annali Ufficio Idrografico e Mareografico di Bari 12/2007), in Basilicata (Agenzia Regionale Pro-
tezione Civile Basilicata Annali Idrologici 12/2007) and temperatures below zero were recorded on the Ionian coast (evalmet.it





(last access: 26/07/2020) 12/2007).(La Repubblica of 13/12/2007, meteolive.it (last access: 26/07/2020) of 19/12/2007).

**17th December 2009**. Most of the Central and Northern Europe has been struck by this cold spell. In December 19th the snow fell in most of the areas of Northern Italy, and it was especially copious in Tuscany (Servizio Idrologico Regionale Regione Toscana, last access: 26/07/2020). A strong glazed frost occurred in Emilia and Liguria (Arpae.it (last access: 26/07/2020)).

The absolute minimum temperatures were recorded in some lowland locations (especially on December 20th) in Friuli Venezia Giulia (Arpa Friuli Venezia Giulia last access: 26/07/2020): Udine Rivolto −18°C, Pordenone −12.4 °C, Cervignano del Friuli −17.3 °C, in coastal locations as Lignano −6.3°C, in some alpine valleys as Tarvisio (754 m) −18.3°C, Fusine (850 m) −22°C. (La Repubblica 19/12/2009 ; meteogiornale.it (last access: 26/07/2020) of 19/12/2014; Il Quotidiano 19/12/2009).

**12th February 2010**. Snowfalls affected several regions, from Emilia Romagna to Calabria, from the Marche to Sardinia

(Arpae.it (last access: 26/07/2020), Annali Idrologici della Sardegna 02/2010). Bologna airport was closed for several hours: 17 canceled flights, 15 diverted ones. The heaviest snowfall in Rome (ARSIAL: Agenzia Regionale per lo Sviluppo e l'Innovazione dell'Agricoltura del Lazio, last access: 26/07/2020) since February 1986 was recorded. It caused the difficulties on ground transportation and many roads were closed both inside and outside the city. Many interventions were planned to rescue motorists involved in collisions and stuck on the highway between the Marche and Romagna (distribution of comfort items for at

least 2000 people stuck in cars). A blizzard hit the Sila region where the schools were closed for few days. (roma artigiana.it (last access: 26/07/2020) 26/02/2018, La Stampa of 12/02/2010; La Repubblica 12/02/2010; ansa.it (last access: 26/07/2020) of 12/02/2010).

**11th December 2010**. An Arctic-continental mass of air reached Italy giving rise to intense snowfalls on the Adriatic and

Thyrrenian coasts (especially the coasts surrounding the city of Livorno). Snow also whitened Tuscany (25 cm fell in Florence), Umbria and part of Lazio, with snow flakes that were observed in Rome. An exceptional snowstorm hit Ancona and the surrounding areas between 14th and 15th December. There, the Adriatic Effect Snow contributed to reach snow heights up to 30 cm in the Chieti area and 40 cm in Lanciano. The temperatures were extremely cold over most of Italy. Malpensa Milan airport measured a minimum temperature of −14°C and Rome reached −7.7°C, a record value for the month of December

(ARSIAL: Agenzia Regionale per lo Sviluppo e l'Innovazione dell'Agricoltura del Lazio, last access: 26/07/2020). Polar temperatures were recorded on 17th December in Forlì (−6°C), in Parma (−7.5°C, Arpae.it (last access: 26/07/2020)), as well as in Ancona -6.8°C (Annali Idrologici, Centro Funzionale Mutlirischi per la Meteorologia, l'Idrologia e la Sismologia Regione Marche 12/2010) and in Firenze with its −7.3°C (Servizio Idrologico Regionale Regione Toscana, last access: 26/07/2020), in Isernia −11.8°C , in Salerno (Tirrenean coast) −7.2°C (Annali Idrologici, Centro Funzionale Mutlirischi per la Meteo-

rologia, l'Idrologia e la Sismologia Regione Capania)(Il Messaggero 17/12/2010, 3bmeteo.com (last access: 26/07/2020) of 19/12/2015, cemcer.it (last access: 26/07/2020) of 17/12/2015, Randi and Ghiselli (2013)).

**02nd February 2012**. The February 2012 cold spell affected a large part of Europe and spread down to North Africa in the period between 27th January and 20th February 2012, causing over 650 deaths in the areas concerned. It was characterized by extremely low temperatures, especially in Eastern Europe, which reached an absolute minimum of −39.2°C in Finland and

heavy snowfall on the remaining European countries (Assessment of the observed extreme conditions during late boreal winter





2011/2012. WMO, 2015). On February 4th, 2012 the snow fell even in Algiers with an accumulation of about 20 cm and the cold air brought snow even in the Sahara Desert (ansamed.info (last access: 26/07/2020) of 08/02/2012). In Italy the cold spell caused serious hardships and at least 57 victims (La Repubblica 12/02/2012). From the end of January, a stream of continental Arctic air reached the peninsula. At first, only the northern regions were affected (e.g. Alessandria reached low temperatures

of −20°C and in Milan the minimum temperature downfall to −14.5°C) but then the cold extended also to the Central and Southern regions (Annali Idrologici, Centro Meteorologico Lombardo 02/2012). The snow fell on most of Italy, especially in Emilia-Romagna and in the provinces of Pesaro-Urbino, Ancona, Macerata and Fermo on the Central regions. Bologna reached −7.6°C on 6th of February, Parma −10.2°C, Rimini-6.2°C, Arpae.it (last access: 26/07/2020)). In many areas of the South Italy like Basilicata and Calabria snow fell. In Palermo (Sicily) snow fell on Monte Pellegrino on the 14th of February (Agenzia

Regionale Protezione Civile Basilicata Annali Idrologici 2012; palermotoday.it (last access: 26/07/2020) of 14/02/2012). The hinterland and the rest of the region were affected by accumulations beyond 20-30 cm. (La Repubblica 12/02/2012), Annali Idrologici, Centro Meteorologico Lombardo 15/02/2012).

**07th February 2013**. This cold spell consisted of a polar trough spreading towards the Mediterranean region: the −12°C 850 hPa isotherm reached Central Europe and the isotherm −6.7°C at the altitude of 850 hPa was measured at midnight on

10th February in the skies of Linate Milan airport (Wetterzentrale.de (last access: 26/07/2020)). The minimum temperatures of 10th February are everywhere very cold in the lower Ticino valley. February 11th was a snowy day on most of the northern Italian plains, with temperatures around zero degrees. In Milan, in about 36 hours of snowfall, more than 20 cm of snow accumulated. It was been an important snow event on the Po Valley. In particular, the largest accumulations were found on the Brianza province, near Lecco, where peaks exceeding 35 cm were recorded Annali Idrologici, Centro Meteorologico Lom-

bardo. Heavy snowfalls were reported in Emilia, in the Lombardy plain, Veneto, lower Trentino and Friuli. Accumulations reached up to 10-15 cm snow-height between Emilia, low Veneto, low Lombardy. Stormy weather also affected the rest of Italy, but snow affected only mountains areas. Only on the central regions snowflakes reached very low altitudes, especially in Tuscany. Annali Idrologici, Centro Meteorologico Lombardo 29/02/2016, Arpae.it (last access: 26/07/2020); Arpav (Arpa Veneto, last access: 26/07/2020), Servizio Idrologico Regionale Regione Toscana, last access: 26/07/2020, meteogiornale.it

(last access: 26/07/2020) of 11/02/2018 milanotoday.it (last access: 26/07/2020) 18/12/2013; Milano.Repubblica (Milano Repubblica 11/02/2013)).

**28th December 2014**. The cold spell affected the South of Italy with locally exceptional snowfalls, especially in Sicily. Even Apulia recorded huge accumulations on the plains and on the coasts. Snow has also appeared in Naples and on the Amalfi Coast. Sicily had the snowstorm on Messina and whitewashed on the coasts, snow on the hills of Palermo and huge accumula-

tions all over the hinterland, in particular over all the low hills of the northern part of Sicily. The snow appeared in Syracuse, the event was modest in Catania, with accumulations only in the reliefs of the city. The extreme south-eastern tip of Sicily, observed snow during the night of the New Year 2015, an extremely rare event for these southernmost areas of the country. These are the less snowy areas of Italy where it snowed last time in January 1905. Historic snowfall were also recorded in Pachino, a city famous for the production of a special type of cherry tomatoes. Intense snowfalls affected the Sicilian towns of the Ionian

side, like Avola and Noto. This cold spell was extraordinary also in the south of Sardinia, for Cagliari and surroundings were





covered by snow (Arpa Sardegna, Analisi Agrometeorologica e Climatologica della Sardegna. 2014-2015). (Annali Idrologici 12/2014, Osservatorio delle Acque, La Repubblica 27/12/2014).

**05th February 2015**. Italy was affected by stormy and snowy conditions. It snowed extensively in Piemonte as well as in Liguria, a region also affected by strong winds. A lot of snow, even at low altitude and in the lowlands in the Northern Italy

and in part of the Centeral Italy. During this event, due to the strong winds, Sicily was isolated and the connections with the smaller islands were interrupted. In the surrounding of Etna a violent blizzard raging of snow, wind gusts and ice suddenly caused freezing conditions. At Enna temperature dropped to $-4.2$°C (Annali Idrologici 02/2015, Osservatorio delle Acque). In Ustica the most difficult situation was observed as ferries could not reach the island for 12 days, and essential medicines were delivered in dangerous operations by helicopter. Many flights were canceled at Sicilian airports. Many roads and high-

ways remained closed as the snow cover in some places exceeded 50 centimeters. In Central Italy on 9th of February ANAS (Azienda nazionale autonoma delle strade) reports that heavy snowfall is causing traffic jams in the provinces of L'Aquila and Teramo. In Northern Italy a snow and electrical blackout happened: numerous municipalities in the Bologna area, and many others in the region, had blackout in light heating and water supply, as well as malfunctions on the telephone network and the Internet. On 7th February a big snowfall involved the city of Parma and the whole Emilia-Romagna region ($-7.4$°C on

9th of February, Arpae.it (last access: 26/07/2020)), with road accidents and problems in the supply of electricity for about 12 thousand customers in the Municipality of Parma. (livesicilia.it (last access: 26/07/2020) of 05/02/2015; today.it (last access: 26/07/2020) 09/02/2015; La Repubblica Bologna 10/02/2015).

**16th January 2016**. On 17th January an icy wind from Siberia, over-passed the barrier of the Alps and reached the Apennine chain. Due to the strong the Eolian islands were isolated and the highest peaks of the islands (Stromboli and Salina) were white-

washed. Storm-surges stroke the Sicilian north coasts (Palermo Osservatorio delle Acque). The temperature suddenly dropped from 18°C to 8°C in 24 hours. In Molise snow covered almost the entire region, even at low altitudes. At Lanciano (Regione Abruzzo Dipartimeto Politiche dello Sviluppo Rurale e dell'Ambiente), the city center was covered by up to 25 cm of snow. 50 cm to 60 cm snow height was also measured in the bordering side of the coasts. The snow fell throughout Basilicata (Agenzia Regionale Protezione Civile Basilicata Annali Idrologici 01/2016) and temperatures reached -8°C. Whitewashed peaks ap-

peared at lower altitudes, as well as the Aspromonte. In Savona were recorded $-11$°C degrees. Accumulations reached up to 20 cm on some area of Cosenza (150 m above sea level) and over 30 cm on the highest hills surrounding the city. The snow brought with it many accidents, above all to the viability with traffic jams on the Salerno-Reggio Calabria highway. Calabria has been the area most affected by the snow wuth few casualties.(meteopalermo.it (last access: 26/07/2020) of 19/01/2016, La Repubblica 17/01/2016 , today.it (last access: 26/07/2020) of 19/01/2016)

**05th January 2017**. From 5th to 21st January 2017, a cold spell affected most of eastern and Central Europe and part of southern Europe, causing the death of at least 60 people. The cold spell and the snowfalls mainly affected Central and Southern Italy. The regions most affected by this cold spell were the Adriatic ones, namely Marche, Abruzzo, Molise, Puglia and Basilicata. Snow reached almost all coastal areas of these regions, with snow cover up to 40 cm. On January 8th, the beach of Porto Cesareo (LE) in Apulia was covered at some points with accumulations of 22–23 cm, resulting as the third most

snowy Italian beach since 2000. The situation was worse in inland areas of the regions, where the snow often exceeded 2



meters height. A strong snowstorm affected the entire Marsicano sector (Abruzzo) with temperatures ranging between −10°C and −13°C, with final accumulation near one meter and temperatures above −20°C below 1300 meters of altitude (Regione Abruzzo, Servizio Presidi Tecnici di Supporto al Settore Agricolo.). On 9th of January the cold air moved to Central Italy (Abruzzo, Marche, Umbria e Molise). The heavy snowfall (with a series of earthquake in Central Italy) caused a disastrous

avalanche that hit the town of Rigopiano in Abruzzo: a landslide swept and destroyed a hotel, causing several deaths. Heavy snowfalls (up to a meter of snow) considerably complicated rescue operations in this region (Corriere della Sera of 19/01/2017). (Aljazeera of 07/01/2017 of 07/01/2017, severe weather.eu (last access: 26/07/2020) 05/01/2017 and of 08/01/2017, La Repubblica 05/01/2017 05/01/2017).

**18th February 2018**. The Siberian cold spell affected Europe between the end of February 2018 and the beginning of March.

The major anomalies concern the Central and Northern sector of Europe with temperatures between 5°C and 9°C below the reference average 1971-2000. Strong anomalies of this brief but intense cold spell were recorded all over Italy. The cold was felt more intensely in the Central and Northern and marginally in the far South Italy and on Sicily. In the Northern regions the values were also 8/9°C below the seasonal averages. Rome experienced a moderate snowfall (3–4 cm of snow) which caused temporary disruptions in ground transportation. The last snowfall in Rome, in chronological order, was February 2012,

when the city was covered with snow for the first time after many years. At Cagliari, the Mistral wind was extremely strong and the gusts reached up to 100 km/h, creating disruptions in the maritime connections between the continent and Sardinia. The wind in Capo Caccia (Alghero) exceeded 70 km/h and 80km/h at Capo Carbonara, on the south coast (Decimomannu, Aeronautica.Militare (last access: 26/07/2020)). The snow-covered whitewashed slopes of the Riviera di Ponente, in Rome, Naples (the last snowing event on 1956), Olbia and Bari Arpa (Puglia). The minimum temperatures of 27th-28th February, were

the lowest in the last 20–30 years above 1500 m at many locations in the Alps. Low temperatures of −25°C were recorded at 2500 m (Protezione Civile Provincia Autonoma di Bolzano). A second pulse of cold air mass reached Italy through the Carso between night and morning on Sunday 25th February, spreading throughout Northern Italy during the daytime along with winds and irregular snowfalls on the plains between Emilia and Piedmont. At low altitude the absolute records remained unbroken, but some localities recorded new temperature records for February as −9.1°C in Bologna, −6.2°C (Arpae.it (last

access: 26/07/2020)) in Rome-Ciampino (129 m, on 2nd March 1963 it reached −6.5°C), −1.1°C in Brindisi (15 m, on 11th March 1956 it reached −4.2°C)(Aeronautica.Militare, last access: 26/07/2020). (Il Foglio 26/02/2017; La Gazzetta di Parma 14/02/2018; La Gazzetta del Serchio 22/02/2018, nimbus.it (last access: 26/07/2020) 02/03/2018).

**Acknowledgements**

This work was supported by ERC grant No. 338965-A2C2 and CEA grant DAMA. We thank Fabio D'Andrea and Aglaé Jézéquel (Laboratoire de Météorologie Dynamique, Paris, France) for useful discussion on the paper. FL acknowledge support by the Deutsche Forschungsgemeinschaft (DFG,German Research Foundation) through the University Hamburg's Cluster of Excellence Integrated Climate System Analysis and Prediction (CliSAP), and under Germany's Excellence Strategy - EXC



2037 'Climate, Climatic Change, and Society' (CliCCS) - Project Number: 390683824, as contribution to the Center for Earth
System Research and Sustainability (CEN) of University Hamburg's. The authors acknowledge the support of the INSU-
CNRS-LEFE-MANU grant (project DINCLIC).

**Author Contribution**

M.D. performed computations, D.F. conceived the idea and the study, D.F. C.N. designed methodology. M.D. made collection
and analysis of the database and performed model simulations in consultation with F.L., M.D. wrote the manuscript and P.Y.,
C.N., F.L. and D.F. discussed the results and implications and commented on and edited the manuscript.

*Competing interests.*  The authors declare no competing interest.



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

Ufficio Idrografico del Po, A. I.: Annali Idrologici Ufficio Idrografico del Po.

Ufficio Idrografico di Roma, A. I.: Ufficio Idrografico di Roma.

Ufficio Idrografico di Venezia, A. I.: Annali Idrologici Ufficio Idrografico Magistrato delle Acque di Venezia.

Ufficio Idrografico e Mareografico di Bari, A. I.: Annali Ufficio Idrografico e Mareografico di Bari.

Ufficio Idrografico e Mareografico di Parma, A. I.: Annali Idrologici Ufficio Idrografico e Mareografico di Parma.

Ufficio Idrografico e Mareografico di Pesacra, A. I.: Annali Idrologici Servizio Idrografico e Mareografico di Pescara.

Ufficio Idrografico e Mareografico di Pisa, A. I.: Annali Ufficio Idrografico e Mareografico di Pisa.

Ufficio Idrografico e Mareografico di Roma, A. I.: Annali Ufficio Idrografico e Mareografico di Roma.

Ufficio Idrografico e Mareografico di Venezia, A. I.: Annali Idrologici Ufficio Idrografico e Mareografico di Venezia.

valdarnopost.it: www.valdarnopost.it, last access: 26/07/2020.

Van Vuuren, D. P., Edmonds, J., Kainuma, M., Riahi, K., Thomson, A., Hibbard, K., Hurtt, G. C., Kram, T., Krey, V., Lamarque, J.-F., et al.: The representative concentration pathways: an overview, Climatic change, 109, 5, 2011.

Vautard, R., Gobiet, A., Sobolowski, S., Kjellström, E., Stegehuis, A., Watkiss, P., Mendlik, T., Landgren, O., Nikulin, G., Teichmann, C., et al.: The European climate under a 2 C global warming, Environmental Research Letters, 9, 034 006, 2014.

Vavrus, S., Walsh, J., Chapman, W., and Portis, D.: The behavior of extreme cold air outbreaks under greenhouse warming, International Journal of Climatology: A Journal of the Royal Meteorological Society, 26, 1133–1147, 2006.

Volosciuk, C., Maraun, D., Semenov, V. A., Tilinina, N., Gulev, S. K., and Latif, M.: Rising mediterranean sea surface temperatures amplify
extreme summer precipitation in central Europe, Scientific reports, 6, 1–7, 2016.

Wetterzentrale.de: www.Wetterzentrale.de, last access: 26/07/2020.

WMO: International Meteorological Vocabulary: Vocabulaire Météorologique International. Vocabulario Meteorológico Internacional, Secretariat of the World Meteorological Organization, 1966.

Wu, Q. and Zhang, X.: Observed forcing-feedback processes between Northern Hemisphere atmospheric circulation and Arctic sea ice
coverage, Journal of Geophysical Research: Atmospheres, 115, 2010.

Yiou, P., Salameh, T., Drobinski, P., Menut, L., Vautard, R., and Vrac, M.: Ensemble reconstruction of the atmospheric column from surface pressure using analogues, Climate dynamics, 41, 1333–1344, 2013.





Zscheischler, J., Martius, O., Westra, S., Bevacqua, E., Raymond, C., Horton, R. M., van den Hurk, B., AghaKouchak, A., Jézéquel, A., Mahecha, M. D., et al.: A typology of compound weather and climate events, Nature reviews earth & environment, pp. 1–15, 2020.

| Analog quantiles | | 2% | 5 % | 10 % |
|---|---|---|---|---|
| CTRL | a) | 0.88 | 0.92 | 0.94 |
| | b) | 0.97 | 0.98 | 0.98 |
| RCP8.5 | a) | 0.91 | 0.94 | 0.95 |
| (RCP8.5/CTRL) | | (1.03) | (1.02) | (1.01) |
| | b) | 0.98 | 0.99 | 0.99 |
| | | (1.01) | (1.01) | (1.01) |
| 4K-SST | a) | 0.73 | 0.80 | 0.84 |
| (4K-SST/CTRL) | | (0.83) | (0.87) | (0.89) |
| | b) | 0.93 | 0.95 | 0.96 |
| | | (0.96) | (0.97) | (0.98) |

**Table 1.** Cold spell frequency per year in the control simulation (CTRL), RCP8.5 and 4K-SST scenarios. The cold spell is defined at different quantiles for analogues of observed cold spells for SLP and anomalies of a) T850 ; b) Z500. The values in parentheses indicate the ratios of frequencies with respect to the CRTL simulations. Values above 1 indicate an increase of frequency.

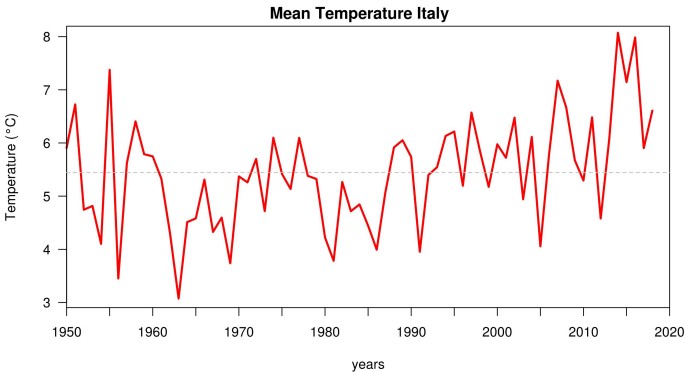

**Figure 1. Mean temperature.** Data of the winter mean temperature (°C) over Italy (35–45N,7.5–18E,) for the period 1950-2018 (E-OBS.v19 Cornes et al. (2018) ).

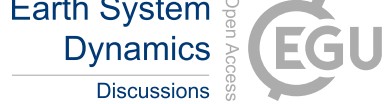

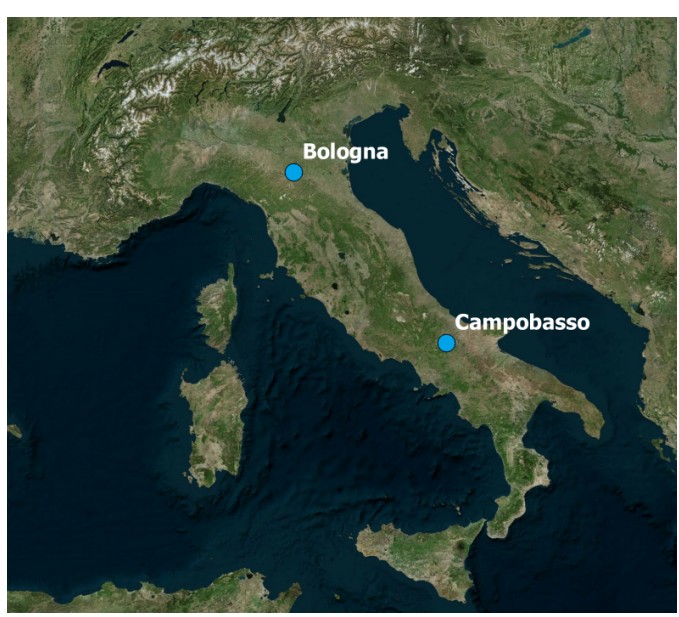

**Figure 2. Map.** Italy with highlighted the cities of Bologna and Campobasso. The image is created using QGis 2.18 software (QGIS Development Team 2020. QGIS Geographic Information System. Open Source Geospatial Foundation Project. http://qgis.osgeo.org) and using image from NASA Worldview: we acknowledge the use of imagery from the NASA Worldview application (https://worldview.earthdata.nasa.gov), part of the NASA Earth Observing System Data and Information System (EOSDIS).





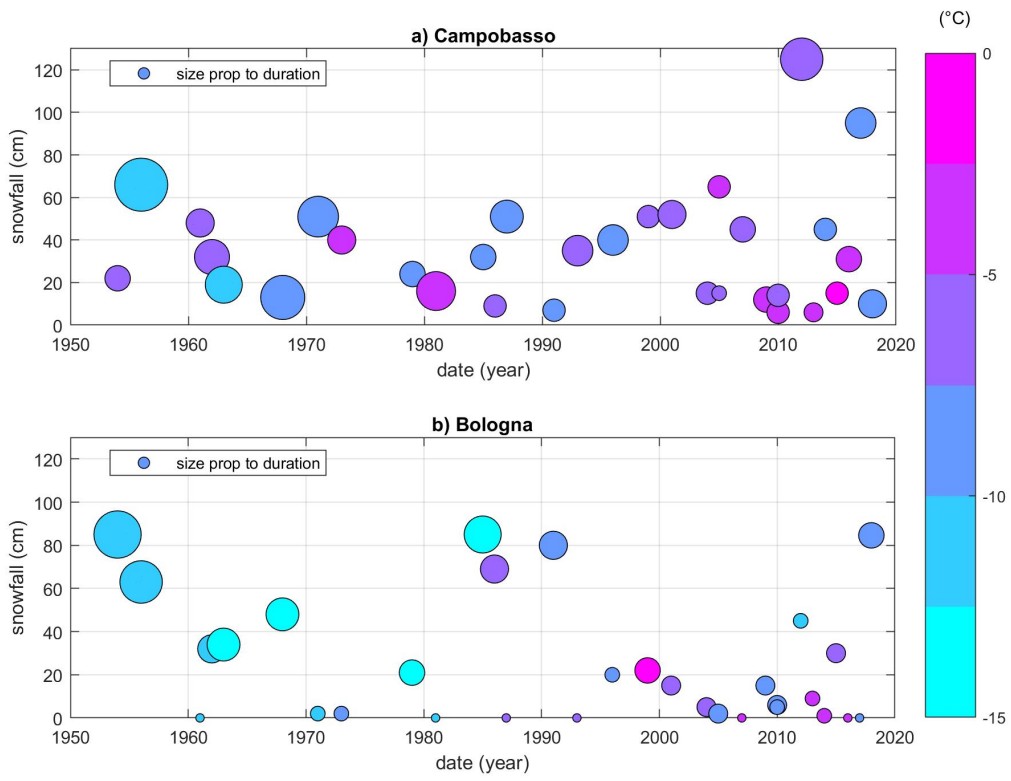

**Figure 3. Cold spells from documentary sources.** Data recorded in a) Campobasso (686 m of altitude); b) Bologna (54 m of altitude). Each ball represents one cold spell event. The size is proportional to the number of snowfall days. The $y$-axis shows the snowfall measured during each event. The color shows the minimum near surface temperature recorded during the event (see Sources and data-set).

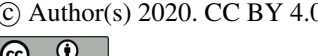



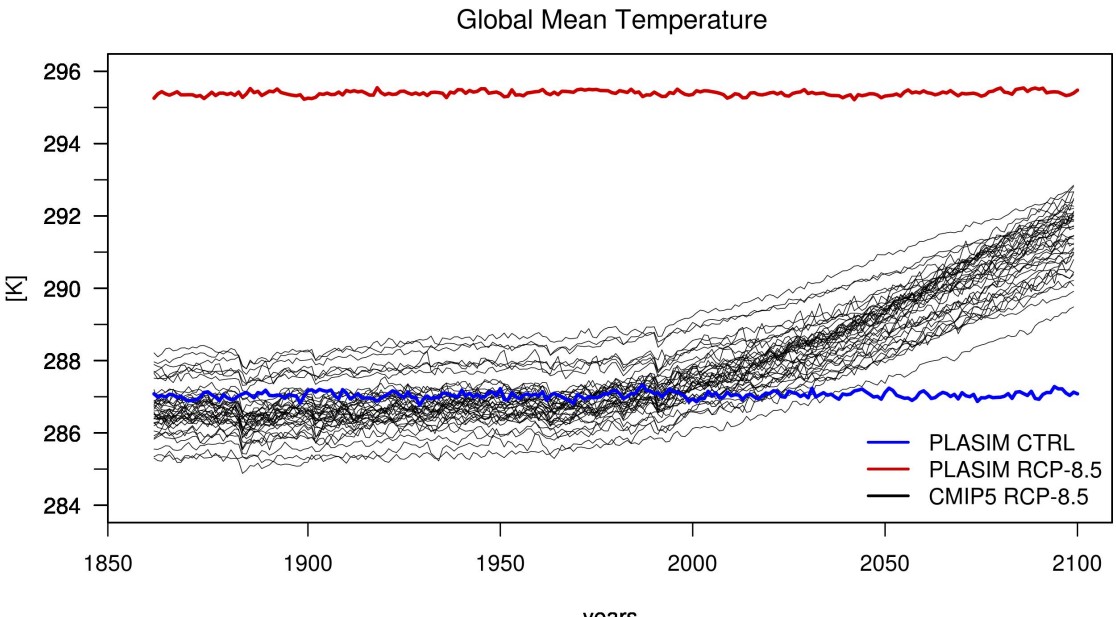

**Figure 4. Global Mean Surface Temperature.** Part (240 years) of the the 500 years long simulation of the global mean surface temperature (K) in PlaSim CTRL run (blue line), PlaSim RCP-8.5 run and CMIP5 RCP-8.5 members 0 to 38 from 1861-2100.



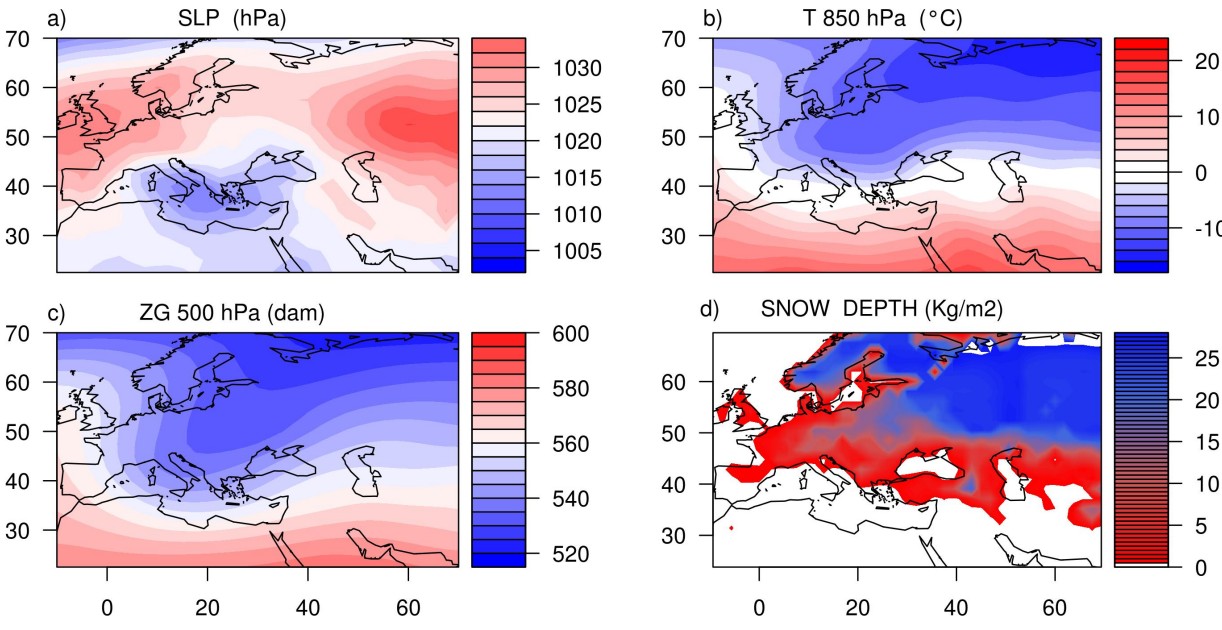

**Figure 5. Averaged cold spells for NCEP reanalysis.** Average of the 32 cold events for a) Sea level pressure (hPa); b) Temperature at 850 hPa (°C); c) Geopotential height at 500 hPa (dam); d) Snow depth (kgm$^{-2}$) for the NCEP reanalysis.

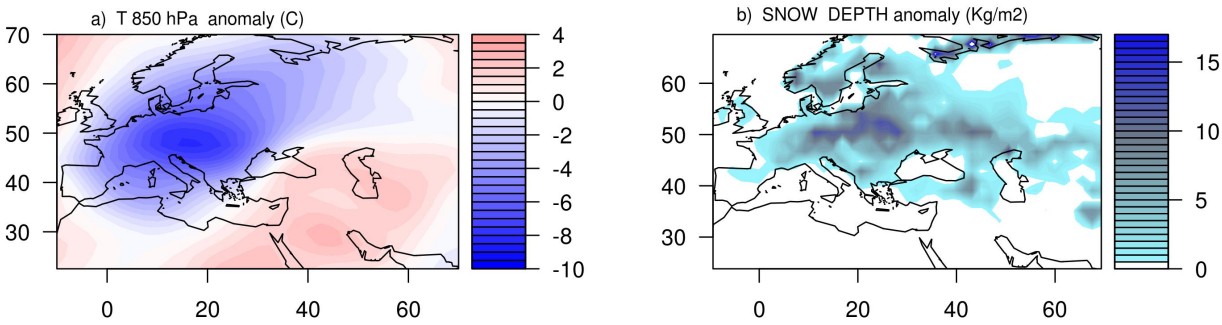

**Figure 6. Cold spell anomaly for NCEP reanalysis.** The anomalies of the 32 cold spells are computed with respect to the winter season over 1948-2018 of a) temperature at 850 hPa (°C) and b) snow depth (Kg/m$^2$) for NCEP reanalysis.

**Figure 7. Averaged cold spells for PlaSim** The average of 32 best analogues of PlaSim for (a,b,c) Sea level pressure (SLP) , for (d,e,f) Temperature at 850 hPa (T850), for (g,h,i) Geopotential height at 500 hPa (Z500) and for (j,k,l) snow depth (Kg/m$^2$). a,d,g,j) CTRL run; b,e,h,k) RCP-8.5 run and c,f,i,l) 4K-SST run.



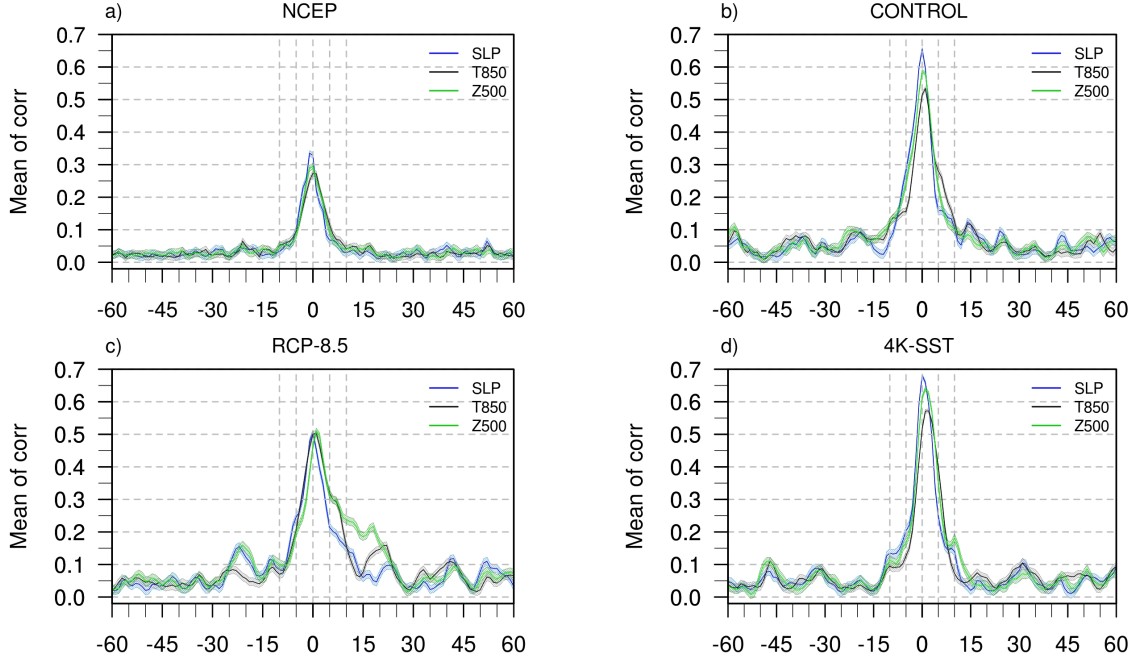

**Figure 8. Mean of correlation matrix between the events.** The pairwise correlations are computed among the anomalies of SLP (blue line), T850 (black line) and Z500 (green line) of the a) 32 cold spells in NCEP reanalyses; of the 32 best analogues in b) control run; c) RCP8.5 run and d) 4K-SST run at different time lags.





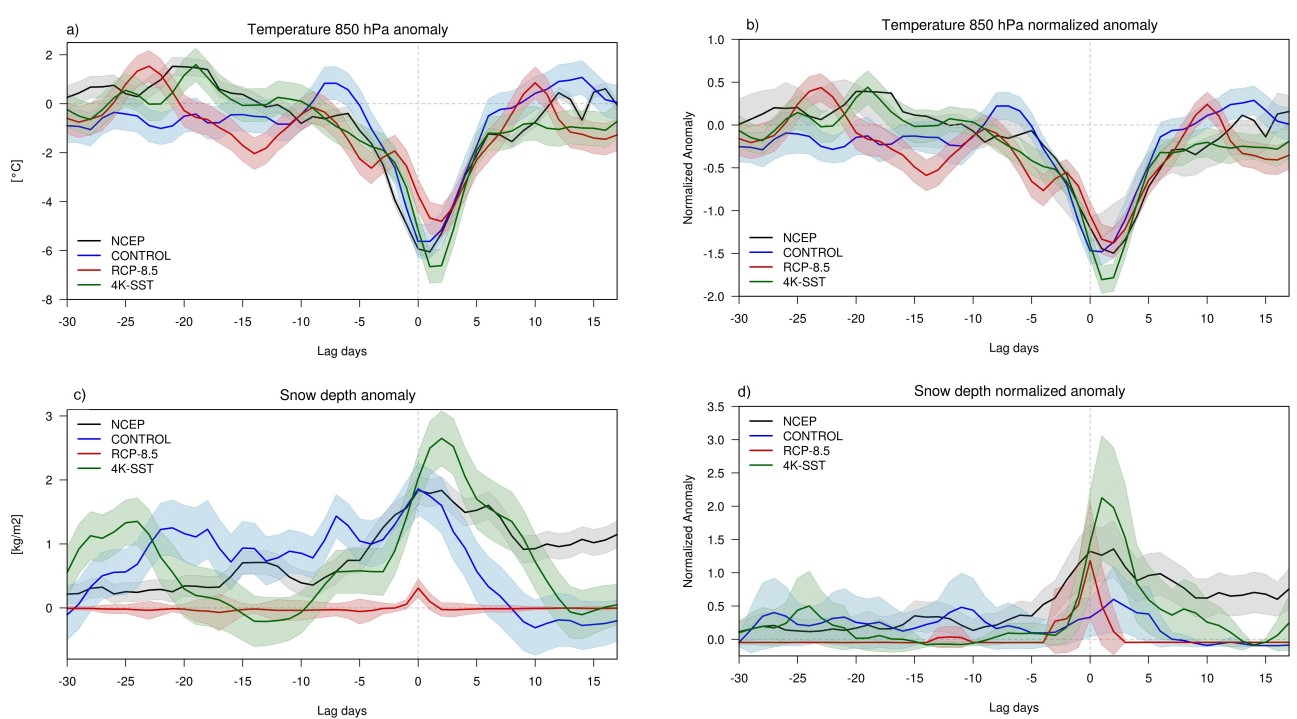

**Figure 9. Cold spell anomalies.** a,b) T850 (°C) and c,d) snow depth ($\mathrm{kgm}^{-2}$) anomalies of the cold spells in NCEP reanalyses (black line) and of best analogues of PlaSim for control (blue line), RCP8.5 ($10^{-2}\mathrm{kgm}^{-2}$) (red line) and 4K-SST (green line) runs at different time lags. Standard deviation represented as shading;in a,c) anomalies are obtained by subtracting the seasonal average, in b,d) normalized anomalies are obtained by subtracting the seasonal average and dividing by the seasonal standard deviation.



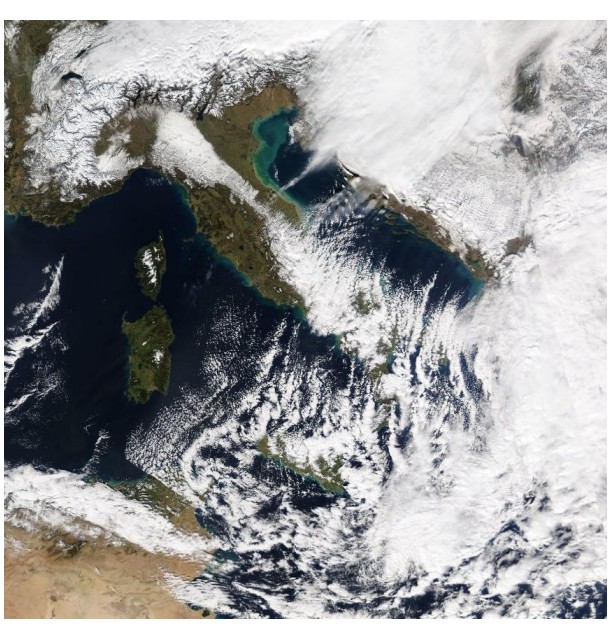

**Figure 10. NASA satellite image.** The image shows the snow convective cloud bands associated to the cold spells (Visible Infrared Imaging Radiometer Suite (VIIRS), Aqua and Terra, images of 09-02-2015. NASA Worldview: https://worldview.earthdata.nasa.gov/?v=-15.737534490207954,27.95902835268575,38.67376580662584,52.84965335268576&t=2015-02-09-T00. We acknowledge the use of imagery from the NASA Worldview application (https://worldview.earthdata.nasa.gov), part of the NASA Earth Observing System Data and Information System (EOSDIS).

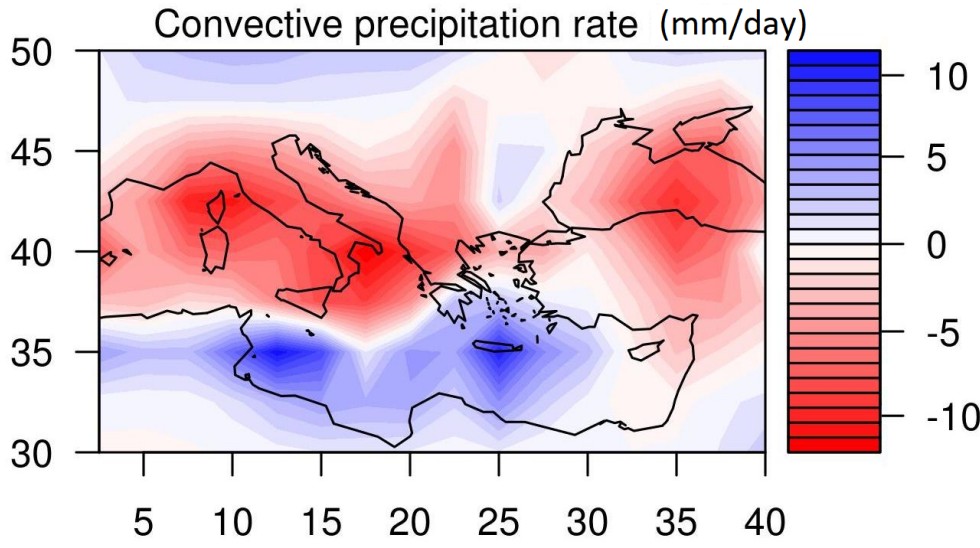

**Figure 11. Convective precipitation rate.** The rate of the convective precipitation (mm/day, $\sim 10^{-8}$m/s) is shown as difference of 4K-SST run and control run to highlight the atmospheric instability over the Mediterranean sea. The 4K-SST run is more unstable, thus explaining the convective feedback triggering heavy snowfalls.