# Peer review of "Present and future synoptic circulation patterns associated with cold and snowy spells over Italy"

_Earth System Dynamics, 2020_

## Referee Comment (RC1) · Anonymous Referee #1 · 10 Oct 2020

**GENERAL COMMENTS**

This paper first uses reanalysis data to document the large-scale circulation conditions that have led to heavy snowfall events together with low temperatures in Italy between the years 1954 and 2018. After this, an intermediate complexity climate model (PlaSim) is used to explore how the occurrence of such events might change in a warmer future climate.

The paper has its positive aspects but also severe limitations. To start with the former, it includes a valuable compilation of 32 major snowfall / cold spell events that have affected Italy since the mid-20th century. The analysis of the corresponding circulation

anomalies in the NCEP reanalysis also makes good sense. Furthermore, the paper is written in good English.

On the other hand, there are many problems in the PlaSim simulations and in the interpretation of their results. The first is, obviously, the coarse  $(2.8^{\circ} \times 2.8^{\circ})$  resolution of the model. Simulations at such a resolution give very little direct information on snowfall in Italy. In the control simulation, the (country mean) snow water equivalent anomalies in the identified extreme cases are of the order of 2 mm, which is at least an order of magnitude smaller than the observed local snowfalls. Therefore, in practice, the analysis mainly gives information on the atmospheric circulation events that resemble the circulation during the observed snowfall extremes.

A second important problem is that the severity of the cold spells is only analysed based on the magnitude of the 850 hPa cold anomalies relative to the climatological mean values. These anomalies are found to (more or less) retain their earlier magnitude, leading to the suggestion that such events in the future will be no less severe than those observed this far. Yet the warmer mean climate in these simulations also means that the actual temperatures during the cold spells will become higher. In the RCP8.5 scenario, this change is large enough to nearly eliminate all snowfall in Italy. Thus, a cold anomaly with the same magnitude will not have the same effects in a warmer climate.

In addition to the simulation based on the RCP8.5 forcing scenario, the study uses another simulation in which the sea surface temperature (SST) has been uniformly increased by 4 K, without changing the atmospheric composition. Such a simulation may be useful for process studies but does not represent a plausible future. Increasing the SST without increasing atmospheric greenhouse gas concentrations creates an artificial energy source at the sea surface, which distorts the dynamics of the climate system. The finding that the simulated snowfall extremes increase under such conditions is therefore difficult to interpret.
Aside from these scientific issues, the selection of figures requires consideration. For example, Figure 8 is hardly at all discussed in the text, suggesting that it is redundant. Figure 4 is also a candidate for deletion (see comment 17 below). On the other hand, to aid the reader to assess how severe the simulated future cold spells are, figures and/or other information on the average winter warming would be needed.

In conclusion, large improvements are still needed in this paper.

**SPECIFIC COMMENTS**

1. L131. What is the length of the cold spells and their analogues? Does it vary from case to case, and if so, how is the length determined?

2. L156-157. How much does the average winter T850 in Europe / in Italy increase relative to CTRL in RCP8.5 and 4SST?

3. L163-168 and Fig. 8. If there is nothing more to say about Fig. 8, the figure and this paragraph can be deleted. To me there are two main messages: (i) the zero-lag correlations between the observed events are not very strong ( $\sim$ 0.3) suggesting that there is actually quite a lot of case-to-case variability, and (ii) the correlations in PlaSim are stronger, indicating that there is less case-to-case variation in the model.

4. L179-181. Although the anomalies remain similar, the absolute temperatures must be higher (how much higher?) in RCP8.5 and 4SST than in CTRL. I don't see anything particularly counterintuitive in your results.

5. L179-181. Are these differences in the average magnitude of the cold anomaly statistically significant in comparison with the inter-event variability?

6. L181. Warmer mean temperatures are expected, but not necessarily smaller warm or cold anomalies (the latter depends on location and season).

7. L193-194. Is this really cooler than in CTRL in terms of the absolute temperature?

8. L196-197. There is nothing about the lapse rate in Eq. (1). Delta\_T is the tempera-

**ESDD**
ture tendency caused by the parameterized condensation of water vapour.

9. L197-199. If the colour scale in Fig. 11 is correct, then there is less convective precipitation over the northern parts of the Mediterranean (including surroundings of Italy) although more in the south.

10. L211-212. Although a similar frequency of the analogue events (in terms of the SLP and T850 anomalies) occurs, the absolute values of T850 during these events are higher. Therefore, the sentence (This mean that ...) is misleading.

11. L215-217. Overinterpretation of a very subtle difference. Even if the difference were larger, why would the decrease in the frequency of good circulation anomalies in 4K-SST indicate that dynamic processes are more favoured than thermodynamic processes?

12. L222-224. The 4K-SST increase without increasing greenhouse gases is not a physically consistent possibility in the real world.

13. Figure 7d. The values of T850 in CTRL (particularly) in Italy seem surprisingly high compared with those in NCEP (Fig. 5b), RCP8.5 and 4SST (Figs. 7e-f). Are they correct? In particular, given the 8 K increase in the global mean temperature in RCP8.5 (Fig. 4), a much larger difference between CTRL and RCP8.5 would be expected.

**COMMENTS ON PRESENTATION**

- 14. L29. Dynamics of compound extreme cold and snowy events?
- 15. L35. the Great Lakes

16. L89. deteriorates the realism of the resulting climate?

17. L117. Figure 4 seems unnecessary. Just mention how much the global mean temperature increases in your RCP8.5 simulation and how much it increases in the CMIP5 simulations by the end of the century.
18. L125. +3.5 K ... under which scenario?

- 19. L139. western half of Eurasia
- 20. L164. at time lags up to  $\pm$  60 days

21. L207. SLP, not SPL

22. L254. main characteristics 23. Table 1. Write T850 and Z850 (rather than a) and b)) directly in the table.

---

## Referee Comment (RC2) · Anonymous Referee #2 · 1 Nov 2020

The paper discusses the topic of heavy and extreme snowfall in Italy in current and future climate. This is a scientifically interesting and societally relevant subject. The starting point of the analysis is a set of 32 extreme historic cases with significant snowfall in at least one of two Italian cities of Bologna and Campobasso. The authors go at length in describing these cases (in the Appendix), which vary from relatively short outbursts, to long-lasting episodes involving cold spells in large parts of Europe. This is followed by an analysis of snowfall under similar circulation types, occurring in 500 year simulations conducted with an intermediate complexity model (PlaSim). It is concluded that extreme snowfall may increase or decrease, depending on whether or not future climate change will express in more than average warming of the Mediterranean.

[Figure]

The paper provides an interesting set of observed cases, along with some interesting analysis of simulations in a coarse resolution intermediate complexity model. However, as the paper is presently formulated, it lacks to provide a convincing story that connects the two. There are moreover serious shortcomings in the current description and presentation of the results, which I will try to motive in more detail below. Based on this, however, I recommend to reject the paper in its current form.

Major remarks

1. Event definition. In the Introduction the authors argue, that while there is general consensus that temperature is increasing and mean snowfall is decreasing, knowledge of the changes of extreme "snowy" cold spells is inconclusive, because of inconsistencies in their respective definitions. From this statement I had anticipated that the paper would start with such a definition. However, it is absent. Instead the authors implicitly "define" the case by means of the observed large-scale circulation that accompanied the (start of the) events. Despite the circulations being "very similar" as the authors write on p5 L140, there is apparently enough variation to allow the huge differences in the observed snowfall amount (Fig3). The correlation figures, though only briefly described, also seem to hint in this direction (rather low correlations).

2. Snowfall/depth in intermediate complexity models. The way in which the study attempts to address its main question, involves the use of an intermediate complexity model. While there is nothing wrong with using such intermediate complexity models, it can be questioned whether they are suitable for the problem at hand. Cold spells, especially when defined with respect to a fixed temperature, and in particularly snowfall, will depend sensitively on a lot of parameters, microphysics, precipitation, the representation of the underlying orography and much more. Since for snowfall to occur, the temperature has to be around freezing point, biases in temperature will all too easily imply biases in snowfall. To the knowledge of this reviewer, intermediate complexity models are relevant to the real world mostly because of their reasonably well resolved "dynamics", not so much because of the details of their resolved thermodynamics /

surface parameters / precipitation, let alone snowfall. As a consequence, I think the results in this paper should be treated with extreme care, and can basically only be interpreted within the limited validity of the intermediate complexity model itself, and not as a direct proxy of what may happen in the real world at a local scale, such as, in this case, in Italy.

3. Reanalysis. The principal source of reanalysis data is well known for its shortcomings, of especially its surface variables. Some reasons are given in https://journals.ametsoc.org/bams/article/77/3/437/55258/The-NCEP-NCAR-40-Year-Reanalysis-Project. As such it is questioned whether the snowfall, t2m temperature and consequently snow depth are variables that can be meaningfully used. Upper-level air temperature, and Z500, as well as possibly mean sea-level pressure can be safely used.

4. Unrealistic SST+4K simulation Three different simulations are carried out with PlaSim. In one of them the global SST is increased uniformly by 4 degrees. By not changing atmospheric forcing, this leads to an unrealistic situation. The situation of lakeside snow effects might be an important aspect of snowfall changes in the future, but it is likely that some sort of compensating effect occurs in reality. As a consequence, the statements in this paper are likely over-confident. Without doubt there is a role for both circulation and thermodynamic processes. It is worthwhile to lookup some recent literature by e.g. O'Gorman on this subject.

5. Statistical significance. The study starts with a description of the 32 cases (or in fact the description is only given in the appendix). Reading through this interesting and expansive list I get the conclusion that there is a substantial difference between the historic cases, both in scale, in duration, in extremity, etc. As exemplified by Fig3 the variance in local snowfall accompanying these events is huge. Despite this variance, the authors state that the underlying T850/SLP or Z500/SLP fields are quite similar. Why then, do the users restrict themselves to use only 32 cases from the simulations? To me this is unclear. It basically means that for every historic event, only the closest

single model event is selected, whereas already from the observations it becomes clear that there is a huge variability within these cases. In other words, there must be many similar circulations where no snowfall occurs. I could imagine that more robust (model) results could be obtained by considering a larger subset of similar circulation types.

6. Given my comments above, it is my feeling that the paper could benefit from a radical change of viewpoint. By letting the simulations of the model of intermediate complexity form the heart of the paper, and providing context from observed cases in an added discussion, the claims could be made more specific to what is achievable with such a model. For example, how do cold spells change in such a model, and can these be used to examine extreme snowfall. Because you run a simple model, you can afford to run as many long simulations as are required to achieve at least significant results with respect to the circulation changes. The thermodynamic changes will be hard given the limitations of the model, but perhaps some knowledge can be squeezed out, if results are considered at larger spatial scales. I do not think PlaSim can be reasonably expected to give realistic results at local scale.

Minor remarks:

Note that I will not comment on all minor textual and graphical aspects, since I believe the paper should first be rewritten. The other reviewer has already commented on some of the figures.

1. On page 4, it is stated that five sigma levels are used. However, on page 3, the model is introduced with ten vertical levels. 2. Figures 1,2,4 can be left out. Graphics of the snowfall panels in Figure 5-7 should be improved. Currently, they make a rather unconvincing case of why you would analyze the snowdepth in central Italy. 3. Figure 8. It is unclear over which domain the correlations are computed. Furthermore, it seems totally irrelevant to consider a lag running up to +/- two (!) months. A point-wise correlation between 0.2 and 0.3 in the observations suggest to me that there are huge differences between the fields. If anything, the larger correlations in the PlaSim

simulations suggest that the simpler model is not at all able to capture the variability as observed. 4. Figure 9. These are already more meaningly lags, but here the significance of the results are questioned. Furthermore, it is not clear whether deviations from REF climatology are used, or from each simulations' own climatology. The mean snow-depth anomalies are also rather small $\sim$ O(cm), suggesting that the events are not as extreme as the text suggests. 5. Fig 11., I don't understand the units of convective precipitation ($\sim 10^-8$ m/s), neither do I understand whether this is a composite over all winter days, or only over the 32 selection. Moreover, the blue area may indicate enhanced precipitation, but over Italy the signal is predominantly red indicating a decrease. 6. At some places, strange formulations are used (e.g., in the Appendix, in one of the cases (p9, L284), it is stated that "The cold primates belong to Sweden and Finland", a sentence that is hard to understand. I would recommend to let an native speaker spell check the entire document upon resubmission.

---

## Author Comment (AC1) · 18 Jan 2021

GENERAL COMMENTS This paper first uses reanalysis data to document the large-scale circulation conditions that have led to heavy snowfall events together with low temperatures in Italy between the years 1954 and 2018. After this, an intermediate complexity climate model (PlaSim) is used to explore how the occurrence of such events might change in a warmer future climate. The paper has its positive aspects but also severe limitations. To start with the former, it includes a valuable compilation of 32 major snowfall / cold spell events that have affected Italy since the mid-20th cen-

tury. The analysis of the corresponding circulation anomalies in the NCEP reanalysis also makes good sense. Furthermore, the paper is written in good English.

**We thank the reviewer for the appreciation of our work. Indeed the reconstruction of the 32 major snowfall events via multiple sources is an achievement of this work on its own as we wanted to investigate the compound aspects of cold and snowy spells over large populated areas of Italy.**

On the other hand, there are many problems in the PlaSim simulations and in the interpretation of their results. The first is, obviously, the coarse ($2.8\mathring{a}\mathring{U}\c{e} \times 2.8\mathring{a}\mathring{U}\c{e}$) resolution of the model. Simulations at such a resolution give very little direct information on snowfall in Italy. In the control simulation, the (country mean) snow water equivalent anomalies in the identified extreme cases are of the order of 2 mm, which is at least an order of magnitude smaller than the observed local snowfalls. Therefore, in practice, the analysis mainly gives information on the atmospheric circulation events that resemble the circulation during the observed snowfall extremes.

**We agree with the comment of the referee, which is also reflected by the other reviewer about the inadequacy of PlaSim in reproducing physical quantities such as snowfall. We will change the point of the view of the manuscript and focus the results obtained with PlaSim on the analogues of the circulation fields associated with the events detected in the reanalysis. Our study therefore provides a procedure to evaluate the role of atmospheric circulation in the occurrence of those compound events. The manuscript will then have the following key points: 1) Detection of compound snowy and cold events over Italy in documentary sources.**
**2) Assessment of the existence of common circulation patterns for such events.**
**3) Detection of Analogues in climate simulations in present and future climate conditions and consideration about the associated temperature, snow depth and pressure fields.**

A second important problem is that the severity of the cold spells is only analysed based on the magnitude of the 850 hPa cold anomalies relative to the climatological mean values. These anomalies are found to (more or less) retain their earlier magnitude, leading to the suggestion that such events in the future will be no less severe than those observed this far. Yet the warmer mean climate in these simulations also means that the actual temperatures during the cold spells will become higher. In the RCP8.5 scenario, this change is large enough to nearly eliminate all snowfall in Italy. Thus, a cold anomaly with the same magnitude will not have the same effects in a warmer climate. In addition to the simulation based on the RCP8.5 forcing scenario, the study uses another simulation in which the sea surface temperature (SST) has been uniformly increased by 4 K, without changing the atmospheric composition. Such a simulation may be useful for process studies but does not represent a plausible future. Increasing the SST without increasing atmospheric greenhouse gas concentrations creates an artificial energy source at the sea surface, which distorts the dynamics of the climate system. The finding that the simulated snowfall extremes increase under such conditions is therefore difficult to interpret.

**Following the suggestion of the reviewer, in the version of the manuscript we will also display the changes in the identified observables for all winter days and for the days corresponding to analogues of NCEP reanalyses. This figure is reported here as Figure A1. It suggests indeed the need to rephrase some statements of the manuscript and to clearly state the thermodynamic limitations of our study, namely that:**
**A warmer mean climate in RCP8.5 simulation implies that temperatures during cold spells will be way higher and that snowfall will possibly disappear.**
**Snowfall in Italy is almost eliminated in the RCP8.5 scenarios.**
**The +4K SST is an idealized simulation, taken from AMIP runs, that we use to push to an extreme set-up in order to observe clear thermodynamic changes in PlaSim. We will stress furthermore that this simulation is just used to understand the possible thermodynamic feedback of warmer Mediterranean sea dur-**

**ing events whose atmospheric circulation matches cold and snowy spells. The comments of both the referees made us rethink to the conclusions that can be drawn from this analysis: the fact that the convection potential is enhanced with warmer seas can i) produce snowfalls in some cases where the temperatures remain below the melting threshold ii) transform the snowy events in events where large amounts of convective precipitation falls on the ground in liquid or mixed phase, with important consequences for hydrology and winter tourism.**

Aside from these scientific issues, the selection of figures requires consideration. For example, Figure 8 is hardly at all discussed in the text, suggesting that it is redundant. Figure 4 is also a candidate for deletion (see comment 17 below). On the other hand, to aid the reader to assess how severe the simulated future cold spells are, figures and/or other information on the average winter warming would be needed.

**Figure 4 will be replaced by that presented at the end of this answer (Figure A1).**

In conclusion, large improvements are still needed in this paper.

SPECIFIC COMMENTS L131. What is the length of the cold spells and their analogues? Does it vary from case to case, and if so, how is the length determined?

**For each cold-spell we have taken the 32 days documented in the appendix as the ones for the analogous search. Then, for each of those days, we have taken the best analogues and performed a lagged analysis in time. So the analysis is based on a single day. The lag extends back in the past and forward in the future from this day. We will further specify this in the new version of the manuscript.**

L156-157. How much does the average winter T850 in Europe / in Italy increase relative to CTRL in RCP8.5 and 4SST?

**To answer this question, we have produced Figure A1 at the end of this review. It will substitute with Figure 4 in the previous version of the manuscript. Indeed the RCP8.5 scenario is about 10 degrees warmer in winter, over Italy, than the**

**CTRL scenario. Interestingly, the 4SST scenario produces about the same temperatures as the CTRL scenario. This explains why in the RCP8.5 scenario we barely observe snowfalls while we do observe it for the 4SST one.**

L163-168 and Fig. 8. If there is nothing more to say about Fig. 8, the figure and this paragraph can be deleted. To me there are two main messages: (i) the zero-lag correlations between the observed events are not very strong (âĹ­ij0.3) suggesting that there is actually quite a lot of case-to-case variability, and (ii) the correlations in PlaSim are stronger, indicating that there is less case-to-case variation in the model.

**We will rephrase the manuscript accordingly.**

L179-181. Although the anomalies remain similar, the absolute temperatures must be higher (how much higher?) in RCP8.5 and 4SST than in CTRL. I don't see anything particularly counterintuitive in your results.

**The new boxplot figure at the end of the end of the review shows that indeed for the RCP8.5 scenario, the temperature is about 10 degrees higher than in CTRL.**

L179-181. Are these differences in the average magnitude of the cold anomaly statistically significant in comparison with the inter-event variability?

**Figure A1 clearly shows that: analogues of cold spells yield significantly lower temperature and geopotential height and higher Snow Depth amounts than the statistics for all winter days.**

L181. Warmer mean temperatures are expected, but not necessarily smaller warm or cold anomalies (the latter depends on location and season).

**We agree with the reviewer. This sentence will be rephrased.**

L193-194. Is this really cooler than in CTRL in terms of the absolute temperature?

**We will correct the sentence as: "This effect is amplified by a warmer ocean in the case of 4K-SST simulation causing cold spells with temperatures and snow-**

**fall comparable to those observed in the CTRL scenario"**

L196-197. There is nothing about the lapse rate in Eq. (1). Delta_T is the temperature tendency caused by the parameterized condensation of water vapour.

**The distribution of temperature $Tcl$ in the cloud is found by first lifting the air dry adiabatically and corrected due to condensation of water vapor. The temperature tendency $(\Delta T)_{cl}$ is the temperature difference between the environmental heating and cloud temperature of each cloud layer $(T_{cl} - T_e)$.**
**Cumulus clouds are assumed to exist only if the environmental air with temperature Te is unstable stratified with regard to the rising cloud parcel: $(T_{cl}) > (T_e)$. The top of the cloud $\sigma_{Top}$ is then defined as $\sigma_{Top} = \sigma_{l+1/2}$ if $(T_{cl})_l < (T_e)_l$ and $(T_{cl})_{l+1} > (T_e)_{l+1}$ . The final temperature $\partial T/\partial t$ which appears in the diabatic leap frog time step is given by $(\Delta T)cl/2\Delta t$, where $2\Delta t$ is the leap frog time step of the model. The convective precipitation rate $P_c[m/s]$ of each cloud layer is**

$$P_c = \frac{c_p \Delta p}{Lg\rho_{\mathsf{H_2O}}} \frac{(\Delta T)^{cl}}{2\Delta t} \tag{1}$$

**where$\Delta p$ is the pressure thickness of the layer and $\rho_{H_2O}$ is the density of water. Note that in the previous expression, the larger the convection, the more negative is the $P_c$ value because of the definition of $(\Delta T)_{cl}$ which is itself negative.**

L197-199. If the colour scale in Fig. 11 is correct, then there is less convective precipitation over the northern parts of the Mediterranean (including surroundings of Italy) although more in the south.

**We will rephrase this sentence to make it clearer to the readers.**

L211-212. Although a similar frequency of the analogue events (in terms of the SLP and T850 anomalies) occurs, the absolute values of T850 during these events are higher. Therefore, the sentence (This means that . . .) is misleading.

**We will rephrase this sentence "This means that the chance of cold event hap-**

pening in this region does not decrease under anthropogenic forcing." as: "This means that, in an RCP8.5 warmer climate, geopotential patterns similar to those leading to historical cold spells will be observed with the same frequency and lead to relatively lower temperature than average winter days. However, the large increase of mean temperature in the RCP8.5 scenarios will prevent snowfall precipitation during most of those events".

L215-217. Overinterpretation of a very subtle difference. Even if the difference were larger, why would the decrease in the frequency of good circulation anomalies in 4K-SST indicate that dynamic processes are more favoured than thermodynamic processes?

**We see the point of the reviewer. We will replace "are more favored" with "concur".**

L222-224. The 4K-SST increase without increasing greenhouse gases is not a physically consistent possibility in the real world.

**As we stressed in a previous answer, we will stress that we have used the 4SST only to study lake effect snow and that it is not physically consistent with the real world.**

Figure 7d. The values of T850 in CTRL (particularly) in Italy seem surprisingly high compared with those in NCEP (Fig. 5b), RCP8.5 and 4SST (Figs. 7e-f). Are they correct? In particular, given the 8 K increase in the global mean temperature in RCP8.5 (Fig. 4), a much larger difference between CTRL and RCP8.5 would be expected.

**Let us say that many climate simulations have an offset with respect observations and Plasim does not make exceptions for that. We will further underline it in the new version of the manuscript. However, we checked that FIgure 7 is correct and coherent with new Figure A1.**

COMMENTS ON PRESENTATION

L29. Dynamics of compound extreme cold and snowy events?

L35. the Great Lakes

L89. deteriorates the realism of the resulting climate?

L117. Figure 4 seems unnecessary. Just mention how much the global mean temperature increases in your RCP8.5 simulation and how much it increases in the CMIP5 simulations by the end of the century. C4

L125. +3.5 K . . . under which scenario?

L139. western half of Eurasia

L164. at time lags up to +/- 60 days

L207. SLP, not SPL 22. L254. main characteristics 23. Table 1. Write T850 and Z850 (rather than a) and b)) directly in the table

**Technical corrections will be implemented in the new version of the paper.**

**Please find figure A1 in attachment:**
**Figure A1: Boxplots of the spatial average over Italy of SLP (hPa) (a), T850 ($^\circ$C) (b), Geopotential Height HGT (dam) (c) and Snow depth (Kg/m2) (d) for all winter days (grey) and for the analogues of cold spells (blue).**
* * *
**Fig. 1.**

---

## Author Comment (AC2) · 18 Jan 2021

The paper discusses the topic of heavy and extreme snowfall in Italy in current and future climate. This is a scientifically interesting and societally relevant subject. The starting point of the analysis is a set of 32 extreme historic cases with significant snow-fall in at least one of two Italian cities of Bologna and Campobasso. The authors go at length in describing these cases (in the Appendix), which vary from relatively short outbursts, to long-lasting episodes involving cold spells in large parts of Europe. This is followed by an analysis of snowfall under similar circulation types, occurring in 500 year

simulations conducted with an intermediate complexity model (PlaSim). It is concluded that extreme snowfall may increase or decrease, depending on whether or not future climate change will express in more than average warming of the Mediterranean. The paper provides an interesting set of observed cases, along with some interesting analysis of simulations in a coarse resolution intermediate complexity model. However, as the paper is presently formulated, it lacks to provide a convincing story that connects the two. There are moreover serious shortcomings in the current description and presentation of the results, which I will try to motive in more detail below. Based on this, however, I recommend to reject the paper in its current form.

We understand the extended criticism expressed by the referee and we are willing to take all necessary steps to provide a coherent presentation of our results. This includes, as also suggested by reviewer #1: i) refocusing the results obtained with PlaSim on the role of atmospheric circulation and on the abundance of patterns simulated with future conditions, ii) better underlying the strong limitations in the representation of thermodynamics of these events in PlaSim. A detailed answer to the specific comments is provided below.

Major remarks 1.

Event definition. In the Introduction the authors argue, that while there is general consensus that temperature is increasing and mean snowfall is decreasing, knowledge of the changes of extreme "snowy" cold spells is inconclusive, because of inconsistencies in their respective definitions. From this statement I had anticipated that the paper would start with such a definition. However, it is absent. Instead the authors implicitly "define" the case by means of the observed large-scale circulation that accompanied the (start of the) events. Despite the circulations being "very similar" as the authors write on p5 L140, there is apparently enough variation to allow the huge differences in the observed snowfall amount (Fig3). The correlation figures, though only briefly described, also seem to hint in this direction (rather low correlations). **ESDD**
In the new version of the manuscript we will provide a clear statement on the definition of the events, namely that we consider documented events which have produced at least a record of minimum temperatures and/or a record snowfall amount (or snow at locations where snowfall has never been observed before) at one or more locations in Italy. Then we have not considered the starting day of the cold/snowy spells, but rather the day where most of the records are documented as the central day of the cold/snow spell for the search of analogues. We will also remove figure 8 and just comment on the value of the autocorrelation function at lag0. As observed by the referee, we will underline that , despite the similarity in the large scale patterns of SLP and Z500, there is a large variability of the position of local pressure minima. Due to the complex geography of Italy, small changes in the position of the cyclonic minima can drive precipitations on the Adriatic, or the Thyrrenian sides of the peninsula thus explaining the different recorded amounts. This also suggests, as stated by the referee, to center our results on the occurrence of large scale circulation patterns associated with these events.

Snowfall/depth in intermediate complexity models. The way in which the study attempts to address its main question, involves the use of an intermediate complexity model. While there is nothing wrong with using such intermediate complexity models, it can be questioned whether they are suitable for the problem at hand. Cold spells, especially when defined with respect to a fixed temperature, and in particularly snowfall, will depend sensitively on a lot of parameters, microphysics, precipitation, the representation of the underlying orography and much more. Since for snowfall to occur, the temperature has to be around freezing point, biases in temperature will all too easily imply biases in snowfall. To the knowledge of this reviewer, intermediate complexity models are relevant to the real world mostly because of their reasonably well resolved "dynamics", not so much because of the details of their resolved thermodynamics / surface parameters / precipitation, let alone snowfall. As a consequence, I think the results in this paper should be treated with extreme care, and can basically only be
interpreted within the limited validity of the intermediate complexity model itself, and not as a direct proxy of what may happen in the real world at a local scale, such as, in this case, in Italy.

We will completely rephrase the manuscript and present our approach as a possible way to detect compound extreme events and then analyse the role of atmospheric circulation in their occurrence via a simple model capable of producing several hundred thousands of large scale circulation patterns in relatively short time. We will take particular care in clearly stating the thermodynamic limitations of our study.

Reanalysis. The principal source of reanalysis data is well known for its shortcomings, of especially its surface variables. Some reasons are given in https://journals.ametsoc.org/bams/article/77/3/437/55258/The-NCEP-NCAR40-Year-Reanalysis-Project. As such it is questioned whether the snowfall, t2m temperature and consequently snow depth are variables that can be meaningfully used. Upper-level air temperature, and Z500, as well as possibly mean sea-level pressure can be safely used.

In the new version of the manuscript, as suggested by the reviewer, the focus will mostly be on the large scale circulation patterns associated with the identified extreme events. We will therefore carefully report results on the physical variables (especially snowfall) associated with the events detected.

Unrealistic SST+4K simulation Three different simulations are carried out with PlaSim. In one of them the global SST is increased uniformly by 4 degrees. By not changing atmospheric forcing, this leads to an unrealistic situation. The situation of lakeside snow effects might be an important aspect of snowfall changes in the future, but it is likely that some sort of compensating effect occurs in reality. As a consequence, the statements in this paper are likely over-confident. Without doubt there is a role for both circulation and thermodynamic processes. It is worthwhile to lookup some recent
**literature by e.g. O'Gorman on this subject.**

The +4K SST is an idealized simulation, taken from AMIP runs, that we use to push to an extreme set-up in order to observe clear thermodynamic changes in PlaSim, mostly to study the possible lake effect snow in the Mediterranean. We will stress furthermore that this simulation is just used to understand the possible thermodynamic feedback of warmer Mediterranean sea during events whose atmospheric circulation matches cold and snowy spells. The comments of both the referees made us rethink to the conclusions that can be drawn from this analysis: the fact that the convection potential is enhanced with warmer seas can i) produce snowfalls in some cases where the temperatures remain below the melting threshold ii) transform the snowy events in events where large amounts of convective precipitation falls on the ground in liquid or mixed phase, with important consequences for hydrology and winter tourism. iii) Clearly states that snowfall can disappear in cold spells in RCP8.5 scenarios.

Statistical significance. The study starts with a description of the 32 cases (or in fact the description is only given in the appendix). Reading through this interesting and expansive list I get the conclusion that there is a substantial difference between the historic cases, both in scale, in duration, in extremity, etc. As exemplified by Fig3 the variance in local snowfall accompanying these events is huge. Despite this variance, the authors state that the underlying T850/SLP or Z500/SLP fields are quite similar. Why then, do the users restrict themselves to use only 32 cases from the simulations? To me this is unclear. It basically means that for every historic event, only the closest C3 single model event is selected, whereas already from the observations it becomes clear that there is a huge variability within these cases. In other words, there must be many similar circulations where no snowfall occurs. I could imagine that more robust (model) results could be obtained by considering a larger subset of similar circulation types.

**We definitely agree with the reviewer that there are other possible ways to define**

ESDD
the analogues. For example, one could take the N closest fields for each event having 32xN events in the database. However, our choice is motivated by the will to have, in the simulations, exactly the same statistical sample as in the reanalysis and being able to directly compare the statistics obtained in the different datasets.

Given my comments above, it is my feeling that the paper could benefit from a radical change of viewpoint. By letting the simulations of the model of intermediate complexity form the heart of the paper, and providing context from observed cases in an added discussion, the claims could be made more specific to what is achievable with such a model. For example, how do cold spells change in such a model, and can these be used to examine extreme snowfall. Because you run a simple model, you can afford to run as many long simulations as are required to achieve at least significant results with respect to the circulation changes. The thermodynamic changes will be hard given the limitations of the model, but perhaps some knowledge can be squeezed out, if results are considered at larger spatial scales. I do not think PlaSim can be reasonably expected to give realistic results at local scale.

Following the suggestion of the reviewers, we will completely reorganize the manuscript as suggested. First we will give more space to the reconstruction of the 32 events, which was one of the major challenges of this study. Then, we will specify the use of intermediate complexity climate simulations as a tool to study the atmospheric circulation associated with the detected extreme events, through the use of analogues. Finally, the +4K simulation will be exploited only as an additional tool to understand the possible role of a warmer mediterranean sea in determining a thermodynamic feedback to the events. We will underline the limitations and suggest our approach (documentary sources + analogues search) as a way to investigate compound extremes.

Minor remarks:
Note that I will not comment on all minor textual and graphical aspects, since I believe the paper should first be rewritten. The other reviewer has already commented on some of the figures.

1. On page 4, it is stated that five sigma levels are used. However, on page 3, the model is introduced with ten vertical levels.

We thank the reviewer and the sentence "the vertical, five non-equally spaced sigma (pressure divided by surface pressure) levels are used" will be deleted from the manuscript.

2. Figures 1,2,4 can be left out. Graphics of the snowfall panels in Figure 5-7 should be improved. Currently, they make a rather unconvincing case of why you would analyze the snowdepth in central Italy.

We will remove Figure 2 and 1. Figure 4 will be replaced by the boxplots shown in Figure A1 at the end of this review where boxplots of the spatial average over Italy of SLP (a), T850hPa (b), Geopotential Height HGT (c) and Snow depth (d) for all winter days (grey) and for the analogues of cold spells (blue), are presented. The boxplots make a convincing case that the analysis over Italy makes sense.

3. Figure 8. It is unclear over which domain the correlations are computed. Furthermore, it seems totally irrelevant to consider a lag running up to +/- two (!) months. A pointwise correlation between 0.2 and 0.3 in the observations suggest to me that there are huge differences between the fields. If anything, the larger correlations in the PlaSim C4 simulations suggest that the simpler model is not at all able to capture the variability as observed. \*

**We remove Figure 8 as suggested by the reviewer. We will rephrase the sentence to account for the large variability.**

4. Figure 9. These are already more meaningly lags, but here the significance of the results are questioned. Furthermore, it is not clear whether deviations from REF
climatology are used, or from each simulations' own climatology. The mean snowdepth anomalies are also rather small  $\hat{a}$ Lij O(cm), suggesting that the events are not as extreme as the text suggests.

To answer this question, we have produced Figure A1 at the end of this review. IT will substitute Figure 4 in the previous version of the manuscript. Indeed the RCP8.5 scenario is about 10 degrees warmer in winter, over Italy, than the CTRL scenario. Interestingly, the 4SST scenario produces about the same temperatures as the CTRL scenario. This explains why in the RCP8.5 scenario we barely observe snowfalls while we do observe it for the 4SST one.

5. Fig 11., I don't understand the units of convective precipitation (âLij10ËĘ-8 m/s), neither do I understand whether this is a composite over all winter days, or only over the 32 selection. Moreover, the blue area may indicate enhanced precipitation, but over Italy the signal is predominantly red indicating a decrease.

We agree with the reviewer that the convective precipitation rate  $P_c$  as expressed in Plasim has a non-common expression. Indeed,  $P_c$  is computed using the following step: first, the distribution of temperature  $T_{cl}$  in the cloud is found by lifting the air dry adiabatically and corrected due to condensation of water vapor. Then, the temperature tendency  $(\Delta T)_{cl}$  is the temperature difference between the environmental heating and cloud temperature of each cloud layer  $(T_{cl} - T_e)$ . Cumulus clouds are assumed to exist only if the environmental air with temperature Te is unstable stratified with regard to the rising cloud parcel:  $T_{cl} > (T_e)$ . The top of the cloud  $\sigma_{Top}$  is then defined as  $\sigma_{Top} = \sigma_{l+1/2}$  if  $(T_{cl})_l < (T_e)_l$  and  $(T_{cl})_{l+1} > (T_e)_{l+1}$ . Then, the final temperature  $\partial T/\partial t$  which appears in the diabatic leap frog time step is given by  $(\Delta T)_{cl}/2\Delta t$ , where  $2\Delta t$  is the leap frog time step of the model. The convective precipitation rate  $P_c[m/s]$  of each cloud layer **ESDD**
is therefore given by the expression

$$P_c = \frac{c_p \Delta p}{Lg \rho_{\mathsf{H}_2\mathsf{O}}} \frac{(\Delta T)^{cl}}{2\Delta t} \tag{1}$$

where  $\Delta p$  is the pressure thickness of the layer and  $\rho_{H_2O}$  is the density of water. Note that, in the previous expression, the larger the convection, the more negative is the  $P_c$  value because of the definition of  $(\Delta T)_{cl}$  which is itself negative. Therefore, the interpretation given in the manuscript is correct : large negative values of  $P_c$  correspond to heavier convection. In the new version of the manuscript this will be specified together with the computation of the composite that is over the 32 events. We will include all the terms needed for the computation of  $P_c$  so that the readers will find it easier to understand figure 11.

6. At some places, strange formulations are used (e.g., in the Appendix, in one of the cases (p9, L284), it is stated that "The cold primates belong to Sweden and Finland", a sentence that is hard to understand. I would recommend to let an native speaker spell check the entire document upon resubmission

We will take care of checking the document for wrong expressions.

Please find figure A1 in attachment: Figure A1: Boxplots of the spatial average over Italy of SLP (a), T850hPa (b), Geopotential Height HGT (c) and Snow depth (d) for all winter days (grey) and for the analogues of cold spells (blue).

**ESDD**
**ESDD**

---

## Author Response (AR1)

Dear Jakob,

We are happy to resubmit a revised version of the manuscript "A dynamical and thermodynamic mechanism to explain heavy snowfalls in current and future climate over Italy during cold spells" now entitled "Present and future synoptic circulation patterns associated with cold and snowy spells over Italy".  We understand that the reviewers and the editorial board have raised several issues with our manuscript and we have undertaken the suggested changes. This resulted in additional work which Miriam D'Errico could not completely afford due to her new job in education and few personal unfortunate events. We have therefore asked Flavio Pons, who is currently working in closely related topics on cold spells over France, to take the lead of the analyses and produce a new version coherent with the reviewers and editorial comments. All co-authors agree that Flavio and Miriam will share the first author position of the paper.

About the paper, we would like to stress that:
1) It is now oriented to describing the dynamics of cold spells events, namely the associated circulation patterns and their changes. As demanded by the reviewers, we leave aside the thermodynamic changes that cannot be proficiently investigated using PlaSim.
2) We defend the choice of using PLASIM for circulation studies with the possibility of performing very long stationary climate  simulations and to obtain good analogues of atmospheric circulations associated with cold spells. To this purpose, we also perform a bias correction which ensures that the control run has analogues statistics of temperature patterns compatible with those extracted from NCEP.
3) Since the reviewers questioned our claim that "the detected [cold-spell] events have a common dynamical signature", we have performed a cluster analysis of circulation patterns. This analysis clearly shows that those patterns can be divided into two clusters, of which we provide extensive analyses in the new version of the manuscript.

We hope that the proposed changes constitute a sufficient improvement to meet ESD publication standards. We would be very thankful if the previous reviewers, which provided useful insights and comments on our work, would be willing to review the current version of the manuscript.  Below, we provide an updated answer to the major comments of the reviewers, ensuring that we have taken into account the minor remarks while writing the new version of the manuscript.

Best Wishes,
Davide (on behalf of all the authors)

Reviewer 1

GENERAL COMMENTS
This paper first uses reanalysis data to document the large-scale circulation conditions that have led to heavy snowfall events together with low temperatures in Italy between the years 1954 and 2018. After this, an intermediate complexity climate model (PlaSim)is used to explore how the occurrence of such events might change in a warmer future climate.The paper has its positive aspects but also severe limitations. To start with the former,it includes a valuable compilation of 32 major snowfall / cold spell events that have affected Italy since the mid-20th century. The analysis of the corresponding circulation anomalies in the NCEP reanalysis also makes good sense. Furthermore, the paper is written in good English.

On the other hand, there are many problems in the PlaSim simulations and in the interpretation of their results. The first is, obviously, the coarse ($2.8° \times 2.8°$) resolution of the model. Simulations at such a resolution give very little direct information on snowfall in Italy. In the control simulation, the (country mean) snow water equivalent anomalies in the identified extreme cases are of the order of 2 mm, which is at least an order of magnitude smaller than the observed local snowfalls. Therefore, in practice, the analysis mainly gives information on the atmospheric circulation events that resemble the circulation during the observed snowfall extremes.

**We thank the reviewer for this comment. We acknowledge that, in the previous version of the manuscript, we have used PlaSim beyond its capabilities to study small scales thermodynamic processes.  Indeed, we largely reformulated the paper giving much more focus to the large scale circulation than to the phenomena at the ground.**

A second important problem is that the severity of the cold spells is only analysed based on the magnitude of the 850 hPa cold anomalies relative to the climatological mean values. These anomalies are found to (more or less) retain their earlier magni-tude, leading to the suggestion that such events in the future will be no less severe than those observed this far. Yet the warmer mean climate in these simulations also means that the actual temperatures during the cold spells will become higher. In the RCP8.5scenario, this change is large enough to nearly eliminate all snowfall in Italy. Thus, a cold anomaly with the same magnitude will not have the same effects in a warmer climate.

**In the new version of the paper, we rely on absolute temperature rather than anomalies. We have also performed bias corrections against reanalysis data to ensure coherence between PlaSim and NCEP.**

In addition to the simulation based on the RCP8.5 forcing scenario, the study uses another simulation in which the sea surface temperature (SST) has been uniformly increased by 4 K, without changing the atmospheric composition. Such a simulation may be useful for process studies but does not represent a plausible future. Increasing the SST without increasing atmospheric greenhouse gas concentrations creates an artificial

energy source at the sea surface, which distorts the dynamics of the climate system. The finding that the simulated snowfall extremes increase under such conditions is therefore difficult to interpret.

**We do agree with the reviewer that the +4K SST is an unphysical scenario. This is now clearly stressed in the new version of the manuscript. When using this scenario our objective is to analyse the retroaction of high SST values on the atmospheric circulation. Indeed in the area of study (Mediterranean Basin), the SST are expected to increase faster than in the surrounding oceans. We believe that this scenario is still useful because we show that warmer SST do not produce any major change in large scale dynamics, with results comparable with the control run.**

Aside from these scientific issues, the selection of figures requires consideration. For Example, Figure 8 is hardly at all discussed in the text, suggesting that it is redundant. Figure 4 is also a candidate for deletion (see comment 17 below). On the other hand,to aid the reader to assess how severe the simulated future cold spells are, figures and/or other information on the average winter warming would be needed.In conclusion, large improvements are still needed in this paper

**The structure of the paper has been changed radically and only Fig. 1 was maintained from the previous version. We hope that the reviewer will appreciate the new insights of the new version of the manuscript**

Reviewer 2

The paper discusses the topic of heavy and extreme snowfall in Italy in current and future climate. This is a scientifically interesting and societally relevant subject. The Starting point of the analysis is a set of 32 extreme historic cases with significant snow-falls in at least one of two Italian cities of Bologna and Campobasso. The author's goat length in describing these cases (in the Appendix), which vary from relatively short outbursts, to long-lasting episodes involving cold spells in large parts of Europe. This is followed by an analysis of snowfall under similar circulation types, occurring in 500 year simulations conducted with an intermediate complexity model (PlaSim). It is concluded that extreme snowfall may increase or decrease, depending on whether or not future climate change will express more than average warming of the Mediterranean.The paper provides an interesting set of observed cases, along with some interesting analysis of simulations in a coarse resolution intermediate complexity model. How-ever, as the paper is presently formulated, it lacks to provide a convincing story that connects the two. There are moreover serious shortcomings in the current description and presentation of the results, which I will try to motive in more detail below. Based on this, however, I recommend to reject the paper in its current form.

Major remarks

1. Event definition. In the Introduction the authors argue, that while there is general consensus that temperature is increasing and mean snowfall is decreasing, knowledge of the changes of extreme "snowy" cold spells is inconclusive, because of inconsisten-cies in their respective definitions. From this statement I had anticipated that the paper would start with such a definition. However, it is absent. Instead the authors implicitly"define" the case by means of the observed large-scale circulation that accompanied the (start of the) events. Despite the circulations being "very similar" as the authors write on p5 L140, there is apparently enough variation to allow the huge differences in the observed snowfall amount (Fig3). The correlation figures, though only briefly described, also seem to hint in this direction (rather low correlations).

**In the new version of the paper, we classify the 32 events based on the baric configuration, showing that they can be divided into two clusters, where the synoptic situation corresponds to an omega blocking with high pressure on the UK, and to a Scandinavian positioning of the high pressure center, with Easterly flow towards the Mediterranean. Then, the search for similar events in the climate simulations is carried out using analogues based on the dynamic configuration of each event. The high phenomenological variability observed in the events remains; however, the 32 events were indeed chosen because they caused significant impacts at sub-country level in Italy. Using dynamic analogues, we search for simulated events that have the same potential to produce similar situations due to the synoptic configuration of the pressure field.**

2. Snowfall/depth in intermediate complexity models. The way in which the study at-tempts to address its main question, involves the use of an intermediate complexity model. While there is nothing wrong with using such intermediate complexity models,it can be questioned whether they are suitable for the problem at hand. Cold spells, es-pecially when defined with respect to a fixed temperature, and in particular snowfall,will depend sensitively on a lot of parameters, microphysics, precipitation, the repre-sentation of the underlying orography and much more. Since for snowfall to occur, the temperature has to be around freezing point, biases in temperature will all too easily imply biases in snowfall. To the knowledge of this reviewer, intermediate complexity models are relevant to the real world mostly because of their reasonably well resolved"dynamics", not so much because of the details of their resolved thermodynamics /C2 Surface parameters / precipitation, let alone snowfall. As a consequence, I think the results in this paper should be treated with extreme care, and can basically only be interpreted within the limited validity of the intermediate complexity model itself, and not as a direct proxy of what may happen in the real world at a local scale, such as, in this case, in Italy.

**We have completely rephrased the manuscript and presented our approach as a possible way to detect compound extreme events from documentary sources and then analyse the role of atmospheric circulation in their occurrence via a simple model capable of producing several hundred thousands of large scale circulation**

**patterns in relatively short time. We will take particular care in clearly stating the thermodynamic limitations of our study.**

3. Reanalysis.The principal source of reanalysis data is well known for its shortcomings, of especially its surface variables.Some reasons aregiven in https://journals.ametsoc.org/bams/article/77/3/437/55258/The-NCEP-NCAR-40-Year-Reanalysis-Project. As such it is questioned whether the snowfall, t2m tem-perature and consequently snow depth are variables that can be meaningfully used.Upper-level air temperature, and Z500, as well as possibly mean sea-level pressure can be safely used.

**In the new formulation of the paper, we base our analysis on sea level pressure and 850 hPa temperature for the characterization of the events and the definition of the analogues, and we leave the analysis of snow depth more as an additional comment, without it being a crucial parameter in the definition of PlaSim coldspell events.**

4. Unrealistic SST+4K simulation Three different simulations are carried out with PlaSm. In one of them the global SST is increased uniformly by 4 degrees. By not changing atmospheric forcing, this leads to an unrealistic situation. The situation of lakeside snow effects might be an important aspect of snowfall changes in the future,but it is likely that some sort of compensating effect occurs in reality. As a consequence, the statements in this paper are likely over-confident. Without doubt there is arole for both circulation and thermodynamic processes. It is worthwhile to look up some recent literature by e.g. O'Gorman on this subject.

**We do agree with the reviewer that the +4K SST is an unphysical scenario. This is now clearly stressed in the new version of the manuscript. When using this scenario our objective is to analyse the retroaction of high SST values on the atmospheric circulation. Indeed in the area of study (Mediterranean Basin), the SST are expected to increase faster than in the surrounding oceans. We believe that this scenario is still useful because we show that warmer SST do not produce any major change in large scale dynamics, with results comparable with the control run.**

5. Statistical significance. The study starts with a description of the 32 cases (or in fact the description is only given in the appendix). Reading through this interesting and expansive list I get the conclusion that there is a substantial difference between the historic cases, both in scale, in duration, in extremity, etc. As exemplified by Fig3 the variance in local snowfall accompanying these events is huge. Despite this variance,the authors state that the underlying T850/SLP or Z500/SLP fields are quite similar.Why then, do the users restrict themselves to use only 32 cases from the simulations?To me this is unclear. It basically means that for every historic event, only the closestC3single model event is selected, whereas already from the observations it becomes clear that there is a huge variability within these cases. In other words, there must be many similar

circulations where no snowfall occurs. I could imagine that more robust (model)results could be obtained by considering a larger subset of similar circulation types.

**In the new version of the paper we have radically changed the design of the numerical experiment. Now we look for analogues being the closest x% of the PlaSim simulated field to the event of interest, leading to more robust samples.**

6. Given my comments above, it is my feeling that the paper could benefit from a radical change of viewpoint. By letting the simulations of the model of intermediate complexity from the heart of the paper, and providing context from observed cases in an added discussion, the claims could be made more specific to what is achievable with such a model. For example, how do cold spells change in such a model, and can these be used to examine extreme snowfall. Because you run a simple model, you can afford to run as many long simulations as are required to achieve at least significant results with respect to the circulation changes. The thermodynamic changes will be hard given the limitations of the model, but perhaps some knowledge can be squeezed out, if results are considered at larger spatial scales. I do not think PlaSim can be reasonably expected to give realistic results at local scale.

**Following the suggestion of the reviewers, we have completely reorganized the manuscript. First, we have given more space to the reconstruction of the 32 events, which was one of the major challenges of this study. Then, we have specified the use of intermediate complexity climate simulations as a tool to study the atmospheric circulation associated with the detected extreme events, through the use of analogues. Finally, the +4K simulation has been exploited only as an additional tool to understand the possible role of a warmer mediterranean sea in determining a thermodynamic feedback to the atmospheric circulation associated with these events. We have underlined the limitations and suggest our approach (documentary sources + analogues search) as a way to investigate compound extremes.**

---

## Referee Report (RR1)

**Review of "Present and future synoptic circulation patterns associated with cold and snowy spells over Italy" by M. D' Errico et al.**

**GENERAL COMMENTS**

The manuscript has been extensively revised from its first version. The catalogue of combined heavy snowfall and low temperature events in Italy between the years 1954 and 2018 is retained, but large changes have been made to the rest of the paper. First, a cluster analysis has been used to divide the observed cold spells to two clusters based on the associated sea level pressure (SLP) fields. Second, the analysis of the intermediate complexity PlaSim climate model simulations has been refocused on the search of situations that are analogous to the 32 identified cold spells in terms of the SLP and the 850 hPa temperature (T850) distributions.

The refocusing of the PlaSim analysis from snowfall to the atmospheric dynamical features has improved the manuscript, since the model is clearly more suitable for the simulation of the latter than the former. Furthermore, the revised manuscript is honest in acknowledging that cold spells in warmer future climate will produce less snow.

However, I still have some questions and concerns related to the interpretation of the main results. The analysis of the PlaSim simulations suggests that the frequency of SLP circulation states that resemble those in the observed cold spells is increasing strongly in the RCP8.5 scenario, whereas the corresponding change resulting for an artificial 4 K warming of the sea surface temperatures is negligible. To me this result is counterintuitive, and its physical significance is difficult to assess without further information on what the increase in RCP8.5 results from. Specifically, the increase in the frequency of circulation analogies could be associated with

1. A change in the winter mean SLP field that makes the average SLP distribution more similar to that observed during the cold spells, or
2. A change in the variability of SLP around its mean state, resulting in a larger frequency of SLP anomaly fields that resemble the SLP anomaly fields during the cold spells.

These two possibilities could be distinguished by repeating the cluster analysis for SLP anomalies relative to the observed or simulated (present-day and RCP8.5 separately) DJFM mean SLP field. If the increase in frequency is still seen when the mean state change has been eliminated by focusing on the anomalies, if must originate from changes in the simulated variability. If it disappears, then the change in the mean SLP field is the key.

Furthermore, if the change in the mean SLP field turns out to explain the increase in the frequency of the circulation analogies, a follow-up question is how and where the

mean state changes. In which parts of the (rather large) analysis domain (22.5-70°N, 80°W-50°E) does the new mean state approach the SLP fields during the observed cold spells?

Finally, biases in the simulated present-day winter mean SLP field might affect the change in the frequency of the circulation anomalies in a non-intuitive way. To check for this possibility, it would be prudent to repeat the analysis after also applying the simple linear scaling bias correction to SLP, not only to T850 as was apparently done.

I also have some concern about the ability of the clustering algorithm to identify good circulation anomalies in the model simulations. There appear to be quite large differences between the observed (Fig. 3) and the simulated (Figs. 8-9) SLP fields for both two clusters of the cold spell cases. Furthermore, the difference between the two clusters appears much smaller for the simulations. I wonder if this might be improved by selecting a somewhat smaller domain in the search of the circulation analogies.

Aside of this issue, the figures need improvement. All the maps (Figs. 3-5 and 7-9) use a very fine-grained colour scale, resulting in weak contrasts between the individual shades. Therefore, it is difficult to estimate any quantitative values from the maps. At least for SLP and T850, traditional isoline plots with labelled contours (with or without colours superimposed) would most likely be more informative.

More detailed comments follow below.

**SPECIFIC COMMENTS**

1. Figure 1. Please provide an absolute scale for the duration of the events
2. L277. Where was -23°C observed, if it was even colder in Marcesina?
3. L325. The lowest temperature in Finland in February 2012 was -42.7C (https://www.ilmatieteenlaitos.fi/lampimin-ja-kylmin-paikka-vuosittain)
4. L434-435. respectively 12 and 20 for k = 1 and k = 2?
5. L436. the 20 events in cluster 2
6. Figure 2. y axis labels are only partly visible
7. L503. (expected to lead to ~+3.5 K SST) This kind of numbers are meaningless without mentioning the emission scenario. From reading Adloff et al. (2015), this number most likely represents the high SRES A2 scenario.
8. L585-587. Are "decreased", "increasing" and "unchanged" the right words, when comparing PLASIM with the real world (NCEP) frequencies? Rather "smaller", "larger" and "the same"?
9. Caption of Table 1: cold spell SLP analogues?
10. Caption of Figure 7. Is this really a natural logarithmic scale from 0 to -15? Exp(-15) would mean about $3*10^{-7}$ kg m$^{-2}$ of snow, which seems incredibly little even in Southern Europe (equivalent to having one day with 1 kg m$^{-2}$ of snow

once in 27000 winters!).

11. L604-605. Can you explain how we get the number 179? Is this number the same for all the events, or is it an average?

12. Figs. 8-9. The SLP fields in these two figures seem much more similar with each other than those for the NCEP reanalysis clusters in Fig. 3. From a visual comparison, it is not even clear that PlaSim cluster 1 more similar to NCEP cluster 1 than 2, and vice versa for PlaSim cluster 2 (although comparison is complicated by the different map areas). Can you comment on why this is the case?

13. L631-632 (much more frequent configurations). As discussed in the general comments, more information would be needed on the dynamical origin of this difference.

**TYPOs and minor linguistic issues**

14. L92. main characteristics
15. L96. information … is repeated
16. L164. A very cold
17. L259. dropped to zero
18. L369. caused / was causing
19. L404. The previous snowfall in Rome
20. L502. one reason / one of the reasons
21. L564. consider as cold spell analogues
22. L592. RCP85
23. Captions of Figs. 8 and 9: in the analogies
24. L622. according to two?
25. L646. this type of events

---

## Referee Report (RR2)

Comments on the revised version of Present and future synoptic circulation patterns associated with cold and snowy spells over Italy. By Miriam D'Errico[1,*], Flavio Pons[1,*], Pascal Yiou[1], Cesare Nardini[2], Frank Lunkeit[3], and Davide Faranda

Remarks
The authors have taken many of the earlier comments by the reviewers serious. The paper has improved considerably but also has seen quite a big change compared to the previous version. I am pleased with some of these, yet I have still difficulties understanding the results. Or perhaps I should say, new difficulties, different from before, because there is a lot of new material. New are a K-means clustering approach, and a focus on the frequency changes of the mslp analogs. Although there is potential in this new part, I have a number of concerns and remarks that may influence results and require new analysis. My main concern is that the most important new result (the "huge" increase in RCP85 cold-spell analog mslp conditions, at least for certain of the cases) is hard to digest without any suggestions as to the why of it. The paper does not offer any help with explanations. The same holds for the almost complete absence of effect on frequency, for the SST+4 runs where PLASIM global oceans are increased by 4 degrees.

Without a more proper interpretation of these results I cannot accept this paper. My recommendation based on this version, is revise with major revisions.

I list some of my main comments below.

1. Section 1 and 2 have not changed much. There still is a multipage long descriptive section 2 on the cases, making rather clear that they are quite different. To me providing the entire list with details on all cases is way too much given the amount of analysis that is undertaken subsequently. I leave the decision up to the editor, but I would be happy to see (some/most of it) put in an appendix.

2. To make some order in the chaos of all cases, the authors decided to conduct a K-means clustering analysis. This could be a useful thing sometimes indeed. They end up with two main clusters. However, there is no real argumentation for this. Figure 2, the scree plot, is poorly formatted with labels dropping off. It also doesn't tell anything as far as I can see, except that there is no favourable grouping. I would put this in supplementary material, but definitely tidy up the graphics!

3. I think the domain chosen for the clusters is *way* too large. Although the authors warn the reader that they do this for a reason, the cluster domain now covers 120 degrees in the zonal direction, which is a 3rd of the earth. Have the authors experimented with using a domain that is more compact, to zoom in slightly more on the actual situation over Italy? Although the subsequent PLASim simulations are of course also rather coarse I think it would help make the analysis more relevant useful.

4. Another basic question, has the clustering be performed on anomalies wrt to a climatology or to the full fields? Because of pre-existing large-scale pressure gradients, a full field framework is not recommended.

5. To augment my previous statement, it could help the authors to examine whether the differences in mslp between the cluster centroids are actually statistically significant over the prime region of interest: Italy. If not the authors have a problem with section 2.3.

6. The quality of figures 3-5 is poor and does hardly provide insight in the way they are presented now. They should be improved. My advice is to combine mslp and T850 in the same plot (or maybe even T2M as well), using shading and contours. And use an anomaly framework! So make these plot wrt a DJF or whatever climatology 1981-2010 or so. And use much tighter colour bands. It is almost not possible to make out differences in temperature at all this way and even for mslp it is hard to see how the flow is organised.

7. Then onwards from section 3, we turn to the model world of PlaSIM. I appreciate that the authors have brought some of my earlier suggestions into practice by focussing less on the thermodynamic aspects. However, the results that are produced by the analogon approach are quite surprising/disturbing/alarming, at least the one for the RCP85 scenario. In there we see spectacular increases in the frequency of cases. Although the world also warms, this might not yield extremely cold/snowy situations in the end (which the authors warn for already), but still.

8. To me, the huge increase seen in RCP85 raises an alarm bell. Why/how does this occur? Many existing climate model ensembles exists (e.g. CMIP5, CMIP6), but as far as I know, none does produce such extreme changes in the tails of the distribution. So the authors at least have to come up with a convincing story here.

9. An 11-step scheme is presented to obtain structures that are similar to each of the 32 events. However, by now focusing on each of them (in the table), the reader may wonder how strange/anomalous each of them was. The readers have no idea about the mslp fields underlying each case, and therefore have no feeling about what the numbers in the tables indicate. Why not simply use the two cluster centroids decided on in section 2.3 and use these to find analogs for?! One could even use these two cluster centroids and search for distribution changes in the way done e.g. in the snow paper by de Vries et al. Clim Dyn. DOI 10.1007/s00382-012-1583-x. (eg their figure 6).

10. The same question I had on the domain size applies here as well. First: is the analoging done on anomalies or full field, and have the authors experimented with the domain size? If the final interpretation is to hold for Italy specifically, it should (I believe) be demonstrated that a domain is chosen that at least for that region provides meaningful results.

11. The rationale for using a +4SST is that the MedSea warms faster than the rest, but in PLASIM the oceanwater is globally raised by 4 degrees. I am then surprised to see that this leads to no adjustment whatsoever.

12. Figure 7 is unclear what we see. Is it climatological mean snowcover? Units seem to be kg/m2, which is probably the same as cm snow. But then showing the snowcover up to natural logarithm values of -15 is rather small/meaningless..

13. Figure 8-9 same story as for figures 3-5. (see above comments)

14. Finally, in those figures RCP85 mslp analogs are combined with the T850 conditions. This reduces the number of cases accordingly.

---

## Referee Report (RR3)

**Review comments on esd-2020-61 "Present and future synoptic circulation patterns associated with cold and snowy spells over Italy", version 30 Nov 2021, by M. D' Errico and co-authors**

**GENERAL COMMENTS**

This is my third review of this manuscript. Again, there have been extensive changes that solve some of the previous problems but also raise new questions and concerns.

I particularly appreciate the condensation of the previous case-by-case results to the cluster-wise analysis in Table 1. The way in which the results are represented in this table is useful, because it shows that the large relative increases in the frequency of the circulation analogues in fact reflect relatively modest increases in the absolute numbers.

As the authors now apply their analogy search to anomalies in Z500, which I presume to have been calculated against the time means in the RCP simulations, the increase in the frequency of good analogies apparently cannot be explained by biases or changes in the time mean state. Yet, the manuscript still leaves it unclear to me why this increase takes place. I realize that it may be difficult to give a fully satisfying answer to this question. However, one way could be to pick up a fixed number (e.g. top 2%) of the best analogies for both the CTRL and the RCP simulations, and calculate the local root-mean-square difference between these and the cluster centroids for each grid point in the analysis domain (22.5º-70ºN, 10ºW-70ºE). Doing this and comparing the resulting maps, it would at least be possible to deduce whether the analogies in the RCP simulations are better in a specific part of the domain or whether the improvement is more or less evenly distributed in the area.

In this version of the manuscript, bias corrections have been introduced that adjust both the time mean bias and the standard deviation. The correction for the mean bias seems redundant as far as the analogies are based on anomaly values only. There is also a risk that the correction of the standard deviation makes more harm than good. If the standard deviation biases in PlaSim are different in different parts of the domain, then the correction of the standard deviation may result in anomaly patterns that differ (e.g., in the locations of their maxima and minima) from the originally simulated patterns. In such a situation, the dynamical interpretation of the results becomes difficult.

What made me pay attention to such potential bias correction artefacts is Fig. 4g-h. These maps show, for the RCP8.5 scenario, Z500 height fields with a strong (> 6000 m?) maximum in the northernmost part of the domain. This is completely different from the patterns in the CTRL (Figs. 4a-b). Moreover, as there is a deep minimum in SLP in the same area in Figs. 6g-h, such a maximum in Z500 would only be consistent with the hydrostatic balance if the temperatures in the northern part of the

domain were much higher than those elsewhere. However, Fig. 7g-h shows that, even for RCP8.5, T850 is lower in the northern than the southern part of the domain. Thus, Z500, SLP and T850 and SLP are physically inconsistent with each other. Such physical inconsistences could result if the correction of standard deviation is also affecting the climate change signal and not only the variability around the mean state.

Therefore, my advice is to repeat the analysis without the standard deviation correction, or at least modify the implementation of this correction so that it has no impact on the time mean state in the RCP simulations.

The second main weakness in the revised manuscript concerns the new Section 4 on the CMIP5 simulations. This seems like a last-minute addition since it is mentioned neither in the abstract nor the concluding section in the manuscript. More importantly, the description of what was done is so brief that it is difficult to understand what the results mean.

On L283-285, it is stated that the (observed?) daily Z500 anomalies for the 32 cold spell events were embedded into the historical simulations for 1951-2000 and the RCP4.5 and RCP8.5 simulations for 2051-2100. Does this mean that you only used the time mean Z500 fields in the CMIP5 simulations, and then generated Z500 fields corresponding to the cold-spell cases by adding the observed anomalies to these time mean fields? This sounds rather different from what you did with PlaSim, for which the analysis focused on a comparison of the simulated and observed anomalies. Or did you in fact use the daily Z500 data in the CMIP5 models and repeat the analysis done for PlaSim (which would require a major effort)?

The follow-up on the two types of bias correction on L291-296 is equally difficult to grasp. First, "bias correction" is a somewhat misleading word for removing the spatially averaged Z500 trend (which I assume to mean the area and time mean difference in Z500 between 2051-2100 and 1951-2000). Second, what does the bias correction do if the spatial mean trend is preserved – do you also correct for the biases in the local 1951-2000 mean values? In either case, it is intractable to me how you constructed the anomalies that were compared with the observed (cluster 1 or cluster 2 mean?) anomalies in the end.

Please add detail to the description of the CMIP5 analysis. Otherwise, the results in Section 4 are impossible to interpret. Also, you should mention this analysis both in the abstract and in Section 5, as it appears to be an important part of your work.

Because of these reasons, the manuscript still requires major revisions. Further comments on the details are given below.

**SPECIFIC COMMENTS**

1. L6-8. The CMIP5 analysis should also be mentioned in the abstract.
2. L27-40. Consider also citing O'Gorman (2014): https://www.nature.com/articles/nature13625
3. The legend in 1 indicates that the size is proportional to the duration. Does this indicate that the *diameter* of the circles if proportional to the duration, or that their *area* is proportional to the duration?
4. Figures 3a-d. Please include the temperature anomalies in the maps. You can add them as contour plots and simultaneously retain the shading for the absolute values.
5. L175. A better general reference to the RCP scenarios: van Vuuren, D.P., Edmonds, J., Kainuma, M. et al. The representative concentration pathways: an overview. Climatic Change 109, 5 (2011). https://doi.org/10.1007/s10584-011-0148-z
6. L177-178. The actual $CO_2$ concentrations in the RCP scenarios are lower. The quoted numbers are equivalent $CO_2$ concentrations, which include the net effect of all anthropogenic greenhouse gas forcing.
7. Table 1, cluster 2, RCP85. +71.3% (in parentheses) appears too small. Should rather be ~121%?
8. Caption of Table 1. Please specify the meaning for the numbers that are given for the RCP scenarios (i) before the parentheses (e.g. 0.978) and (ii) in the parentheses (e.g. 0.977).
9. L283-285. Did I understand this correctly? You simply take the 50-year time mean of Z500 in the CMIP5 simulations and add the observed anomalies in the 32 cold spell cases?
10. L284-285. What was the time resolution of the CMIP5 data? Daily or monthly? Overall, Section 4 is difficult to follow because the methodology is not explained in sufficient detail.
11. L297. Euclidean distance relative to the mean of the cluster (1 or 2) in which the event belongs to?
12. L299. Why does the inclusion of the spatially averaged trend lead to closer analogies? One would expect that, with a general warming of climate, the Z500 anomalies relative to the 1951-2000 mean will become systematically positive. If anything, this should make the analogues worse.
13. Section 5. The CMIP5 analysis should also be mentioned in this section.

**TECHNICAL COMMENTS**

1. L203. to the control and RCP simulations?
2. L214. persistence?
3. L216. cold spells
4. L254. these results
5. L292. that trend / those trends
6. L297. more slightly or more strongly?

7. L298. these decreases
8. Caption of Figure 10, L1-2. each dot represents
9. Caption of Figure 10, L2. given
10. L324. requirements of the Paris agreement (Arias et al. 2021).

---

## Referee Report (RR4)

Comments on the re-revised version of Present and future synoptic circulation patterns associated with cold and snowy spells over Italy, *by M. d'Errico et al.*

January 27, 2022

Remarks
The authors have taken many of the earlier comments by the reviewers seriously. Again the paper has seen big changes. The SST-run has disappeared, and new scenario runs have appeared. Some figures are gone, many new have appeared. Section 4 is new material, showing some analysis with CMIP5 data. I appreciate the extra effort but have difficulties to understand them. I think most of the changes work out well and the paper has improved. However, I still have several points/remarks. My suggestion to the editor is that the paper still requires some revisions prior to acceptance.

Points (mostly remarks to the authors' response document)
1. My previous point 3 ("too large size of the clustering domain") has not been used. Instead, arguments are given why the authors think their large domain (120degree wide in longitude) is necessary. I can follow their arguments, but still hesitate as to its justification, especially if the final target area is that small. Large-scale embedding is essential of course, but in my view the large domain now leads to flow fields not tied very strongly to the region of interest. This implies that the results are perhaps less clear than would otherwise have been possible, in the sense that for the target region, the response fields likely regress to the mean. But in the end, it is the responsibility of the authors to squeeze the best possible results out of the data. *My suggestion* would be to include a brief discussion on how experimenting with domain-size would impact the results. This would also help to address issues raised below under 4 of my minor points.
2. I can agree to the authors' reply to my Point 5 ("significant differences"). If one can argue that the flow is different outside the target area, this should also be sufficient. as it points to different types of air masses involved.
3. It is appreciated that the figures have been improved. Yet I have some further suggestions (see below).
4. On my point 8 ("alarming results of PlaSim"): I am pleased that an error was discovered and corrected for and am happy with the authors' response and putting these results into a better context of existing results/debate. Still the huge increase in frequency is difficult to grasp.
5. On my point 9 ("alternative suggestion"). I am pleased to see that the authors have followed up my suggestion.
6. On my point 11 ("4K ocean"). I think it is good that the authors removed that part in the re-revised version. The additional simulations with different scenarios are a good replacement.

Other comments (mostly minor, except 4).
1. In Section 2.1 (line 53) the Delta-SST-run is mentioned. I think it is not used anymore (see my above point 6). This sentence can be removed.
2. Cluster 2. The authors refer to it (sec 2.2, line 124) as a "Scandinavian" blocking. However, when I look at the MSLP pattern, it doesn't look quite like a

Scandinavian blocking, as the high pressure is much too far south. Maybe the authors could rephrase it into "resembles/is more like a Scandinavian blocking pattern". It could help to show MSLP also as an anomaly. Generally, I think also Figure 3 would benefit from taking an anomaly perspective, with fewer colors. The color-scale for precipitation is not well chosen (I'd use the conventional BrBG option in an (relative)-anomaly framework, again with fewer colors).

3. Table 1: the results for Cluster 2 for RCP8.5 (bottom-right numbers) seem inconsistent. The number between brackets should be 53.4% higher than the number outside the bracket (or, likely, the number outside the bracket should be 53.4% lower than the one inside). Can I assume this is a typo?

4. Figure 4: It is not instructive to start with this figure, as you mainly see the increase of the Z500 level with increased warming levels. Figure 5 is the key figure. It is strikingly similar under all 4 scenarios (as they should I think, since you based your analogs on these patterns). I first wondered why these patterns do not agree to the ones obtained from observations... Similarly for figure 6: These MSLP fields don't look at all like the patterns obtained earlier based on the observations (Figure 2e-f)? Based on these figures I *would call neither of them Scandinavian blocking*. The most pronounced features in Z500 (fig 5) are the low's rather than the highs (The highs are far away over the Atlantic). The reason that the patterns are different of course is that you now search for analogues in a much smaller domain, more connected to the flow in the region of the target region. I do appreciate this shift, but it somehow conflicts with your earlier arguments (of choosing a wide domain for the clustering). Why did you first construct cluster centroids for a larger domain (making them not very different in the target area) but subsequently use a smaller domain for the analogues? I would rather use the same domains for both exercises. The current choice makes the story much more complex than needed.

5. Figure 7-9: Again, using an anomaly framework would be much more instructive. Now, upon visual inspection one can see hardly any difference between Cluster 1 and 2.

6. Section 4: I cannot really follow what is being done here based on the information presented. It is stated that the "We then embed these observed events into historical simulations...". What does this mean? Do you project all days in CMIP5 for each winter and model on the observed cold spells, and compute their Euclidean distance? If so, why would a average reduction of the distance between the observed (cold-spell) and modelled (CMIP5) circulation be an indication that the cold-spells become more frequent? Or do you perhaps select cold days in some way from CMIP5 first? In its current formulation I cannot see how the information in Figure 10 helps us.

---

## Referee Report (RR5)

**Review of "Present and future synoptic circulation patterns associated with cold and snowy spells over Italy", version 9.3.2022, by M. D' Errico et al.**

**GENERAL COMMENTS**

This is my fourth review of this manuscript. I believe that most of the major problems have been settled, but I still need to return to the bias correction issue. The authors do wisely when not applying it to Z500 and SLP, but the results in their Figs. 8-9 suggest that they should seriously reconsider the cases of T850 and T2M as well.

In particular, Figures 9g-h show a completely unrealistic pattern with 2-m winter temperatures exceeding 25°C in Greenland, combined with an abrupt shift to more reasonable (5-10°C) values to the north-west of Iceland. Spurious local hotspots also appear elsewhere in the same maps, for example over the Gulf of Finland. I am convinced that the patterns in the original PlaSIM data are much smoother and more physically reasonable.

The likely cause of this problem is the standard deviation correction included in the bias correction method. I assume that the transformation

$$T_{corr} = (\bar{T}_{obs} - \bar{T}_{ctrl}) + \frac{s_{obs}}{s_{ctrl}}(T - \bar{T}_{ctrl})$$

has been used, where $T$ is the original and $T_{corr}$ the corrected temperature, $\bar{T}_{obs}$ and $\bar{T}_{ctrl}$ are the mean values in the observations and the control simulation, and $s_{obs}$ and $s_{ctrl}$ are the corresponding standard deviations. The problem is that the latter term, which is needed to ensure that the standard deviation in the control simulation agrees with observations, also affects the mean climate change in the model if $s_{obs}$ and $s_{ctrl}$ differ. Denoting the temperatures in a future RCP simulation as $T_{RCP}$, their mean value after the bias correction becomes

$$\bar{T}_{RCP,corr} = (\bar{T}_{obs} - \bar{T}_{ctrl}) + \frac{s_{obs}}{s_{ctrl}}(\bar{T}_{RCP} - \bar{T}_{ctrl})$$

whereas the mean for the control simulation is simply

$$\bar{T}_{corr} = \bar{T}_{obs}$$

Taking the difference, we get

$$\bar{T}_{RCP,corr} - \bar{T}_{corr} = \frac{s_{obs}}{s_{ctrl}}(\bar{T}_{RCP} - \bar{T}_{ctrl}) - \bar{T}_{ctrl} = (\frac{s_{obs}}{s_{ctrl}} - 1)(\bar{T}_{RCP} - \bar{T}_{ctrl})$$

For example, if the standard deviation in the control simulation is too small, ($s_{ctrl} <$

$s_{obs}$), the bias correction amplifies the change in the mean temperature and, therefore, also the highest temperatures. I suspect that the extremely high temperatures in (e.g.) Greenland in Figs. 9g-h result from this effect.

To avoid this problem, a mean-conserving version of the bias correction should be used instead. When the corrected daily temperatures in the RCP simulations are defined as

$$T_{RCP,corr} = (\bar{T}_{obs} + \bar{T}_{RCP} - \bar{T}_{ctrl}) + \frac{s_{obs}}{s_{ctrl}}(T_{RCP} - \bar{T}_{RCP})$$

then

$$\bar{T}_{RCP,corr} - \bar{T}_{corr} = (\bar{T}_{obs} + \bar{T}_{RCP} - \bar{T}_{ctrl}) - \bar{T}_{obs} = \bar{T}_{RCP} - \bar{T}_{ctrl}$$

just as expected.

Alternatively, the problems associated with the standard deviation correction could be avoided simply by omitting this correction, i.e., by only correcting the time mean bias.

**SPECIFIC COMMENTS**

1. L122. Why 18? On L120 the numbers 22 and 10 are mentioned.
2. Figure 4. The small numerical values indicate that the units are not as given in the figure headers (m and hPa). I assume that the values are non-dimensional.
3. Figure 5. Same comment as for Figure 4.
4. L179-180. Earlier (L172-173) ten levels were mentioned, but here only five. Which is correct?
5. L241. Z500 anomaly or standardized Z500 anomaly?
6. L248. $p_1$ or $\pi_{1,r}$? Table 1 seems to report both alternatives.
7. L281-283. Do you see similar "winter heat waves" without using bias correction? The fields for T2M in Fig. 9 look so unphysical (particularly for RCP8.5) that I strongly suspect that they are an artifact of the standard deviation adjustment in your bias correction technique (see the general comments).
8. L293. Consider putting Figures 11-15 in an Appendix or in Supplementary material. They take too much space from the main article compared with their information content, or the amount of text devoted to them.
9. Figures 11-15. It seems that the values are dimensionless, and not in the units indicated in the figure headings.
10. Caption of Figure 11. Root mean square difference in standardized geopotential height [standard deviations]? The same also applies to captions of Figs. 12-15.

**TECHNICAL COMMENTS**

1. L54. Section 4
2. L55. "Cold" with capital C

3. L99. Section 2.1
4. L102. Put "Jézéquel et al. 2018" in parentheses.
5. L141. hPa
6. L159. by high uncertainty
7. L197. the the
8. L232. persistence
9. Table 1. Why are 6-7 decimals used for RCP2.6, instead of just 4?
10. L297. "Similar" with capital S
11. L317. to be
12. L322. Temperatures are only shown in Figures 8-9.

---

## Author Response (AR2)

**Dear editor,**

**We are glad to resubmit the article titled "Present and future synoptic circulation patterns associated with cold and snowy spells over Italy" to be considered for publication on Earth System Dynamics.**

**We took several actions to improve the paper according to and beyond the suggestions of the reviewers. In particular, we produced new climate simulations to correct an issue in a physical parameterization in the PlaSim model, and we took a chance to add two more RCP scenarios, including one that is representative of the current climate targets according to Paris agreements. Soulivanh Thao has taken care of performing the simulations and he has been added to the authors'list.**

**We now base our analysis on an anomaly framework rather than on full atmospheric fields, as recommended by both reviewers. To improve readability, we moved the detailed description of the phenomena associated to each cold spell of interest to an appendix, and we changed all the figures, adopting a custom colorscale for better graphical clarity.**

**We are confident that the changes we made satisfy the requirements of the two reviewers, and that they have actually improved both the scientific quality of the results and the cogency of the article.**

**Best regards**
**Flavio Pons & Davide Faranda on behalf of all the authors**

———————————————————————————————————————————

Comments on the revised version of Present and future synoptic
circulation patterns associated with cold and snowy spells
over Italy. By Miriam D'Errico1,*, Flavio Pons1,*, Pascal Yiou1, Cesare Nardini2,
Frank Lunkeit3, and Davide Faranda

Remarks
The authors have taken many of the earlier comments by the reviewers seriously. The paper has improved considerably but also has seen quite a big change compared to the previous version. I am pleased with some of these, yet I have still difficulties understanding the results. Or perhaps I should say, new difficulties, different from before, because there is a lot of new material. New are a K-means clustering approach, and a focus on the frequency changes of the mslp analogs. Although there is potential in this new part, I have a number of concerns and remarks that may influence results and require new analysis. My main concern is that the most important new result (the "huge" increase in RCP85 cold-spell analog mslp conditions, at least for certain of the cases) is hard to digest without any suggestions as to the why of it. The paper does not offer any help with explanations. The same holds for the almost complete absence of effect on frequency, for the SST+4 runs where PLASIM global oceans are increased by 4

degrees.

Without a more proper interpretation of these results I cannot accept this paper. My recommendation based on this version, is revise with major revisions.

I list some of my main comments below.

1. Section 1 and 2 have not changed much. There still is a multipage long descriptive section 2 on the cases, making rather clear that they are quite different. To me providing the entire list with details on all cases is way too much given the amount of analysis that is undertaken subsequently. I leave the decision up to the editor, but I would be happy to see (some/most of it) put in an appendix.

**We thank the reviewer and we agree with them, the detailed description of the cold spells has been moved to an Appendix.**

2. To make some order in the chaos of all cases, the authors decided to conduct a K-means clustering analysis. This could be a useful thing sometimes indeed. They end up with two main clusters. However, there is no real argumentation for this. Figure 2, the scree plot, is poorly formatted with labels dropping off. It also doesn't tell anything as far as I can see, except that there is no favourable grouping. I would put this in supplementary material, but definitely tidy up the graphics!

**We agree with the reviewer that the scree plot is poorly informative. As now pointed out in the text, this is probably due to the fact that gridded atmospheric fields are not an ideal type of dataset to satisfy technical requirements for k-means clustering, and in our case the cluster sizes also differ. However, k-means is also an established way to find weather regimes. In the new version, we remove the scree plot since it is not informative. The ratio of choosing 2 clusters comes from the fact that we tried with 3 and 4 groups, and in this case we obtain redundant clusters containing essentially the same configuration. We specify this in the text.**

3. I think the domain chosen for the clusters is *way* too large. Although the authors warn the reader that they do this for a reason, the cluster domain now covers 120 degrees in the zonal direction, which is a 3rd of the earth. Have the authors experimented with using a domain that is more compact, to zoom in slightly more on the actual situation over Italy? Although the subsequent PLASim simulations are of course also rather coarse I think it would help make the analysis more relevant useful.

**We decided to include this domain because the patterns influencing cold spell weather at the European level have a larger scale, and it would be more difficult to discriminate between them. From the maps in Fig. 3 it is evident that, while the position of the high and low pressures over Europe changes between clusters, the biggest difference is found between the Atlantic and North America, with a basically inverted position of positive and negative geopotential anomalies. Including this**

**portion of the domain helps better discriminate between the two weather regimes associated to Mediterranean cold spells.**

4. Another basic question, has the clustering be performed on anomalies wrt to a climatology or to the full fields? Because of pre-existing large-scale pressure gradients, a full field framework is not recommended.

**Thank you for the suggestion, it is now clarified in the text, we used standardized anomalies respect to the DJFM climatology of the historical period.**

5. To augment my previous statement, it could help the authors to examine whether the differences in mslp between the cluster centroids are actually statistically significant over the prime region of interest: Italy. If not the authors have a problem with section 2.3.

**We thank the reviewer for this suggestion; we have now given an extended description of the dynamical differences between the two clusters that is especially visible in the Z500 standardized anomalies and the SLP patterns.**

**More in general, we do not agree that a statistical test over the region is necessary nor sufficient to explain differences between the two regimes. Even in the case the SLP and Z500 anomalies over Italy were not significantly different, it is the position of the coupled high pressure that drives cold air over different paths (the Rhone Valley in the Atlantic ridge case, or from Russia with a NE-SW direction on the Scandinavian blocking case).**

6. The quality of figures 3-5 is poor and does hardly provide insight in the way they are presented now. They should be improved. My advice is to combine mslp and T850 in the same plot (or maybe even T2M as well), using shading and contours. And use an anomaly framework! So make these plot wrt a DJF or whatever climatology 1981-2010 or so. And use much tighter colour bands. It is almost not possible to make out differences in temperature at all this way and even for mslp it is hard to see how the flow is organised.

**We thank the reviewers for the suggestions, the figures have been completely re-made in the current version of the paper.**

7. Then onwards from section 3, we turn to the model world of PlaSIM. I appreciate that the authors have brought some of my earlier suggestions into practice by focussing less on the thermodynamic aspects. However, the results that are produced by the analogon approach are quite surprising/disturbing/alarming, at least the one for the RCP85 scenario. In there we see spectacular increases in the frequency of cases. Although the world also warms, this might not yield extremely cold/snowy situations in the end (which the authors warn for already), but still.

8. To me, the huge increase seen in RCP85 raises an alarm bell. Why/how does this occur? Many existing climate model ensembles exists (e.g. CMIP5, CMIP6), but as

far as I know, none does produce such extreme changes in the tails of the distribution. So the authors at least have to come up with a convincing story here.

**We thank the reviewer for these comments. Indeed, we found that the previous PlaSim run contained an error due to a problem in the parameterization of ocean fluxes. We performed new simulations correcting the problem and including more emission scenarios. We still observe an increase in the frequency of these analogues with increasing $CO_2$ concentration, but results are less dramatic.**
**As we point out in the article, the increased frequency of these configurations may be due to a more wavy jet-stream under climate change due to reduced meridional temperature gradient. We are aware and we state in the article that there is no consensus about this, and that the result's validity is limited to the framework of this specific model.**

9. An 11-step scheme is presented to obtain structures that are similar to each of the 32 events. However, by now focusing on each of them (in the table), the reader may wonder how strange/anomalous each of them was. The readers have no idea about the mslp fields underlying each case, and therefore have no feeling about what the numbers in the tables indicate. Why not simply use the two cluster centroids decided on in section 2.3 and use these to find analogs for?! One could even use these two cluster centroids and search for distribution changes in the way done e.g. in the snow paper by de Vries et al. Clim Dyn. DOI 10.1007/s00382-012-1583-x. (eg their figure 6).

**We thank the reviewer for this suggestion, we agree that this makes more sense and we decided to follow and implement it. In the new version, we consider analogues of the Z500 anomaly fields averaged over the two clusters.**

10. The same question I had on the domain size applies here as well. First: is the analoging done on anomalies or full field, and have the authors experimented with the domain size? If the final interpretation is to hold for Italy specifically, it should (I believe) be demonstrated that a domain is chosen that at least for that region provides meaningful results.

**Thank you for the question. We use the extended domain only to find the regimes associated to the two clusters. For analogues search, we use the smaller domain indicated in Section 3.3. Figures 5 and 6 show that these analogues catch well the difference between the two configurations over the region of interest, with the high pressure centred over the UK for analogues of cluster 1, and over Scandinavia for analogues of cluster 2.**

11. The rationale for using a +4SST is that the MedSea warms faster than the rest, but in PLASIM the oceanwater is globally raised by 4 degrees. I am then surprised to see that this leads to no adjustment whatsoever.

**We thank the reviewer for the comment. Indeed, we realised that the PlaSim framwork did not allow to tackle properly the role of Mediterranean SSTs for this and other**

**reasons, and we decided to remove it from our study, and focus on more emission scenarios instead.**

12. Figure 7 is unclear what we see. Is it climatological mean snowcover? Units seem to be kg/m2, which is probably the same as cm snow. But then showing the snowcover up to natural logarithm values of -15 is rather small/meaningless..

13. Figure 8-9 same story as for figures 3-5. (see above comments)

14. Finally, in those figures RCP85 mslp analogs are combined with the T850 conditions. This reduces the number of cases accordingly.

**Thank you for these comments. Figures have been re-made completely, and we do not consider combined analogues anymore.**
* * *
–
* * *
–

Review of "Present and future synoptic circulation patterns associated with cold and snowy spells over Italy" by M. D' Errico et al.

GENERAL COMMENTS

The manuscript has been extensively revised from its first version. The catalogue of combined heavy snowfall and low temperature events in Italy between the years 1954 and 2018 is retained, but large changes have been made to the rest of the paper. First, a cluster analysis has been used to divide the observed cold spells to two clusters based on the associated sea level pressure (SLP) fields. Second, the analysis of the intermediate complexity PlaSim climate model simulations has been refocused on the search of situations that are analogous to the 32 identified cold spells in terms of the SLP and the 850 hPa temperature (T850) distributions.

The refocusing of the PlaSim analysis from snowfall to the atmospheric dynamical features has improved the manuscript, since the model is clearly more suitable for the simulation of the latter than the former. Furthermore, the revised manuscript is honest in acknowledging that cold spells in warmer future climate will produce less snow.

However, I still have some questions and concerns related to the interpretation of the main results. The analysis of the PlaSim simulations suggests that the frequency of SLP circulation states that resemble those in the observed cold spells is increasing strongly in the RCP8.5 scenario, whereas the corresponding change resulting for an artificial 4 K warming of the sea surface temperatures is negligible. To me this result is counterintuitive, and its physical significance is difficult to assess without further information on what the increase in RCP8.5 results from. Specifically, the increase in the frequency of circulation analogies could be associated with

1. A change in the winter mean SLP field that makes the average SLP distribution more similar to that observed during the cold spells, or
2. A change in the variability of SLP around its mean state, resulting in a larger frequency of SLP anomaly fields that resemble the SLP anomaly fields during the cold spells.

These two possibilities could be distinguished by repeating the cluster analysis for SLP anomalies relative to the observed or simulated (present-day and RCP8.5 separately) DJFM mean SLP field. If the increase in frequency is still seen when the mean state change has been eliminated by focusing on the anomalies, if must originate from changes in the simulated variability. If it disappears, then the change in the mean SLP field is the key.

Furthermore, if the change in the mean SLP field turns out to explain the increase in the frequency of the circulation analogies, a follow-up question is how and where the mean state changes. In which parts of the (rather large) analysis domain (22.5-70°N, 80°W-50°E) does the new mean state approach the SLP fields during the observed cold spells?

**We thank the reviewer for this comment. Indeed, we moved to an anomaly framework, and we also decided to use Z500 instead of SLP for clustering.**

Finally, biases in the simulated present-day winter mean SLP field might affect the change in the frequency of the circulation anomalies in a non-intuitive way. To check for this possibility, it would be prudent to repeat the analysis after also applying the simple linear scaling bias correction to SLP, not only to T850 as was apparently done.

**We thank the reviewer for the suggestion, we proceeded to apply the scaling bias correction also to SLP and Z500 fields to eliminate the effect of biases.**

I also have some concern about the ability of the clustering algorithm to identify good circulation anomalies in the model simulations. There appear to be quite large differences between the observed (Fig. 3) and the simulated (Figs. 8-9) SLP fields for both two clusters of the cold spell cases. Furthermore, the difference between the two clusters appears much smaller for the simulations. I wonder if this might be improved by selecting a somewhat smaller domain in the search of the circulation analogies.

**We thank the reviewer for the comment. Besides these issues, we also realised there was a problem with the parameterization of fluxes in PlaSim, which we corrected before running new simulations. In the new version of the paper, we directly look for analogues of the two clusters; results obtained from the simulations (Fig 5 and 6) are in good agreement with the fields associated to the two clusters.**

Aside of this issue, the figures need improvement. All the maps (Figs. 3-5 and 7-9) use a very fine-grained colour scale, resulting in weak contrasts between the individual shades. Therefore, it is difficult to estimate any quantitative values from the maps. At least for SLP and T850, traditional isoline plots with labelled contours (with

or without colours superimposed) would most likely be more informative.

**We thank the reviewer for the comment; after trying isoline plots and also considering the comments from the other reviewer, we produce new figures with an enhanced colorscale, but we decided to keep a map format. We believe that the new maps are nonetheless clear and easily interpretable by the reader.**

More detailed comments follow below.

SPECIFIC COMMENTS

**We thank for the detailed comments, they have been addressed in the new version of the paper, or they refer to sentences that have been removed or deeply changed based on the new results.**

1. Figure 1. Please provide an absolute scale for the duration of the events
2. L277. Where was -23°C observed, if it was even colder in Marcesina?
3. L325. The lowest temperature in Finland in February 2012 was -42.7C (https://www.ilmatieteenlaitos.fi/lampimin-ja-kylmin-paikka-vuosittain)
4. L434-435. respectively 12 and 20 for k = 1 and k = 2?
5. L436. the 20 events in cluster 2
6. Figure 2. y axis labels are only partly visible
7. L503. (expected to lead to ~+3.5 K SST) This kind of numbers are meaningless without mentioning the emission scenario. From reading Adloff et al. (2015), this number most likely represents the high SRES A2 scenario.
8. L585-587. Are "decreased", "increasing" and "unchanged" the right words, when comparing PLASIM with the real world (NCEP) frequencies? Rather "smaller", "larger" and "the same"?
9. Caption of Table 1: cold spell SLP analogues?
10. Caption of Figure 7. Is this really a natural logarithmic scale from 0 to -15? Exp(-15) would mean about $3*10^{-7}$ kg m-2 of snow, which seems incredibly little even in Southern Europe (equivalent to having one day with 1 kg m-2 of snow

---

## Author Response (AR3)

**Dear editor,**

**We are glad to resubmit the article titled "Present and future synoptic circulation patterns associated with cold and snowy spells over Italy" to be considered for publication on Earth System Dynamics.**

**We took several actions to improve the paper according to the suggestions of the reviewers. We are confident that the changes we made satisfy the requirements of the two reviewers, and that they have actually improved both the scientific quality of the results and the cogency of the article.**

**Best regards**
**Flavio Pons & Davide Faranda on behalf of all the authors**

**Reviewer 1**

Review comments on esd-2020-61 "Present and future synoptic circulation patterns associated with cold and snowy spells over Italy", version 30 Nov 2021, by M. D' Errico and co-authors

**GENERAL COMMENTS**
This is my third review of this manuscript. Again, there have been extensive changes that solve some of the previous problems but also raise new questions and concerns. I particularly appreciate the condensation of the previous case-by-case results to the cluster-wise analysis in Table 1. The way in which the results are represented in this table is useful, because it shows that the large relative increases in the frequency of the circulation analogues in fact reflect relatively modest increases in the absolute numbers.
As the authors now apply their analogy search to anomalies in Z500, which I presume to have been calculated against the time means in the RCP simulations, the increase in the frequency of good analogies apparently cannot be explained by biases or changes in the time mean state. Yet, the manuscript still leaves it unclear to me why this increase takes place. I realize that it may be difficult to give a fully satisfying answer to this question. However, one way could be to pick up a fixed number (e.g. top 2%) of the best analogies for both the CTRL and the RCP simulations, and calculate the local root-mean-square difference between these and the cluster centroids for each grid point in the analysis domain (22.5º-70ºN, 10ºW-70ºE). Doing this and comparing the resulting maps, it would at least be possible to deduce whether the analogies in the RCP simulations are better in a specific part of the domain or whether the improvement is more or less evenly distributed in the area.

**We thank the reviewer for the suggestion. This analysis is now included in the new version of the manuscript.**

In this version of the manuscript, bias corrections have been introduced that adjust both the time mean bias and the standard deviation. The correction for the mean bias seems redundant as far as the analogies are based on anomaly values only. There is also a risk that the correction of the standard deviation makes more harm than good. If the standard deviation biases in PlaSim are different in different parts of the domain, then the correction of the standard deviation may result in anomaly patterns that differ

(e.g., in the locations of their maxima and minima) from the originally simulated patterns. In such a situation, the dynamical interpretation of the results becomes difficult.

What made me pay attention to such potential bias correction artefacts is Fig. 4g-h. These maps show, for the RCP8.5 scenario, Z500 height fields with a strong (> 6000 m?) maximum in the northernmost part of the domain. This is completely different from the patterns in the CTRL (Figs. 4a-b). Moreover, as there is a deep minimum in SLP in the same area in Figs. 6g-h, such a maximum in Z500 would only be consistent with the hydrostatic balance if the temperatures in the northern part of the domain were much higher than those elsewhere. However, Fig. 7g-h shows that, even for RCP8.5, T850 is lower in the northern than the southern part of the domain. Thus, Z500, SLP and T850 and SLP are physically inconsistent with each other. Such physical inconsistences could result if the correction of standard deviation is also affecting the climate change signal and not only the variability around the mean state. Therefore, my advice is to repeat the analysis without the standard deviation correction, or at least modify the implementation of this correction so that it has no impact on the time mean state in the RCP simulations.

**We thank the reviewer for these insightful comments. We agree about the - previously overlooked by us - physical inconsistency of the results obtained with BC data. Moreover, bias correction is not usually performed on variables such as SLP and Z500. We do not perform any BC on the two dynamical observables in the current version of the article. However, we still perform linear transform BC on near-surface and 850 hPa temperatures, and a quantile mapping BC on precipitation, as these variables must be unbiased to correctly evaluate the likelihood of cold and snowy conditions under different forcing scenarios.**

The second main weakness in the revised manuscript concerns the new Section 4 on the CMIP5 simulations. This seems like a last-minute addition since it is mentioned neither in the abstract nor the concluding section in the manuscript. More importantly, the description of what was done is so brief that it is difficult to understand what the results mean.

On L283-285, it is stated that the (observed?) daily Z500 anomalies for the 32 cold spell events were embedded into the historical simulations for 1951-2000 and the RCP4.5 and RCP8.5 simulations for 2051-2100. Does this mean that you only used the time mean Z500 fields in the CMIP5 simulations, and then generated Z500 fields corresponding to the cold-spell cases by adding the observed anomalies to these time mean fields? This sounds rather different from what you did with PlaSim, for which the analysis focused on a comparison of the simulated and observed anomalies. Or did you in fact use the daily Z500 data in the CMIP5 models and repeat the analysis done for PlaSim (which would require a major effort)?

The follow-up on the two types of bias correction on L291-296 is equally difficult to grasp. First, "bias correction" is a somewhat misleading word for removing the spatially averaged Z500 trend (which I assume to mean the area and time mean difference in Z500 between 2051-2100 and 1951-2000). Second, what does the bias correction do if the spatial mean trend is preserved – do you also correct for the biases in the local 1951-2000 mean values? In either case, it is intractable to me how you constructed the anomalies that were compared with the observed (cluster 1 or cluster 2 mean?) anomalies in the end.

Please add detail to the description of the CMIP5 analysis. Otherwise, the results in Section 4 are impossible to interpret. Also, you should mention this analysis both in the abstract and in Section 5, as it appears to be an important part of your work. Because of these reasons, the manuscript still requires major revisions. Further

comments on the details are given below.

**We agree with the reviewer that the analysis of the CMIP5 models would deserve more information. After discussing with the coauthors and in light of the referees' comments, we decided to remove this analysis from the present manuscript and to postpone its publication in another work that will be dedicated to forced CMIP5 and CMIP6 analyses. The main reason is that we cannot really perform on forced simulations the same procedure applied for PlaSim simulations with stationary forcing.**

**SPECIFIC COMMENTS**

1. L6-8. The CMIP5 analysis should also be mentioned in the abstract.

**The CMIP5 analysis has been removed from the manuscript (see below)**

2. L27-40. Consider also citing O'Gorman (2014): https://www.nature.com/articles/nature13625

**We thank the reviewer for the suggestion of this paper that we had somehow overlooked, we added a citation of their main result.**

3. The legend in 1 indicates that the size is proportional to the duration. Does this indicate that the diameter of the circles if proportional to the duration, or that their area is proportional to the duration?

**The duration is proportional to the diameter, as now indicated in the caption of the figure.**

4. Figures 3a-d. Please include the temperature anomalies in the maps. You can add them as contour plots and simultaneously retain the shading for the absolute values.

**Thank you for the suggestion, we now present absolute values in color shading and standardized anomalies as isolines.**

5. L175. A better general reference to the RCP scenarios: van Vuuren, D.P., Edmonds, J., Kainuma, M. et al. The representative concentration pathways: an overview. Climatic Change 109, 5 (2011). https://doi.org/10.1007/s10584-011-0148-z

**Thank you, we replaced the previous citation with your suggestion.**

6. L177-178. The actual $CO_2$ concentrations in the RCP scenarios are lower. The quoted numbers are equivalent $CO_2$ concentrations, which include the net effect of all anthropogenic greenhouse gas forcing.

**We thank the reviewer for the observation. Indeed, PlaSim simulations are based on equivalent $CO_2$ concentrations including the net effect of all GHG. We now specify this in the indicated lines, and we use the expression "equivalent $CO_2$ concentration" throughout the article.**

7. Table 1, cluster 2, RCP85. +71.3% (in parentheses) appears too small. Should rather be ~121%?

**Thank you, indeed there was a typo. However, all values changed due to the differences in our data analysis strategy.**

8. Caption of Table 1. Please specify the meaning for the numbers that are given for the RCP scenarios (i) before the parentheses (e.g. 0.978) and (ii) in the parentheses (e.g. 0.977).

**Thank you, we proceeded to specify this detail in the caption.**

9. L283-285. Did I understand this correctly? You simply take the 50-year time mean of Z500 in the CMIP5 simulations and add the observed anomalies in the 32 cold spell cases?
10. L284-285. What was the time resolution of the CMIP5 data? Daily or monthly? Overall, Section 4 is difficult to follow because the methodology is not explained in sufficient detail.
11. L297. Euclidean distance relative to the mean of the cluster (1 or 2) in which the event belongs to?
12. L299. Why does the inclusion of the spatially averaged trend lead to closer analogies? One would expect that, with a general warming of climate, the Z500 anomalies relative to the 1951-2000 mean will become systematically positive. If anything, this should make the analogues worse.
13. Section 5. The CMIP5 analysis should also be mentioned in this section.

**After discussing with the coauthors and in light of the referees' comments, we decided to remove this analysis from the present manuscript and to postpone its publication in another work that will be dedicated to forced CMIP5 and CMIP6 analyses. The main reason is that we cannot really perform on forced simulations the same procedure applied for PlaSim simulations with stationary forcing.**

**TECHNICAL COMMENTS**
1. L203. to the control and RCP simulations?
2. L214. persistence?
3. L216. cold spells
4. L254. these results
5. L292. that trend / those trends
6. L297. more slightly or more strongly?
7. L298. these decreases
8. Caption of Figure 10, L1-2. each dot represents
9. Caption of Figure 10, L2. given
10. L324. requirements of the Paris agreement (Arias et al. 2021).
* * *
* * *
**Reviewer 2**

Comments on the re-revised version of Present and future synoptic circulation patterns associated with cold and snowy spells over Italy, by M. d'Errico et al.

January 27, 2022

**Remarks**

The authors have taken many of the earlier comments by the reviewers seriously. Again the paper has seen big changes. The SST-run has disappeared, and new scenario runs have appeared. Some figures are gone, many new have appeared. Section 4 is new material, showing some analysis with CMIP5 data. I appreciate the extra effort but have difficulties to understand them. I think most of the changes work out well and the paper has improved. However, I still have several points/remarks. My suggestion to the editor is that the paper still requires some revisions prior to acceptance.

Points (mostly remarks to the authors' response document)

1. My previous point 3 ("too large size of the clustering domain") has not been used. Instead, arguments are given why the authors think their large domain (120degree wide in longitude) is necessary. I can follow their arguments, but still hesitate as to its justification, especially if the final target area is that small. Large-scale embedding is essential of course, but in my view the large domain now leads to flow fields not tied very strongly to the region of interest. This implies that the results are perhaps less clear than would otherwise have been possible, in the sense that for the target region, the response fields likely regress to the mean. But in the end, it is the responsibility of the authors to squeeze the best possible results out of the data. My suggestion would be to include a brief discussion on how experimenting with domain-size would impact the results. This would also help to address issues raised below under 4 of my minor points.

**We thank the reviewer for the observation. Initially, we used a large domain to find the clusters, to get an idea of the large-scale configurations leading to cold spells over Italy, but we considered the smaller domain sufficient to detect analogues producing similar phenomena over the region. As suggested, we repeated our analysis using the same domain for clustering and analogue search, while implementing requests from the other reviewer, along with the correction of issues we detected while revising our work. We find that, indeed, using the same domain is more appropriate, but results are clearer using the larger domain for both analyses, rather than the smaller one. In particular, we find that Z500 analogues found with the smaller domain produce SLP and temperature composites that are less compatible with the cluster centroids compared to the composites of analogues found using the larger domain.**

2. I can agree to the authors' reply to my Point 5 ("significant differences"). If one can argue that the flow is different outside the target area, this should also be sufficient. as it points to different types of air masses involved.

**We are happy that the reviewer agrees with our view.**

3. It is appreciated that the figures have been improved. Yet I have some further suggestions (see below).

**We have taken the further suggestions on figures into account**

4. On my point 8 ("alarming results of PlaSim"): I am pleased that an error was discovered and corrected for and am happy with the authors' response and putting these results into a better context of existing results/debate. Still the huge increase in frequency is difficult to grasp.

**There are now few new studies that confirm important modifications in the atmospheric circulation patterns with climate change. We have added the following discussion: "This increased frequency of Atlantic Ridge and Scandinavian blocking patterns in PlaSim simulations and warming scenarios could be associated to a wealth of phenomena driven by the mean anthropogenic climate change but still debated in the current scientific literature such as the Arctic Amplification or the increased land-sea temperature contrast (Cohen et al., 2020; Hamouda et al., 2021). Contrary arguments show that there is an increase in flow zonality over the North Atlantic but mostly for the Autumn (de Vries et al., 2013) and the summer seasons (Fabiano et al., 2021)."**

5. On my point 9 ("alternative suggestion"). I am pleased to see that the authors have followed up my suggestion.

**Thank you**

6. On my point 11 ("4K ocean"). I think it is good that the authors removed that part in the re-revised version. The additional simulations with different scenarios are a good replacement.

**Other comments (mostly minor, except 4).**

1. In Section 2.1 (line 53) the Delta-SST-run is mentioned. I think it is not used anymore (see my above point 6). This sentence can be removed.

**Thank you, we removed the erroneously retained mention of the 4SST run.**

2. Cluster 2. The authors refer to it (sec 2.2, line 124) as a "Scandinavian" blocking. However, when I look at the MSLP pattern, it doesn't look quite like a Scandinavian blocking, as the high pressure is much too far south. Maybe the authors could rephrase it into "resembles/is more like a Scandinavian blocking pattern". It could help to show MSLP also as an anomaly. Generally, I think also Figure 3 would benefit from taking an anomaly perspective, with fewer colors. The color-scale for precipitation is not well chosen (I'd use the conventional BrBG option in an (relative)-anomaly framework, again with fewer colors).

**We thank the reviewer for the suggestion. We no longer refer to the configuration as "Scandinavian blocking". We changed the colorscale for precipitation and, also following the suggestions of the other reviewer, we now show full fields as color shading and standardized anomalies as isolines.**

3. Table 1: the results for Cluster 2 for RCP8.5 (bottom-right numbers) seem inconsistent. The number between brackets should be 53.4% higher than the number outside the bracket (or, likely, the number outside the bracket should be 53.4% lower than the one inside). Can I assume this is a typo?

**We thank the reviewer for the observation. Indeed, this was a typo. Since we now use a larger domain for all analyses and other adjustments have been made, while the general tendency to an increase in frequency is preserved, absolute numbers are overall different from the previous version.**

4. Figure 4: It is not instructive to start with this figure, as you mainly see the

increase of the Z500 level with increased warming levels. Figure 5 is the key figure. It is strikingly similar under all 4 scenarios (as they should I think, since you based your analogs on these patterns). I first wondered why these patterns do not agree to the ones obtained from observations... Similarly for figure 6: These MSLP fields don't look at all like the patterns obtained earlier based on the observations (Figure 2e-f)? Based on these figures I would call neither of them Scandinavian blocking. The most pronounced features in Z500 (fig 5) are the low's rather than the highs (The highs are far away over the Atlantic). The reason that the patterns are different of course is that you now search for analogues in a much smaller domain, more connected to the flow in the region of the target region. I do appreciate this shift, but it somehow conflicts with your earlier arguments (of choosing a wide domain for the clustering). Why did you first construct cluster centroids for a larger domain (making them not very different in the target area) but subsequently use a smaller domain for the analogues? I would rather use the same domains for both exercises. The current choice makes the story much more complex than needed.

**Thank you for the suggestions. After exploring both possibilities, we decided that it is appropriate to use the same domain, but we chose to maintain the larger one.**

5. Figure 7-9: Again, using an anomaly framework would be much more instructive. Now, upon visual inspection one can see hardly any difference between Cluster 1 and 2.

6. Section 4: I cannot really follow what is being done here based on the information presented. It is stated that the "We then embed these observed events into historical simulations...". What does this mean? Do you project all days in CMIP5 for each winter and model on the observed cold spells, and compute their Euclidean distance? If so, why would a average reduction of the distance between the observed (cold-spell) and modelled (CMIP5) circulation be an indication that the cold-spells become more frequent? Or do you perhaps select cold days in some way from CMIP5 first? In its current formulation I cannot see how the information in Figure 10 helps us.

**This section has been removed, as we decided to keep the analysis of full complexity models, possibly CMIP6, for a more detailed future work.**

---

## Author Response (AR4)

Review of "Present and future synoptic circulation patterns associated with cold and snowy spells over Italy", version 9.3.2022, by M. D' Errico et al.

GENERAL COMMENTS

This is my fourth review of this manuscript.I believe that most of the major problems have been settled, but I still need to return to the bias correction issue. The authors do wisely when not applying it to Z500 andSLP, but the results in their Figs. 8-9 suggest that they should seriously reconsider the cases of T850 and T2M as well. In particular, Figures 9g-h show a completely unrealistic pattern with 2-m winter temperatures exceeding 25°C in Greenland, combined with an abrupt shift to more reasonable (5-10°C) values to the north-west of Iceland. Spurious local hotspots also appear elsewhere in the same maps, for example over the Gulf of Finland. I am convinced that the patterns in the original PlaSIM data are much smoother and more physically reasonable.The likely cause of this problem is the standard deviation correction included in the bias correction method. I assume that the transformation

$$T_{corr} = (\bar{T}_{obs} - \bar{T}_{ctrl}) + s_{obs}/s_{ctrl}(T - T_{ctrl})$$

has been used, where $T$ is the original and $T_{corr}$ the corrected temperature, $\bar{T}_{obs}$ and $\bar{T}_{ctrl}$ are the mean values in the observations and the control simulation, and $s_{obs}$ and $s_{ctrl}$ are the corresponding standard deviations.The problem is that the latter term, which is needed to ensure that the standard deviation in the control simulation agrees with observations, also affects the mean climate change in the model if $s_{obs}$ and $s_{ctrl}$ differ. Denoting the temperatures in a future RCP simulation as $T_{RCP}$, their mean value after the bias correction becomes

$$\bar{T}_{RCP,corr} = (\bar{T}_{obs} - \bar{T}_{ctrl}) + s_{obs}/s_{ctrl}(\bar{T}_{RCP} - \bar{T}_{ctrl})$$

whereas the mean for the control simulation is simply $\bar{T}_{corr} = \bar{T}_{obs}$
Taking the difference, we get

$$\bar{T}_{RCP,corr} - \bar{T}_{corr} = s_{obs}/s_{ctrl}(\bar{T}_{RCP} - \bar{T}_{ctrl}) - \bar{T}_{ctrl} = (s_{obs}/s_{ctrl} - 1)(\bar{T}_{RCP} - \bar{T}_{ctrl})$$

For example, if the standard deviation in the control simulation is too small, ($s_{ctrl}<s_{obs}$), the bias correction amplifies the change in the mean temperature and, therefore, also the highest temperatures.I suspect that the extremely high temperatures in (e.g.) Greenland in Figs. 9g-h result from this effect.To avoid this problem, a mean-conserving version of the bias correction should be used instead. When the corrected daily temperatures in the RCP simulations are defined as
$T_{RCP,corr}=(\overline{T}_{obs}+\overline{T}_{RCP}-\overline{T}_{ctrl})+s_{obs}/s_{ctrl}(T_{RCP}-\overline{T}_{RCP})$ then $\overline{T}_{RCP,corr}-\overline{T}_{corr}=(\overline{T}_{obs}+\overline{T}_{RCP}-\overline{T}_{ctrl})-\overline{T}_{obs}=\overline{T}_{RCP}-\overline{T}_{ctrl}$
just as expected.

Alternatively, the problems associated with the standard deviation correction could be avoided simply by omitting this correction, i.e., by only correcting the time mean bias.

**We thank once again the reviewer for the valuable suggestion, not having direct experience in bias correction of GCMs we did not foresee all of these issues.**

**Considering that we only need to correct temperatures in RCP scenarios to compute averages, and that the paper by Shrestha et al. 2017 suggest to use the simple mean shift, we now only shift the mean without further corrections on the variability. Indeed, the resulting fields are now more realistic. We changed the text accordingly (lines 218-220)**

**We are grateful for these comments, that helped us improve a procedure on which we did not have previous experience, and actually increased the quality of the methodology we used in our article.**

SPECIFIC COMMENTS

1.L122. Why 18? On L120 the numbers 22 and 10 are mentioned.

**Thank you, this was a typo, 22 and 10 are the correct numbers.**

2.Figure 4. The small numerical values indicate that the units are not as given in the figure headers (m and hPa). I assume that the values are non-dimensional.

3.Figure 5. Same comment as for Figure 4.

**Thank you, indeed we mistakenly added physical units to these plots that are in standard deviation units.**

4.L179-180. Earlier (L172-173) ten levels were mentioned, but here only five. Which is correct?

**Thank you for noticing the inconsistency, we used 10 levels, while 5 is the default setting. We have now corrected this in the text.**

5.L241. Z500 anomaly or standardized Z500 anomaly?

**We now specify that we refer to standardized anomalies.**

6.L248. $p_1$ or $\pi_{1,r}$? Table 1 seems to report both alternatives.

**$p_1$ is correct: we use $\pi_{1,r}$ to assess whether the frequency of an event changes, and in which direction; however, $p_1$ is still the tail probability used to define which future fields are analogues of the cluster of interest.**

7.L281-283. Do you see similar "winter heat waves" without using bias correction? The fields for T2M in Fig. 9 look so unphysical(particularly for RCP8.5) that I strongly suspect that they are an artifact of the standard deviation adjustment in your bias correction technique (see the general comments).

**No, the use of the simpler mean-shift bias correction shows a more coherent and expected temperature fields, where temperatures over Central-Northern Europe are much higher than today, but not higher than in the Mediterranean area. We removed this sentence from the article.**

8.L293. Consider putting Figures 11-15 in an Appendix or in Supplementary material. They take too much space from the main article compared with their information content, or the amount of text devoted to them.

**We moved Fig.s 11-15 to an Appendix**

9.Figures 11-15. It seems that the values are dimensionless, and not in the units indicated in the figure headings.

**Once again, we thank the reviewer for noticing that we mistakenly added physical units to these plots that are in standard deviation units.**

10.Caption of Figure 11. Root mean square difference in standardized geopotential height [standard deviations]? The same also applies to captions of Figs. 12-15.

**Thank you, we corrected this mistake linked to the one pointed out in the previous comment.**

TECHNICAL COMMENTS

**We thank the reviewer for the careful reading, we proceeded to fix all these typos that we did not manage to find in our last review of the article.**

1.L54. Section 4

2.L55. "Cold"with capital C

3.L99. Section 2.1

4.L102. Put "Jézéquel et al. 2018" in parentheses.

5.L141. hPa

6.L159. by high uncertainty

7.L197. the the

8.L232. persistence

9.Table 1. Why are 6-7 decimals used for RCP2.6, instead of just 4?

10.L297. "Similar" with capital S

11.L317. to be

12.L322. Temperatures are only shown in Figures 8-9

---

## Author Response (AR5)

**Dear Editor,**

**thank you very much for your guidance and assistance through the review process. We appreciate that you took the time to perform a careful reading of the manuscript, which has undergone several reviews and check again its consistency. We have addressed the remaining comments, you will find below our answers and the markup pdf shows the edits to the manuscript. Hopefully this version is suitable for publication in ESD,**

**best Regards,**
**Davide Faranda & Flavio Pons,**
**on behalf of the authors**

Dear Authors,
thank you for addressing the remaining reviewer comments. After a careful reading by myself I have a few more comments that I would like to ask you to address before the manuscript can be accepted for publication.
The comments are pasted below this message.
Best regards,
Jakob Zscheischler

L245: why does p need a subscript 1? Since there is no $p_2$, consider removing the subscript. Currently the subscript leads to some confusion e.g. in the first line of Table 1.

**Thank you for the suggestion, we now denote the critical probability value as $p^*$**

L248: shouldn't $\pi$ have the subscript i,r instead of 1,r, with i=1,2 for the two clusters?

**Thank you, this was indeed a typo we had not found before.**

L249: step 7, the definition of cold spell analogues is unclear. The equation is incorrect when p1 != $\pi_{1,r}$ and does not provide a selection of events. I guess you mean something like analogues should fulfil $g_{t,r}^{Z500} >= g_m^{Z500}$ where $g_m^{Z500}$ is the $p_1$ th quantile of $g_{t,r}^{Z500}$ in the modelled fields, i.e. $P(g_{t,r}^{Z500} >= g_m^{Z500}) = p_1$. Please adjust accordingly.

**Thank you. Indeed the formulation was not correct; however, the corrected version is different: $g_{t,r}^{Z500} >= g_c^{Z500}$, where $g_c^{Z500}$ is the threshold we used to find analogues within NCEP. This way, we embed the PlaSim Z500 fields in the phase space of NCEP, and we consider analogues those fields that fall within a hypersphere around the reference states defined by the cluster composites.**

Caption of Table 1: In step 7 in L249 cold spell analogues were defined to always have the same occurrence probability p1. The use of "analogues" for different sets of event sets thus creates ambiguity. Please clearly state for which analysis which set of analogues is used and consider using different terminology. (This point has already led to some confusion in the last round of reviews.)

**As pointed out in the editor's previous question, there was a mistake in step 7 of the procedure: we consider analogues the days characterized by a value of $g_{t,r}^{Z500}$ larger than the threshold value $g_c^{Z500}$ in the NCEP simulations. In practice, these are all the events, in each run of the model, that fall within the same hypersphere around the reference state in the phase space of NCEP. In our case, we do this for two reference states, corresponding to the composite Z500 fields of each cluster in the reanalysis.**

L274: I assume the analogues used here are those taken from step 7 in L249, which would be fewer than the ones summarised in Table 1? Or where have the analogues that were selected in step 7 been used otherwise? Please clarify.

**We have reformulated the sentence clarifying better this aspect:**
*"Figure \ref{fig:comp_zg} shows the composites of Z500 analogues fields in the CTRL (a,b) and RCP (c-h) runs for analogues of cluster 1 (a,c,e,g) and cluster 2 (b,d,f,h), found with the rule shown at step 7 of the procedure described above. Given that $\pi_{i,r} < p^*$ in all cases, these analogues are less than the (1-$p^*\%$) of the days in each run; however, given that each PlaSim run in 450 years long, they are more in absolute number."*